# Trade-off shapes diversity in eco-evolutionary dynamics

**Farnoush Farahpour[1]\*, Mohammadkarim Saeedghalati[1], Verena S Brauer[2], Daniel Hoffmann[1,3,4,5]\***

[1]Bioinformatics and Computational Biophysics, University of Duisburg-Essen, Essen, Germany; [2]Biofilm Center, University of Duisburg-Essen, Essen, Germany; [3]Center for Computational Sciences and Simulation, University of Duisburg-Essen, Essen, Germany; [4]Center for Medical Biotechnology, University of Duisburg-Essen, Essen, Germany; [5]Center for Water and Environmental Research, University of Duisburg-Essen, Essen, Germany

**Abstract** We introduce an Interaction- and Trade-off-based Eco-Evolutionary Model (ITEEM), in which species are competing in a well-mixed system, and their evolution in interaction trait space is subject to a life-history trade-off between replication rate and competitive ability. We demonstrate that the shape of the trade-off has a fundamental impact on eco-evolutionary dynamics, as it imposes four phases of diversity, including a sharp phase transition. Despite its minimalism, ITEEM produces a remarkable range of patterns of eco-evolutionary dynamics that are observed in experimental and natural systems. Most notably we find self-organization towards structured communities with high and sustained diversity, in which competing species form interaction cycles similar to rock-paper-scissors games.

DOI: https://doi.org/10.7554/eLife.36273.001

**\*For correspondence:**
farnoush.farahpour@uni-due.de (FF);
daniel.hoffmann@uni-due.de (DH)

**Competing interests:** The authors declare that no competing interests exist.

## Introduction

We observe an immense diversity in natural communities (*Hutchinson, 1961*; *Tilman, 1982*; *Huston, 1994*), but also in controlled experiments (*Maharjan et al., 2006*; *Gresham et al., 2008*; *Kinnersley et al., 2009*; *Herron and Doebeli, 2013*; *Kvitek and Sherlock, 2013*), where many species continuously compete, diversify and adapt via eco-evolutionary dynamics (*Darwin, 1859*; *Cody and Diamond, 1975*). However, the basic theoretical models (*Volterra, 1928*; *Tilman, 1982*) predict that both ecological and evolutionary dynamics tend to decrease the number of coexisting species by competitive exclusion or selection of the fittest. This apparent contradiction between observations and theory gives the stunning biodiversity in communities the air of a paradox (*Hutchinson, 1961*; *Sommer and Worm, 2002*) and hence has begotten a long, ongoing debate on the mechanisms underlying emergence and stability of diversity in communities of competitive organisms (*Hutchinson, 1959*; *Huston, 1994*; *Chesson, 2000*; *Sommer and Worm, 2002*; *Doebeli and Ispolatov, 2010*).

To identify candidate mechanisms that could resolve the problem of generation and maintenance of diversity, the basic theoretical ecological and evolutionary models have been extended by numerous features (*Chesson, 2000*; *Chave et al., 2002*), including spatial structure (*Mitarai et al., 2012*; *Villa Martín et al., 2016*; *Vandermeer and Yitbarek, 2012*), spatial and temporal heterogeneity (*Caswell and Cohen, 1991*; *Fukami and Nakajima, 2011*; *Hanski and Mononen, 2011*; *Kremer and Klausmeier, 2013*), tailored interaction network topologies (*Melián et al., 2009*; *Mougi and Kondoh, 2012*; *Kärenlampi, 2014*; *Laird and Schamp, 2015*; *Coyte et al., 2015*; *Grilli et al., 2017*), predefined niche width (*Scheffer and van Nes, 2006*; *Doebeli, 1996*), adjusted mutation-selection rate (*Johnson, 1999*; *Desai and Fisher, 2007*), and life-history trade-offs

**eLife digest** A patch of rain forest, a coral reef, a pond, and the microbes in our guts are all examples of biological communities. More generally, a community is a group of organisms that live together at the same place and time. Many communities are composed of a large number of different species, and this diversity is maintained for long times.

Although diversity is a key feature of biological communities, the mechanisms that generate and maintain diversity are not well understood. Research had hinted at links between diversity and the trade-offs that species are subject to. For instance, there is a trade-off between competitiveness and reproduction: if there are limited resources in the environment a species may either produce many offspring that are not very competitive, or fewer, more competitive offspring.

Farahpour et al. have now simulated the development of communities of organisms that reproduce, compete, and die in a uniform environment. Crucially, these computational simulations introduced a trade-off between competitive ability and reproduction.

The simulations show that the form of trade-off has a fundamental impact on diversity: moderate trade-offs favor diversity, whereas extreme trade-offs suppress diversity. The simulations also revealed mechanisms that underlie how diversity is generated. In particular, cyclic relationships emerge where one species dominates another but is also dominated by a third, similar to the rock-paper-scissors game.

Since Farahpour et al. used a bare-bone model with only a few essential features the results could apply to a larger class of community-like systems whose evolution is driven by competition. This includes economic and social systems as well as biological communities.

DOI: https://doi.org/10.7554/eLife.36273.002

(*Rees, 1993*; *Bonsall et al., 2004*; *de Mazancourt and Dieckmann, 2004*; *Gudelj et al., 2007*; *Ferenci, 2016*; *Posfai et al., 2017*). However, it is still unclear which features are essential to explain biodiversity. For instance, diversity is also observed under stable and homogeneous conditions (*Gresham et al., 2008*; *Kinnersley et al., 2009*; *Maharjan et al., 2012*; *Herron and Doebeli, 2013*; *Kvitek and Sherlock, 2013*).

So far, models of eco-evolutionary dynamics have been developed in three major categories: models in genotype space, like population genetics (*Ewens, 2012*) and quasispecies models (*Nowak, 2006*); models in phenotype space, like adaptive dynamics (*Doebeli, 2011*) and webworld models (*Drossel et al., 2001*); and models in interaction space, like Lotka-Volterra models (*Coyte et al., 2015*; *Ginzburg et al., 1988*) and evolving networks (*Mathiesen et al., 2011*; *Allesina and Levine, 2011*). Each of these categories has strengths and limitations and emphasizes particular aspects. However, in nature these aspects are entangled by eco-evolutionary feedbacks that link genotype, phenotype, and interaction levels (*Post and Palkovacs, 2009*; *Schoener, 2011*; *Ferriere and Legendre, 2013*; *Weber et al., 2017*). In a closed system of evolving organisms mutations, that is, evolutionary changes at the genetic level (*Figure 1a*), can cause phenotypic variations if they are mapped to novel phenotypic traits in phenotype space (*Figure 1b*)(*Soyer, 2012*). These variations have ecological impact only if they affect biotic or abiotic interactions of species (*Figure 1c*); otherwise they are ecologically neutral. The resulting adaptive variations in the interaction network change the species composition through population dynamics. Finally, frequency-dependence occasionally selects strategies that adapt species to their new environment (*Schoener, 2011*; *Moya-Laraño et al., 2014*; *Hendry, 2016*; *Weber et al., 2017*).

Thus, we have a link from interactions to eco-evolutionary dynamics, suggesting that we do not need to follow all evolutionary changes at the genetic or phenotypic level if we are interested in macro-eco-evolutionary dynamics, but only those changes that affect interactions. In this picture, evolution can be considered as an exploration of interaction space, and modeling at this level can help us to study how complex competitive interaction networks evolve and shape diversity. This neglect of genetic and phenotypic details in interaction-based models (*Ginzburg et al., 1988*; *Solé, 2002*; *Tokita and Yasutomi, 2003*; *Shtilerman et al., 2015*) equals a coarse-graining of the eco-evolutionary system (*Figure 1*). This coarse-graining not only reduces complexity but it should also make the approach applicable to a broader class of biological systems.

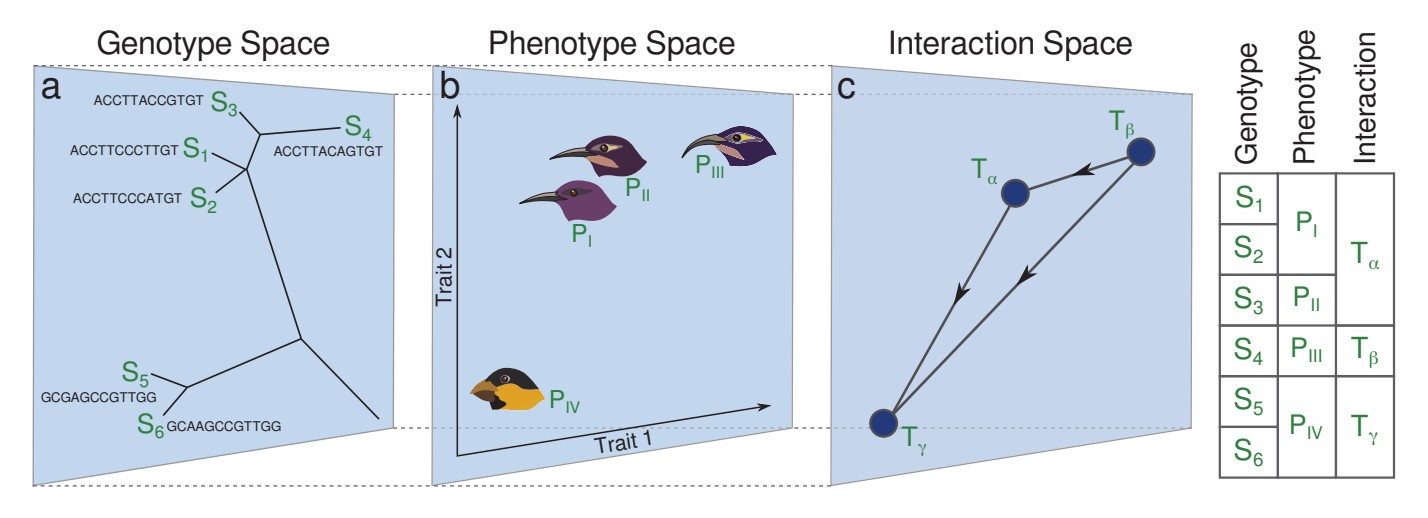

**Figure 1.** Link between genotype, phenotype and interaction space. This schematic shows species in a community of grain-eating and nectar-feeding birds, living in an environment where nectar feeding is advantageous. (a) Six different genotypes (sequences $S_i$) on a distance tree genotype space. (b) Four distinct phenotypes, $P_j$, are present in this space. Genotypes $S_1$ and $S_2$ are mapped to the same phenotype $P_I$, and $S_5$ and $S_6$ are mapped to the same phenotype $P_{IV}$. (c) Interaction space distinguishes only three interaction traits $T_k$ (for definition see Model section below). $P_I$ and $P_{II}$ are mapped to the same interaction trait $T_\alpha$ because the change of feather color does not affect ecological interactions regarding the feeding habit. The table on the right shows how the complexity of the description reduces as we map the system to interaction space.
DOI: https://doi.org/10.7554/eLife.36273.003

Interaction-based evolutionary models have received some attention in the past (*Ginzburg et al., 1988*; *Solé, 2002*) but then were almost forgotten, despite remarkable results. We think that these works have pointed to a possible solution of a hard problem: The complexity of evolving ecosystems is immense, and it is therefore difficult to find a representation suitable for the development of a statistical mechanics that enables qualitative and quantitative analysis (*Weber et al., 2017*). Modeling at the level of interaction traits, rather than modeling of detailed descriptions of genotypes or phenotypes, coarse-grains these complex systems in a natural way so that this approach may be helpful for developing a biologically meaningful statistical mechanics.

The first eco-evolutionary interaction-based model was introduced by *Ginzburg et al. (1988)* based on Lotka–Volterra dynamics for competitive communities. Instead of adding species characterized by random coefficients, taken out of some arbitrary species pool, they made the assumption that a new mutant should be ecologically similar to its parent, which means that phenotypic variations that are not ecologically neutral generate mutants that interact with other species similar to their parents (*Figure 1*). Thus, speciation events were simulated as ecologically continuous mutations in the strength of competitive interactions. This model, although conceptually progressive, was not able to produce a large stable diversity, possibly because diversity requires components not included in this model. Therefore subsequent interaction-based models supplemented it with *ad hoc* features to specifically increase diversity, such as special types of mutations (*Tokita and Yasutomi, 2003*), addition of mutual interactions (*Tokita and Yasutomi, 2003*; *Yoshida, 2003*), enforcement of partially connected interaction graphs (*Kärenlampi, 2014*), or imposed parent-offspring niche separation (*Shtilerman et al., 2015*). While these models generated, as expected, higher diversity than the original Ginzburg model, they could not reproduce key characteristics of real systems, for example emergence of large and stable diversity, diversification to separate species and mass extinctions. Of course, the use of *ad hoc* features that deliberately increase diversity also cannot explain why diversity emerges.

An essential component missing in the previous interaction-based models had been a constraint on strategy adoption. In real systems such constraints prevent the emergence of Darwinian Demons, that is, species that develop in the absence of any restriction and act as a sink in the network of population flow. Among all investigated features responsible for diversity, mentioned above, life-history trade-offs that regulate energy investment in different life-history strategies are fundamentally

imposed by physical laws such as energy conservation or other thermodynamic constraints, and thus present in any natural system (*Stearns, 1989*; *Gudelj et al., 2007*; *Del Giudice et al., 2015*). These physical laws constrain evolutionary trajectories in trait space of evolving organisms and determine plausible evolutionary paths (*Fraebel et al., 2017*; *Ng'oma et al., 2017*), i.e. combinations of strategies adopted or abandoned over time. Roles of trade-offs for emergence and stabilization of diversity have been investigated in previous eco-evolutionary studies (*Posfai et al., 2017*; *Rees, 1993*; *Bonsall et al., 2004*; *de Mazancourt and Dieckmann, 2004*; *Ferenci, 2016*; *Gudelj et al., 2007*) and experiments (*Stearns, 1989*; *Kneitel and Chase, 2004*; *Agrawal et al., 2010*; *Maharjan et al., 2013*; *Ferenci, 2016*). It has been shown, for example, that if metabolic trade-offs are considered, even at equilibrium and in homogeneous environments, stable coexistence of species becomes possible (*Gudelj et al., 2007*; *Beardmore et al., 2011*; *Maharjan and Ferenci, 2016*).

Here, we introduce a new, minimalist model, the *Interaction and Trade-off-based Eco-Evolutionary Model* (ITEEM), with simple and intuitive eco-evolutionary dynamics at the interaction level that considers a life-history trade-off between interaction traits and replication rate, that means, better competitors replicate less (*Jakobsson and Eriksson, 2003*; *Bonsall et al., 2004*). To our knowledge, ITEEM is the first model which joins these two elements, the interaction-space description with a life-history trade-off, that we deem crucial for an understanding of eco-evolutionary dynamics. We use ITEEM to study development of communities of organisms that diversify from one ancestor by gradual changes in their interaction traits and compete under Lotka-Volterra dynamics in well-mixed, closed system.

We show that ITEEM dynamics, without any *ad hoc* assumption, not only generates large and complex biodiversity over long times (*Herron and Doebeli, 2013*; *Kvitek and Sherlock, 2013*) but also closely resembles other observed eco-evolutionary dynamics, such as sympatric speciation (*Tilmon, 2008*; *Bolnick and Fitzpatrick, 2007*; *Herron and Doebeli, 2013*), emergence of two or more levels of differentiation similar to phylogenetic structures (*Barraclough et al., 2003*), occasional collapses of diversity and mass extinctions (*Rankin and López-Sepulcre, 2005*; *Solé, 2002*), and emergence of cycles in interaction networks that facilitate species diversification and coexistence (*Buss and Jackson, 1979*; *Hibbing et al., 2010*; *Maynard et al., 2017*). Interestingly, the model shows a unimodal ('humpback') course of diversity as function of trade-off, with a critical trade-off at which biodiversity undergoes a phase transition, a behavior observed in nature (*Kassen et al., 2000*; *Smith, 2007*; *Vallina et al., 2014*; *Nathan et al., 2016*). By changing the shape of trade-off and comparing the results with a no-trade-off model, we show that diversity is a natural outcome of competition if interacting species evolve under physical constraints that restrict energy allocation to different strategies. The natural emergence of diversity from a bare-bone eco-evolutionary model suggests that a unified treatment of ecology and evolution under physical constraints dissolves the apparent paradox of stable diversity.

## Model

ITEEM is an individual-based model (*Black and McKane, 2012*; *DeAngelis and Grimm, 2014*) with simple intuitive updating rules for population and evolutionary dynamics. A simulated system in ITEEM has $N_s$ sites of undefined spatial arrangement (no neighborhood), each providing permanently a pool of resources that is sufficient for the metabolism of one organism. The community is well-mixed, which means that the probability for an encounter is the same for all pairs of individuals, and that the probability of an individual to enter a site (i.e. to access resources) is the same for all individuals and sites.

We start an eco-evolutionary simulation with individuals of a single strain occupying a fraction of the $N_s$ sites, and then carry out long simulations for millions of generations. Note that in the following, to facilitate discourse, we use the term *strain* for a group of individuals with identical traits, whereas the term *species* denotes a monophyletic cluster of strains with some intraspecific diversity (for a discussion on application of these terms in this study see Appendix 1, Species and strains). Over time $t$, measured in generations, the number of individuals, $N_{ind}(t)$, number of strains, $N_{st}(t)$, and number of species, $N_{sp}(t)$, change by ecological (birth, death, competition) and evolutionary dynamics (mutation, extinction, diversification).

Every generation or time step consists of $N_s$ sequential replication trials of randomly selected individuals, followed at the end by a single death step. In the death step all individuals that have

reached their lifespan at that generation will vanish. Lifespans of individuals are drawn at their births from a Poisson distribution with overall fixed mean lifespan $\lambda$. This is equivalent to an identical per capita death rate for all strains. For comparison, simulations with no attributed lifespan ($\lambda = \infty$) were carried out, too; in this case the only cause of death is defeat in a competitive encounter.

At each replication trial, a randomly selected individual of a strain $\alpha$ can replicate with probability $r_\alpha$. Age of individuals plays no role in their reproduction and thus a newborn individual can be selected and replicate with the same probability as adult individuals. With a fixed probability $\mu$ the offspring mutates to a new strain $\alpha'$. Then, the newborn individual is assigned to a randomly selected site. If the site is empty, the new individual will occupy it. If the site is already occupied, the new individual competes with the current holder in a life-or-death struggle. In that case, the surviving individual is determined probabilistically by the 'interaction' $I_{\alpha\beta}$, defined for each pair of strains $\alpha$, $\beta$. $I_{\alpha\beta}$ is the *survival probability* of an $\alpha$ individual in a competitive encounter with a $\beta$ individual, with $I_{\alpha\beta} \in [0, 1]$ and $I_{\alpha\beta} + I_{\beta\alpha} = 1$ (*Grilli et al., 2017*). All interactions $I_{\alpha\beta}$ form an interaction matrix $\mathbf{I}(t)$ that encodes the outcomes of all possible competitive encounters in this probabilistic sense. Row $\alpha$ of $\mathbf{I}$ defines the 'interaction trait' $\mathbf{T}_\alpha = (I_{\alpha 1}, I_{\alpha 2}, \ldots, I_{\alpha N_{st}(t)})$ of strain $\alpha$, with $N_{st}(t)$ the number of strains at time $t$.

If strain $\alpha$ goes extinct, its interaction elements must be removed, i.e. the $\alpha$th row and column of $\mathbf{I}$ are deleted. Conversely, if a mutation of $\alpha$ generates a new strain $\alpha'$, its trait vector is obtained by adding a small random variation to the parent trait, that is $\mathbf{T}_{\alpha'} = \mathbf{T}_\alpha + \boldsymbol{\eta}$, where $\boldsymbol{\eta} = (\eta_1, \cdots, \eta_{N_{st}(t)})$ is a vector of independent random variations, drawn from a zero-centered normal distribution of fixed width $m$. With this, $\mathbf{I}$ grows by one row and column. The new elements of the matrix are:

$$I_{\alpha'\beta} = I_{\alpha\beta} + \eta_\beta ,$$

$$I_{\beta\alpha'} = 1 - I_{\alpha'\beta} ,$$

$$I_{\alpha'\alpha'} = 0.5 , \tag{1}$$

where $\beta = 1, \cdots, N_{st}(t)$ and thus $\alpha'$ inherits its interactions from $\alpha$, but with a small random modification. Evolutionary variations in ITEEM generate mutants that are ecologically similar to their parents. Such variations can represent any phenotypic variation that influences interactions of strains with their community and thus changes their relative competitive abilities (*Thompson, 1998*; *Thorpe et al., 2011*; *Bergstrom and Kerr, 2015*; *Thompson, 1999*). With *Equation 1* we assume that all the interaction terms of the new mutant can change independently.

To implement trade-off between competitive ability and fecundity, we introduce a relation between competitive ability $C$, defined as average interaction

$$C(\mathbf{T}_\alpha) = \frac{1}{N_{st}(t) - 1} \sum_{\beta \neq \alpha} I_{\alpha\beta}, \tag{2}$$

and replication $r_\alpha$ (for fecundity). When $N_{st} = 1$, competitive ability of that single strain is set to zero. To study the influence of trade-off between competitive ability and replication, we systematically change its shape by varying a parameter $\delta$ ($0 \leq \delta < 1$) (*Figure 2*). For details of trade-off function and its effect on trait distribution and relative fitness see Appendix 1, Trade-off. Trade-off functions can be concave ($\delta < 0.5$), linear ($\delta = 0.5$), or convex ($\delta > 0.5$). The trade-off function ties better competitive ability to lower fecundity and vice versa. The extreme case $\delta = 0$ makes $r = 1$ and thus independent of $C$, which means no trade-off.

We compare ITEEM results to the corresponding results of a neutral model (*Hubbell, 2001*), where we have formally evolving trait vectors $\mathbf{T}_\alpha$ but fixed and uniform replication probabilities and interactions. Accordingly, the neutral model has no trade-off.

ITEEM belongs to the well-established class of generalized Lotka-Volterra (GLV) models in the sense that the population-level approximation of the stochastic, individual-based ecological dynamics of ITEEM leads to the competitive Lotka-Volterra equations (Appendix 1, Generalized Lotka–Volterra (GLV) equation). Thus the results of the model can be interpreted in the framework of competitive GLV equations that model competition for a renewable resource pool and summarize all

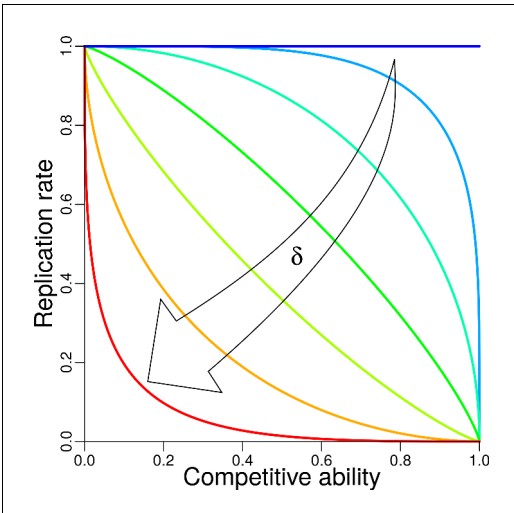

**Figure 2.** Trade-off between replication and competitive ability. The shape of trade-off is controlled by trade-off parameter $\delta$ (Appendix 1, Trade-off). Trade-off functions with $\delta = 0, 0.14, 0.29, 0.43, 0.57, 0.71, 0.86$ are plotted, each in different color, from the dark blue horizontal line for $\delta = 0$ (i.e. no trade-off), to the red convex curve for $\delta = 0.86$.

DOI: https://doi.org/10.7554/eLife.36273.004

types of competition (*Gill, 1974*; *Maurer, 1984*) in the elements of the interaction matrix **I** (see above), i.e. these elements represent the resultant negative effect of all competitor populations on each other.

Our model also allows to study speciation in terms of network dynamics. The interaction matrix **I** defines a complete dominance network between coexisting strains. In this network the nodes are strains $(\alpha, \beta)$, and the directed edges connecting them indicate direction and strength of dominance, i.e. sign and size of $I_{\alpha\beta} - I_{\beta\alpha}$, respectively. Thus, the elements of the weighted adjacency matrix of this network are defined as either $W_{\alpha\beta} = I_{\alpha\beta} - I_{\beta\alpha}$, if $\alpha$ is the superior competitor in the pairwise encounter with $\beta$ ($I_{\alpha\beta} > I_{\beta\alpha}$), or otherwise as $W_{\alpha\beta} = 0$. With this definition all $W_{\alpha\beta}$ are in $[0, 1]$. Accordingly, for the dominance network of *species*, we computed directed edges between any two *species*, $i$ and $j$, by averaging over edges between all pairs of *strains* belonging to these species, that is $W_{ij}^{sp} = \bar{W}_{\alpha\beta}$ for all strains $\alpha$ and $\beta$ in the $i$th and $j$th species, respectively. The strength and direction of dominance edges indicate the effective flow of population between species.

As we consider a trade-off between replication and competitive ability in the framework of GLV equations, we can distinguish between $r$- and $\alpha$-selection (*Gill, 1974*; *Kurihara et al., 1990*; *Masel, 2014*). $r$-selection selects for reproductive ability, which is beneficial in low density regimes, while $\alpha$-selection selects for competitive ability and is effective at high density regimes under frequency-dependent selection. $\alpha$-selection, first introduced by Gill (*Gill, 1974*), can be realized by acquisition of any kind of ability or mechanism that increases the chance of an organism to take over resources, to prevent competitors from gaining resources (*Gill, 1974*), or helps the organism to tolerate stress or reduction of contested resource availability (*Aarssen, 1984*). $\alpha$-selection is different from $K$-selection; although both are effective at high density, the latter is limited to investments in efficient and parsimonious usage of resources (*Masel, 2014*).

The source code of the ITEEM model is freely available at GitHub (*Farahpour, 2018*; copy archived at https://github.com/elifesciences-publications/ITEEM).

## Results

### Generation of diversity

Our first question was whether ITEEM is able to generate and sustain diversity. Since we have a well-mixed system with initially only one strain, a positive answer implies sympatric diversification: the emergence of new species by evolutionary branching without geographic isolation or resource partitioning. In fact, we observe that during long-time eco-evolutionary trajectories in ITEEM new, distinct species emerge, and their coexistence establishes a sustained high diversity in the system (*Figure 3a*).

Remarkably, the emerging diversity has a clear hierarchical structure in the phylogeny tree and trait space: at the highest level we see that the phylogenetically separated strains (*Figure 3a* and Appendix 1, Species and strains) appear as well-separated clusters in trait space (*Figure 3b*) similar to biological *species*. Within these clusters there are sub-clusters of individual strains (*Barraclough et al., 2003*). Both levels of diversity can be quantitatively identified as levels in the distribution of branch lengths in minimum spanning trees in trait space (Appendix 1, SMST and

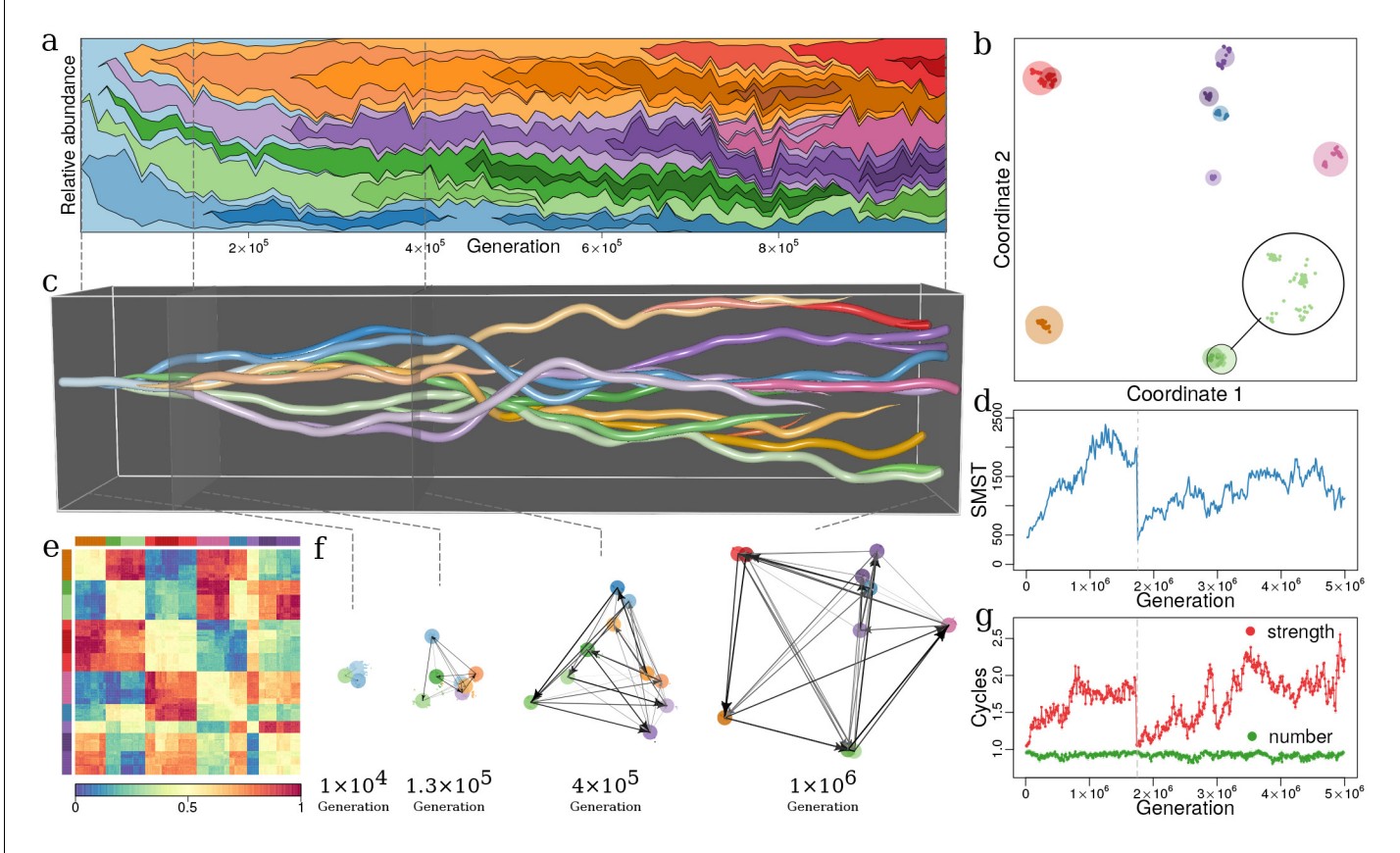

**Figure 3.** Evolutionary dynamics of a community driven by competitive interactions, with trade-off between fecundity and competitive abilities ($\delta = 0.5$, $\lambda = 300$, $\mu = 0.001$, $m = 0.02$, $N_s = 10^5$). (a) Species' frequencies over time (Muller plot): one color per species, vertical width of each colored region is the relative abundance of respective species. Frequencies are recorded every $10^4$ generations over $10^6$ generations. The plot was produced with R-package MullerPlot (*Farahpour et al., 2016*). (b) Distribution over trait space: Snapshot of distribution of strains and species in trait space after $10^6$ generations. By using classical multidimensional scaling the multidimensional trait space is reduced to two dimensions that explain most of the variance in trait space (see Appendix 1, Classical multi-dimensional scaling (CMDS)). Points and discs are strains and species, respectively (see Appendix 1, Species and strains). Magnified disc in lower right corner shows strains in the light green species disc. Discs diameter are proportional to the total abundance of corresponding species, i.e. the sum of relative abundances of all strains that belong to that species. In this snapshot $N_{st} = 660$ and $N_{sp} = 10$. (c) Evolutionary dynamics in trait space: Snapshots as in panel (b), but concatenated for all times (horizontal axis), from the monomorphic first generation to generation $10^6$. *Figure 3—video 1* shows this evolutionary dynamics over time. (d) Functional diversity over time (see Appendix 1, Diversity indexes and parameters of dynamics) measured by the size of minimum spanning tree (SMST) in interaction trait space (see Appendix 1, SMST and distribution of species and strains in trait space). At $1.75 \times 10^6$ generations diversity collapses with all species but one going extinct (vertical dashed line) (Appendix 1, Collapses of diversity). (e) Heatmap of interaction matrix **I** for generation $10^6$. Row and column order reflects species consistent with panel (b) and indicated by color bars along top and left. Colors inside heat map represent values of interaction terms (color-key along bottom). (f) Evolution of dominance network: several snapshots from panel (c) with dominance edges, $W_{ij}^{sp}$ between species (colored discs). (g) Numbers and mean strength of cycles over time in green and red, respectively. The strength of a cycle is defined by its weakest edge. Number and mean strength are given in units of number and mean strength of equivalent random networks, respectively (Appendix 1, Intransitive dominance cycles). Right ends in (a) and (c) correspond to generation panel (b) and (e). Colors of species are the same in panels (a), (b), (c), (e) and (f). Note that time scales differ between panels (a), (c) and (d), (g).

DOI: https://doi.org/10.7554/eLife.36273.005

The following video is available for figure 3:

**Figure 3—video 1.** Divergent eco-evolutionary dynamics in interaction trait space.

DOI: https://doi.org/10.7554/eLife.36273.006

distribution of species and strains in trait space). This hierarchical diversity is reminiscent of the phylogenetic structures in biology (*Barraclough et al., 2003*).

Overall, the model shows evolutionary divergence from one ancestor to several species consisting of a total of hundreds of coexisting strains (*Figure 3c*). This evolutionary divergence in interaction

space is the result of frequency-dependent selection without any further assumption on the competition function, for example a Gaussian or unimodal competition kernel (*Dieckmann and Doebeli, 1999*; *Doebeli and Ispolatov, 2010*), or predefined niche width (*Scheffer and van Nes, 2006*). In the course of this diverging sympatric evolution, diversity measures typically increase and, depending on trade-off parameter $\delta$, high diversity is sustained over hundreds of thousands of generations (*Figure 3d*, and Appendix 1, Diversity over time). This observation holds for several complementary measures of diversity, no matter whether they are based on abundance of strains or species, or on functional diversity, i.e. quantities that measure the spread of the population in trait space (Appendix 1, Functional diversity (FD), functional group and functional niche).

The observed pattern of divergence contradicts the long-held view of sequential fixation in asexual populations (*Muller, 1932*). Instead, we see frequently concurrent speciation with emergence of two or more species in quick succession (*Figure 3a*), in agreement with recent results from long-term bacterial and yeast cultures (*Herron and Doebeli, 2013*; *Maddamsetti et al., 2015*; *Kvitek and Sherlock, 2013*).

ITEEM systems self-organize toward structured communities: the interaction matrix of a diverse system obtained after many generations has a conspicuous block structure with groups of strains with similar interaction strategies (*Figure 3e*), and these groups being well-separated from each other in trait space (*Figure 3b*) (*Sander et al., 2015*). This fact can be interpreted in terms of functional organization as the interaction trait in ITEEM directly determines the functions of strains and species in the community (Appendix 1, Functional diversity (FD), functional group and functional niche). This means that the block structure in *Figure 3e* corresponds to self-organized, well-separated functional niches (*Whittaker et al., 1973*; *Rosenfeld, 2002*; *Taillefumier et al., 2017*), each occupied by a cluster of closely related strains. This niche differentiation among species, which facilitates their coexistence, is the result of frequency-dependent selection among competing strategies. Within each functional niche the predominant dynamics, determining relative abundances of strains in the niche, is neutral. Speciation can occur when random genetic drift in a functional group generates sufficiently large differences between the strategies of strains in that group, and then selection forces imposed by biotic interactions reinforce this nascent diversification by driving strategies further apart.

We observe as characteristic of the dynamics of the dominance network $W$ (see Model) the appearance of strong edges as diversification increases trait distance (or dissimilarity) between species (*Figure 3f*) (*Anderson and Jensen, 2005*).

## Emergence of intransitive cycles

Three or more directed edges in the dominance network can form cycles of strains in which each strain competes successfully against one cycle neighbor but loses against the other neighbor, a configuration corresponding to rock-paper-scissors games (*Szolnoki et al., 2014*). Such intransitive dominance relations have been observed in nature (*Buss and Jackson, 1979*; *Sinervo and Lively, 1996*; *Lankau and Strauss, 2007*; *Bergstrom and Kerr, 2015*), and it has been shown that they stabilize a system driven by competitive interactions (*Allesina and Levine, 2011*; *Mathiesen et al., 2011*; *Mitarai et al., 2012*; *Laird and Schamp, 2015*; *Maynard et al., 2017*; *Gallien et al., 2017*). We find in ITEEM networks that the increase of diversity coincides with growth of mean strength of cycles (*Figure 3d,g* and Appendix 1, Intransitive dominance cycles). Note that these cycles emerge and self-organize in the evolving ITEEM networks without any presumption or constraint on network topology.

Formation of strong cycles could also hint at a mechanistic explanation for another phenomenon that we observe in long ITEEM simulations: Occasionally diversity collapses from medium levels abruptly to very low levels, usually followed by a recovery (*Figure 3d*). Remarkably, dynamics before these mass extinctions are clear exceptions of the generally strong correlation of diversity and average cycle strength. While the diversity immediately before mass extinctions is inconspicuous, these events are always preceded by exceptionally high average cycle strengths (Appendix 1, Collapses of diversity). Because of the rarity of mass extinctions in our simulations we currently have not sufficient data for a strong statement on this phenomenon, however, it is conceivable that the emergence of new species in a system with strong cycles likely leads to frustrations, i.e. the newcomers cannot be accommodated without inducing tensions in the network, and these tensions can destabilize the network and discharge in a collapse. The extinction of a species in a network with strong cycles will

probably have a similar effect. This explanation of mass extinctions would be consistent with related works where collapses of diversity occur if maximization of competitive fitness (here: by the new-comer species) leads to a loss of absolute fitness (here: break-down of the network) (*Matsuda and Abrams, 1994*; *Masel, 2014*). This is a special case of the tragedy of the commons (*Hardin, 1968*; *Masel, 2014*) that happens when competing organisms under frequency-dependent selection exploit shared resources (*Rankin and López-Sepulcre, 2005*), as it is the case in ITEEM.

## Impact of trade-off and lifespan on diversity

The eco-evolutionary dynamics described above depends on lifespan and trade-off between replication and competitive ability. This becomes clear if we study properties of dominance network and trait diversity. *Figure 4a* relates properties of the dominance network to the trade-off parameter $\delta$, at fixed lifespan $\lambda$. Specifically, we plot two indicators of community structure against trade-off parameter $\delta$, namely mean weight of dominance edges $\langle W \rangle$, and mean strength of cycles $\rho$. *Figure 4b* summarizes the behavior of diversity as function of $\delta$ and lifespan $\lambda$. For this summary, we chose ten parameters that quantify different aspects of diversity, for example richness, evenness, functional diversity, and trait distribution, and then averaged over their normalized values to obtain

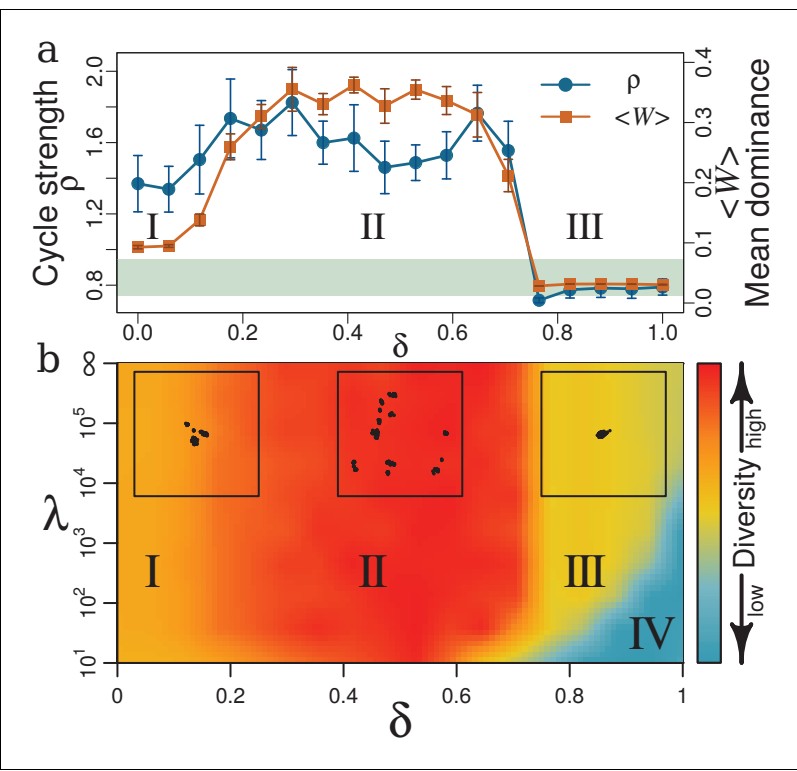

**Figure 4.** Effects of trade-off $\delta$ and lifespan $\lambda$ on community structure and diversity. (a) Mean weight of dominance edges $\langle W \rangle$ (orange squares) and mean strength of cycles $\rho$ (blue circles) as function of $\delta$. Mean cycle strength is given in units of mean strength of corresponding random networks for the respective trade-off (Appendix 1, Intransitive dominance cycles). Points in panel (a) are evaluated as averages over three different simulations, each over $5 \times 10^6$ generations with $\mu = 0.001$, $m = 0.02$, $\lambda = \infty$ and $N_S = 10^5$. Error bars are standard deviations averaged over these three simulations. The shaded area marks mean strength of cycles for a neutral model with corresponding parameters $\pm$ standard deviation. (b) Phase diagram of diversity as function of trade-off $\delta$ and lifespan $\lambda$. Diversity (represented by color spectrum defined in the color bar) is given as consensus of several quantities (Appendix 1, Diversity indexes and parameters of dynamics for different trade-offs and lifespans). Diversity has four distinct phases (I–IV). Insets along the top margin are representative MDS plots (Appendix 1, Classical multi-dimensional scaling (CMDS)) of strain distributions in trait space, with $\lambda = 10^5$ but different values of $\delta$ (left to right: I with $\delta = 0.11$; II with $\delta = 0.5$; III with $\delta = 0.89$). Panel (a) corresponds to a horizontal cross-section through the phase diagram in panel (b) with $\lambda = \infty$ for $\langle W \rangle$ and $\rho$ as indicators of community structure.
DOI: https://doi.org/10.7554/eLife.36273.007

an overall measure of diversity (color bar in the figure). The full set of parameters is detailed in Appendix 1, Diversity indexes and parameters of dynamics for different trade-offs and lifespans. The resulting phase diagram gives us an overview of the community diversity for different trade-off parameters $\delta$ and lifespans $\lambda$. The diagram shows a weak dependency of diversity on $\lambda$ and a strong impact of $\delta$, with four distinct phases (I-IV) from low to high $\delta$ as described in the following.

Without trade-off ($\delta = 0$), strains do not have to sacrifice replication for better competitive abilities. Any resident community can be invaded by a new mutant with relatively higher $C$ that does not have to compensate with a lower $r$. These mutants resemble Darwinian Demons (*Law, 1979*), i.e. strains or species that can maximize all aspects of fitness (here $C$ and $r$) simultaneously and would exist under physically unconstrained evolution. Such Darwinian Demons can then be outcompeted by their own mutant offspring's that have higher $C$ and the same $r$. Thus we have sequential predominance of such strategies with constantly changing traits and improving competitiveness, but no diverse network emerges. As we increase $\delta$ from this unrealistic extreme into phase I ($0 < \delta \leq 0.2$) coexistence is facilitated. However, the small $\delta$ still favors investing in relatively higher competitive ability as a low-cost strategy to increase fitness. In this phase $\langle W \rangle$ and $\rho$ (*Figure 4a*) slightly increase: biotic selection pressure exerted by inter-species interactions starts to generate diverse communities (left inset in *Figure 4b*, Appendix 1, Diversity indexes and parameters of dynamics for different trade-offs and lifespans).

When $\delta$ increases further (phase II), trade-off starts to force strains to choose between higher replication or better competitive abilities. Extremes of these quantities do not allow for viable species: sacrificing $r$ completely for maximum $C$ stalls population dynamics, whereas maximum $r$ leads to inferior $C$. Thus strains seek middle ground values in both $r$ and $C$. The nature of $C$ as mean of interactions (*Equation 2*) allows for many combinations of interaction traits with approximately the same mean. Thus, in a middle range of $r$ and $C$, many strategies with the same overall fitness are possible, which is a condition of diversity (*Marks and Lechowicz, 2006*). From this multitude of strategies, sets of trait combinations emerge in which strains with different combinations keep each other in check, for example by the competitive rock-paper-scissors-like cycles between species described above. An equivalent interpretation is the emergence of diverse sets of non-overlapping compartments or functional niches in trait space (*Figure 3b,e*). Diversity in this phase II is the highest and most stable (middle inset in *Figure 4b*, Appendix 1, Diversity indexes and parameters of dynamics for different trade-offs and lifespans).

As $\delta$ approaches 0.7, $\langle W \rangle$ and $\rho$ plummet (*Figure 4a*) to interaction values comparable to the noise level $m$ (see Model), and a cycle strength typical for the neutral model (horizontal light green ribbon in *Figure 4a*), respectively. The sharp drop of $\langle W \rangle$ and $\rho$ at $\delta \approx 0.7$ is reminiscent of a phase transition. As expected for a phase transition, the steepness increases with system size (Appendix 1, Size of the system). For $\delta \geq 0.7$, weights of dominance edges never grow and no structures, for example cycles, emerge. Diversity remains low and close to that of a neutral system. The sharp transition at $\delta \approx 0.7$ which is visible in practically all diversity measures (between phases II and III in *Figure 4b*, see also Appendix 1, Diversity indexes and parameters of dynamics for different trade-offs and lifespans) is a transition from a system dominated by biotic selection pressure to a neutral system. In high trade-off phase III, a small relative change in $C$ produces a large relative change in $r$ (Appendix 1, Strength of trade-off function). For instance, given a resident strain $R$ with $r$ and $C$, a closely related mutant $M$ increases the fitness by adopting a relatively high $r$ while paying a relatively small penalty in $C$ (see Appendix 1, Strength of trade-off function for the relative impacts of the traits), and therefore will invade $R$. Thus, diversity in phase III will remain stable and low, and is characterized by a group of similar strains with no effective interaction and hence no diversification to distinct species (right inset in *Figure 4b* and Appendix 1, Diversity indexes and parameters of dynamics for different trade-offs and lifespans). In this high trade-off regime, lifespan comes into play: here, decreasing $\lambda$ can make lives too short for replication. These hostile conditions minimize diversity and favor extinction (phase IV).

## Trade-off, resource availability, and diversity

There is a well-known but not well understood unimodal relationship ('humpback curve') between biomass productivity and diversity: diversity as function of productivity has a convex shape with a maximum at middle values of productivity (*Smith, 2007*; *Vallina et al., 2014*). This productivity-diversity relation has been reported at different scales in a wide-range of natural communities, for

example phytoplankton assemblages (*Vallina et al., 2014*), microbial (*Kassen et al., 2000*; *Horner-Devine et al., 2003*; *Smith, 2007*), plant (*Guo and Berry, 1998*; *Michalet et al., 2006*), and animal communities (*Bailey et al., 2004*). This behavior is reminiscent of horizontal sections through the phase diagram in *Figure 4b*, though here the driving parameter is not productivity but trade-off. However, we can make the following argument for a monotonic relation between productivity and trade-off shape. First we note that biomass productivity is a function of available resources (*Kassen et al., 2000*): the larger the available resources, the higher the possible productivity. This allows us to argue in terms of available resources. For eco-evolutionary systems with scarce resources, species with high replication rates will have low competitive ability because for each individual of the numerous offspring there is little material or energy available to develop costly mechanisms that increase competitive ability. On the other hand, if a species under these resource-limited conditions produces competitively constructed individuals it cannot produce many of them. This argument shows a correspondence between a resource-limited condition and high $\delta$ for trade-off between replication and competitive ability. At the opposite, rich end of the resource scale, evolving species are not confronted with hard choices between replication rate and competitive ability, which is equivalent to low $\delta$. Taken together, the trade-off axis should roughly correspond to the inverted resource axis: high $\delta$ for poor resources (or low productivity) and low $\delta$ for rich resources (or high productivity); a detailed analytical derivation will be presented elsewhere. The fact that ITEEM produces this frequently observed humpback curve proposes trade-off as underlying mechanism of this productivity-diversity relation.

## Frequency-dependent selection

Observation of eco-evolutionary trajectories as in *Figure 3* suggested the hypothesis that speciation and extinction events in ITEEM simulations do not occur at a constant rate and independently of each other, but that one speciation or extinction makes a following speciation or extinction more likely. Such a frequency-dependence occurs if emergence or extinction of one species creates the niche for emergence and invasion of another species, or causes its decline or extinction (*Herron and Doebeli, 2013*). Without frequency-dependence such evolutionary events should be uncorrelated.

To test for frequency-dependent selection we checked whether the probability distribution of inter-event times (time intervals between consecutive speciation or extinction events) is compatible with a constant rate Poisson process, i.e. a purely random process, or whether such events are correlated (Appendix 1, Frequency-dependent selection). We find that for long inter-event times the decay of the distribution in ITEEM simulations is indistinguishable from that of a Poisson process. However, for shorter times there are significant deviations from a Poisson process for speciation and extinction events: at inter-event times of around $10^4$ the probability *decreases* for a Poisson process but significantly *increases* in ITEEM simulations. Thus, the model shows frequency-dependent selection with the emergence of new species increasing the probability for generation of further species, and the loss of a species making further losses more likely. This behavior of ITEEM is similar to microbial systems where new species open new niches for further species, or the loss of species causes the loss of dependent species (*Herron and Doebeli, 2013*; *Maddamsetti et al., 2015*).

The above analysis illustrates a further application of ITEEM simulations. Eco-evolutionary trajectories from ITEEM simulations can be used to develop analytical methods for the inference of competition based on observed diversification patterns. Such methods could be instrumental for understanding the reciprocal effects of competition and diversification.

## Effect of mutation on diversity

Mutations are controlled in ITEEM by two parameters: mutation probability $\mu$, and width $m$ of trait variation. In simulations, diversity grew faster and to a higher level with increasing mutation probability ($\mu = 10^{-4}, 5 \times 10^{-4}, 10^{-3}, 5 \times 10^{-3}$), but without changing the overall structure of the phase diagram (Appendix 1, Mutation probability). One interesting tendency is that for higher $\mu$, the lifespan becomes more important at the interface of regions III and IV (high trade-offs), leading to an expansion of region III at the expense of the hostile region IV: long lifespans in combination with high mutation probability establish low but viable diversity at large $\delta$. The humpback curve of diversity over $\delta$ is observed for all mutation probabilities. Thus, the diversity in ITEEM is not a simple result of a mutation-selection balance but trade-off plays an important role in shaping diversity in trait space.

The width of trait variation, $m$, influences both the speed of evolutionary dynamics and the maximum variation inside species, i.e. clusters of strains. The smaller $m$ the slower the dynamics and the smaller the clusters. However extreme values of $m$ can completely suppress the diverging evolution: Very small variations are wiped out by rapid ecological dynamics, and very large variations disrupt selection forces by imposing big fluctuations.

## Comparison of ITEEM with neutral model

The neutral model introduced in the Model section has no meaningful interaction traits, and consequently no meaningful competitive ability or trade-off with fecundity. Instead, it evolves solely by random drift in trait space. Similarly to ITEEM, the neutral model generates clumpy structures of traits (Appendix 1, Neutral model), though here the clusters are much closer and thus the functional diversity is much lower. This can be demonstrated quantitatively by the size of the minimum spanning tree of populations in trait space that are much smaller for the neutral model than for ITEEM at moderate trade-off (Appendix 1, Neutral model). The clumpy structures generated with the neutral model do not follow a stable trajectory of divergent evolution, and, hence, niche differentiation cannot be established. In a neutral model, without frequency-dependent selection and trade-off, stable structures and cycles cannot form in the community network, and consequently, diversity cannot grow effectively (Appendix 1, Neutral model). The comparison with the neutral model points to frequency-dependent selection as a promoter of diversity in ITEEM. For high trade-offs (region III in *Figure 4b*), diversity and number of strong cycles in ITEEM are comparable to the neutral model (*Figure 4a*).

## Discussion

### Phenotype traits and interaction traits

In established eco-evolutionary models, organisms are described in terms of one or a few *phenotype traits*. In contrast, the phenotype space of real systems is often very high-dimensional; competitive species in their evolutionary arms race are not confined to few predefined phenotypes but rather explore new dimensions in that space (*Maharjan et al., 2006*; *Maharjan et al., 2012*; *Zaman et al., 2014*; *Doebeli and Ispolatov, 2017*). Coevolution systematically pushes species toward complex traits that facilitate diversification and coexistence (*Zaman et al., 2014*; *Svardal et al., 2014*), and evolutionary innovation frequently generates phenotypic dimensions that are completely novel in the system (*Doebeli and Ispolatov, 2017*). Complexity and multi-dimensionality of *phenotype space* have recently been the subject of several experimental and theoretical studies with different approaches that demonstrate that evolutionary dynamics and diversification in high-dimensional *phenotype trait space* can produce more complex patterns in comparison to evolution in low-dimensional space (*Doebeli and Ispolatov, 2010*; *Gilman et al., 2012*; *Svardal et al., 2014*; *Kraft et al., 2015*; *Doebeli and Ispolatov, 2017*). For example, it has been shown that the conditions needed for frequency-dependent selection to generate diversity are satisfied more easily in high-dimensional *phenotype spaces* (*Doebeli and Ispolatov, 2010*). Moreover, the level at which diversity saturates in a system depends on its dimensionality, with higher dimensions allowing for more diversity (*Doebeli and Ispolatov, 2017*), and the probability of intransitive cycles in species competition networks grows rapidly with the number of *phenotype traits*. The conventional way to tackle this problem is to use models with a larger number of *phenotype traits*. However, this is not really a solution of the problem because this still confines evolution to the chosen fixed number of traits, and it also makes these models more complex and thus computationally less tractable. As will be discussed below, interaction-based models such as ITEEM offer a natural solution to this problem by mapping the system to an *interaction trait space* that can dynamically expand by the emergence of novel interaction traits as eco-evolutionary dynamics unfolds.

### Eco-evolutionary dynamics in interaction trait space

Interaction-based eco-evolutionary models rely on the assumption that phenotypic evolution can be coarse-grained to the interaction level (*Figure 1*). This means that regardless of the details of phenotypic variations, we just study the resultant changes in the interaction network. In an eco-evolutionary system dominated by competition this is justified because phenotypic variations are relevant only

when they change the interaction of organisms, directly or indirectly; otherwise they do not impact ecological dynamics. The interaction level is still sufficiently detailed to model macro-evolutionary dynamics that are dominated by ecological interactions.

A transition from phenotype space to interaction space requires a mapping from the former to the latter, based on the rules that characterize the interaction of individuals with different phenotypic traits. As a concrete example, we might consider the competition kernel of adaptive dynamics models (*Doebeli, 2011*) that determines the competitive pressure of two individuals with specific traits. That formalism describes well how, after mapping phenotypic traits to the interaction space, ecological outcome eventually is determined by interactions between species. In Appendix 1, Phenotype-interaction map, some properties of this mapping are discussed.

## Interaction-based models

In the first interaction-based model by *Ginzburg et al. (1988)*, emergence of a new mutant was counted as speciation, and it was shown that simulating speciation events as ecologically continuous mutations in the strength of competitive interactions resulted in stable communities. However the Ginzburg model produced stable coexistence of only a few similar interaction traits, without branching and diversification to distinct species. As outlined in the introduction, subsequent interaction-based models tried to solve this problem by supplementing the Ginzburg model with some *ad hoc* features. For example, *Tokita and Yasutomi (2003)* mixed mutualistic and competitive interactions, and showed that only local mutations, i.e. changes in one pair-wise interaction rate, can produce stable diversity. Recently, *Shtilerman et al. (2015)* enforced diversification in purely competitive communities by imposing a large parent-offspring niche separation. To our knowledge, ITEEM is the first interaction-based model in which, despite its minimalism and without *ad hoc* features, diversity gradually emerges under frequency-dependent selection by considering physical constraints of eco-evolutionary dynamics.

In all previous interaction-based models, eco-evolutionary dynamics has been divided into iterations over two successive steps: each first step of continuous population dynamics, implemented by integration of differential equations, was followed by a stochastic evolutionary process, namely speciation events and mutations, as a second step. However, in nature these two steps are not separated but intertwined in a single non-equilibrium process. Hence, the artificial separation necessitated the introduction of model components and parameters that do not correspond to biological phenomena and observables. In contrast, individual-based models like ITEEM operate with organisms as units, and efficiently simulate eco-evolutionary dynamics in a more natural and consistent way, with parameters that correspond to biological observables.

## Trade-off anchors eco-evolutionary dynamics in physical reality

Life-history trade-offs, like the trade-off between replication and competitive ability, now experimentally established as essential to living systems (*Stearns, 1989*; *Agrawal et al., 2010*; *Masel, 2014*), are inescapable constraints imposed by physical limitations in natural systems. Our results with ITEEM show that trade-offs fundamentally impact eco-evolutionary dynamics, in agreement with other eco-evolutionary models with trade-off (*Huisman et al., 2001*; *Bonsall et al., 2004*; *de Mazancourt and Dieckmann, 2004*; *Beardmore et al., 2011*). Remarkably, we observe with ITEEM sustained high diversity in a well-mixed homogeneous system. This is possible because moderate life-history trade-offs force evolving species to adopt different strategies or, in other words, lead to the emergence of well-separated functional niches in interaction space (*Gudelj et al., 2007*; *Beardmore et al., 2011*).

Given the accumulating experimental and theoretical evidence, the importance of trade-off for diversity is becoming more and more clear. ITEEM provides an intuitive and generic conceptual framework with a minimum of specific assumptions or requirements. This makes the results transferable to different systems, for example biological, economical and social systems, wherever competition is the driving force of evolving communities. Put simply, ITEEM shows generally that in a bare-bone eco-evolutionary model withal standard population dynamics (birth-death-competition) and a basic evolutionary process (mutation), diverse set of strategies will emerge and coexist if physical constraints force species to manage their resource allocation.

## Power and limitations of ITEEM

Despite its minimalism, ITEEM reproduces in a single framework several phenomena of eco-evolutionary dynamics that previously were addressed with a range of distinct models or not at all, namely sympatric and concurrent speciation with emergence of new niches in the community, mass extinctions and recovery, large and sustained functional diversity with hierarchical organization, spontaneous emergence of intransitive interactions and cycles, and a unimodal diversity distribution as function of trade-off between replication and competition. The model allows detailed analysis of eco-evolutionary mechanisms and could guide experimental tests.

The current model has important limitations. For instance, the trade-off formulation was chosen to reflect reasonable properties in a minimalist way. This should be revised or refined as more experimental data become available. Secondly, individual lifespans in this study came from a random distribution with an identical fixed mean. Hence we have no adaptation and evolutionary-based diversity in lifespan. This limits the applicability of the current model to communities of species that have similar lifespans, and that invest their main adaptation effort into growth or reproduction and competitive ability. Furthermore, our model assumes an undefined pool of *steadily replenished shared* resources in a well-mixed system. This was motivated by the goal of a minimalist model for competitive communities that could reveal mechanisms behind diversification and niche differentiation, without resource partitioning or geographic isolation. However, in nature, there will in general be few or several limiting resources and abiotic factors that have their own dynamics. For this scenario, which is better explained by a resource-competition model than by the GLV equation, it is possible to consider resources as additional rows and columns in the interaction matrix $\mathbf{I}$ and in this way to include abiotic interactions as well as biotic ones.

In an interaction-based model like ITEEM the interaction terms of the mutants change gradually and independently (*Equation 1*). This assumption of random exploration of interaction space can be violated, for example, in simplified models with few fixed phenotypic traits. Further studies are necessary to investigate the general properties and restrictions of the map between phenotype and interaction space. In Appendix 1, Phenotype-interaction map we briefly introduced and discussed some properties of this map.

## Acknowledgements

We thank S Moghimi-Araghi for helpful suggestions on the trade-off function. We also thank the reviewers for their comments and insights, which helped us to improve the paper.

## Additional information

### Funding
No external funding was received for this work.

### Author contributions
Farnoush Farahpour, Conceptualization, Software, Formal analysis, Validation, Investigation, Visualization, Methodology, Writing—original draft, Project administration, Writing—review and editing; Mohammadkarim Saeedghalati, Conceptualization, Software, Formal analysis, Validation, Investigation, Visualization, Methodology, Writing—review and editing; Verena S Brauer, Conceptualization, Methodology, Writing—review and editing; Daniel Hoffmann, Conceptualization, Resources, Supervision, Funding acquisition, Validation, Methodology, Writing—original draft, Project administration, Writing—review and editing

### Author ORCIDs
Farnoush Farahpour (iD) http://orcid.org/0000-0002-4510-8483
Mohammadkarim Saeedghalati (iD) http://orcid.org/0000-0003-3387-6263
Daniel Hoffmann (iD) http://orcid.org/0000-0003-2973-7869

### Decision letter and Author response
Decision letter https://doi.org/10.7554/eLife.36273.029

Author response https://doi.org/10.7554/eLife.36273.030

## Additional files

### Data availability

The source code of the model is freely available at https://github.com/BioinformaticsBiophysicsUDE/ITEEM; copy archived at https://github.com/elifesciences-publications/ITEEM).

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

## Appendix 1

DOI: https://doi.org/10.7554/eLife.36273.008

### Species and strains

There is no universally accepted definition of species (*Zachos, 2016*), especially for asexual populations (*Zachos, 2016*; *Birky Jr and Barraclough, 2009*; *Richards, 2013*). In the present work, we follow *Rosselló-Mora and Amann (2001)* and use the concept of phylo-phenetic species applicable to asexual populations. A phylo-phenetic species is defined as a monophyletic cluster of strains that show a high degree of overall similarity with respect to many independent characteristics (*Rosselló-Mora and Amann, 2001*). We used genealogical trees to define species as group of strains that share a most recent common ancestor and are separated by long-lasting gaps in the tree. Each of these clusters that is branching off from a point of divergence in the tree was counted as an individual species if it has existed for more than a certain number of generations ($7 \times 10^4$ in the manuscript), considering all of its sub-branches. Changing the threshold in a range from $2 \times 10^4$ to $10^5$ had no profound impact on the results. Branches that lasted shorter than this threshold were counted as strains of their parents (*Appendix 1—figure 1*).

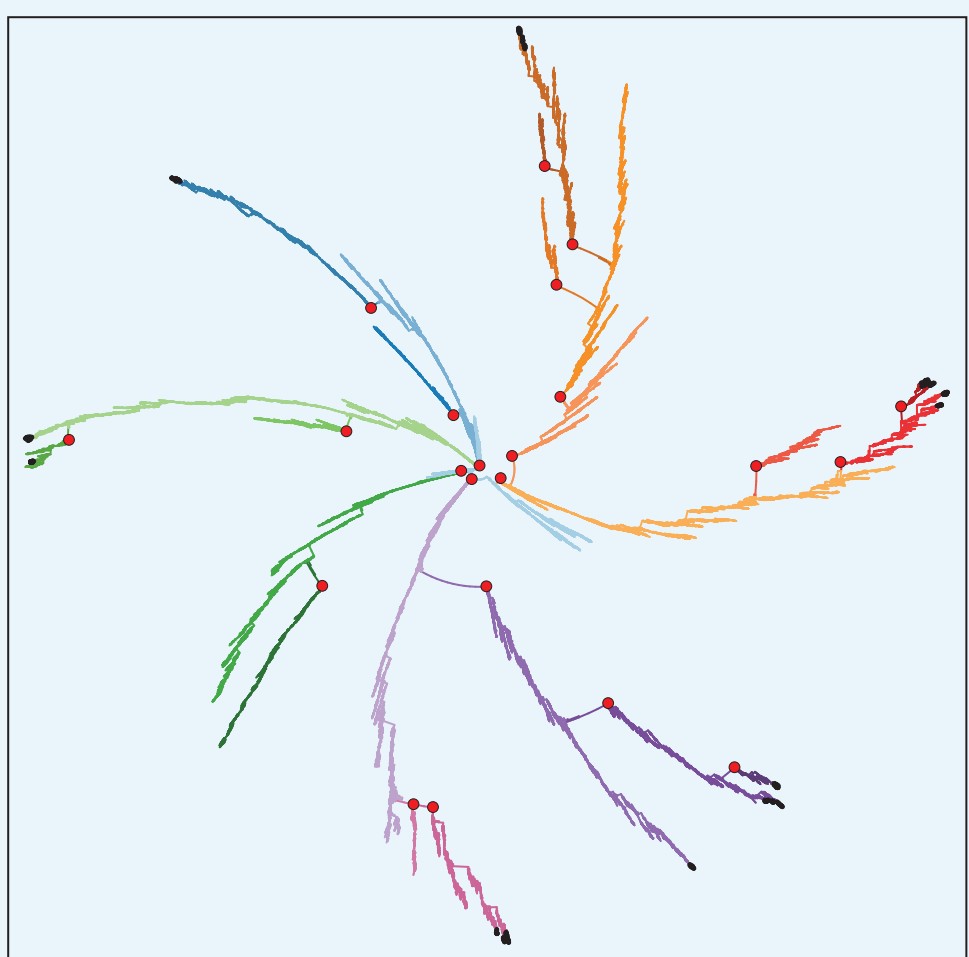

**Appendix 1—figure 1.** Genealogical tree corresponding to the simulation reported in *Figure 3* of the main text. This tree contains around 80000 different strains clustered into 24 species with the threshold of $7 \times 10^4$ generations. Colors are the same as in *Figure 3a, b, c and f* of the main text. Black dots represent the strains that are present in the last snapshot

of *Figure 3a and b* (Generation $10^6$). 10 out of 24 species are still extant in the last snapshot.
Red circles show the branching points.
DOI: https://doi.org/10.7554/eLife.36273.009

Distribution of strains in trait space and their diversification (see *Figure 3b and c* of the main text) shows that these monophyletic clusters are also the well-separated clusters in functional space which together allow us to consider them as phylo-phenetic species.

See *Birky Jr and Barraclough, (2009)*; *Ereshefsky (2010)*; *Richards (2013)*; *Wilkins (2018)* for a discussion on pros and cons of this definition.

## Trade-off

### Trade-off function

Linear, concave and convex trade-offs among different life-history traits have been reported in many studies (*Jessup and Bohannan, 2008*; *Maharjan et al., 2013*; *Saeki et al., 2014*; *Ferenci, 2016*). It has been shown that different forms of trade-off reproduce different diversity and coexistence patterns (*Levins, 1968*; *Maharjan et al., 2013*; *Kasada et al., 2014*; *Ehrlich et al., 2017*). The shape of trade-off is determined by various factors like quantitative relationship between resource allocations in life-histories (*Saeki et al., 2014*), physiological mechanisms (*Bourg et al., 2017*), and environment (*Jessup and Bohannan, 2008*).

In this study, to implement a trade-off between reproduction $r_\alpha$ and competitive ability $C(\mathbf{T}_\alpha)$ with a variable form, we used a function with one shape parameter $s$:

$$r(\mathbf{T}_\alpha) = \left(1 - C(\mathbf{T}_\alpha)^{1/s}\right)^s \tag{3}$$

where $\mathbf{T}_\alpha$ is the trait vector of strain $\alpha$. This trade-off function maps competitive ability of species to reproduction probability in the range $[0, 1]$. By changing the exponent $s$ between simulations, we can study the effect of trade-off shape on eco-evolutionary dynamics. For a systematic scan of trade-off shapes (*Figure 2* in the main text), we formulated the shape parameter $s$ as

$$s = -log_2(1 - \delta), \tag{4}$$

with *trade-off parameter* $\delta$ covering $[0, 1]$ in equidistant steps. Of course, other functional forms of the trade-off are conceivable.

### Trade-off and explored interaction trait space

Trade-offs between life-history traits are constraints imposed by fundamental resource-allocation principles. They confine evolutionary adaptations and innovations to a permissible subspace of all trait combinations. However the outcome of evolution – determined by the underlying mechanisms of the system and selection forces – is a subset of this permissible subspace, occupied by the selected coexisting organisms, in which all organisms should have more or less the same fitness to be able to coexist. Thus, in each community and at each time point, just a small part of this permissible space is usually occupied (*Bourg et al., 2017*).

In ITEEM, the competitive ability $C$ of a strain (*Equation 2* of the main text) quantifies how successfully individuals of this strain compete against individuals of all co-existing strains in direct encounters. Thus, $C$ is a relative and density-dependent component of the fitness. In the course of evolutionary dynamics, $C$ never explored extreme regimes, which means that we never observed $C \approx 0$ (organisms that fail nearly in all encounters) or $C \approx 1$ (organisms that defeat nearly all the rivals). Instead, we saw a distribution around middle values, $C \approx 0.5$. In the low trade-off regime, emergence of strategies with relatively high competitiveness, without a considerable cost in reproduction, drives the system to low diversity by outfighting the competitors. In this case, as $C$ is a relative, interaction-based measure (*Equation 2* of the main text) extinction of species with low competitive ability pushes the distribution of $C$ again toward 0.5. In the high trade-off regime, even small gains in $C$ come with a severe penalty in $r$,

that is, increasing competitive ability to high values is very unlikely; thus, strategies with a relatively high $r$ can prevail by a negligible decrement in their relative competitive ability, which again limits diversity of strategies and brings them back to $C \approx 0.5$. For moderate trade-off values between the above limiting cases, $C$ also stabilizes around 0.5, as described in the main text (Results, *Impact of trade-off and lifespan on diversity*).

### Strength of trade-off function

At first sight, the strength of a trade-off function – how strongly changes in one trait influence the other trait – seems to be just a synonym for the slope (=first derivative) of the trade-off function. For instance, consider two traits $x$ and $y$ that both contribute to the fitness and are related by trade-off function $y = f(x)$ with first derivative $f'(x)$. A small change $\Delta x$ in trait $x$ will cause a change $\Delta y \approx f'(x)\Delta x$ in trait $y$, so that, obviously, for given $\Delta x$ the slope $f'(x)$ determines the change $\Delta y$. However, the effect of such a change will very much depend on the community context: the same change $\Delta x$ or $\Delta y$ may be relatively large or relatively small, depending on the actual values of $x$ and $y$. For example, a $\Delta y = 0.1$ will change $y = 0.1$ by 100%, but a larger $y = 0.8$ by a mere 12.5%, and accordingly the change $\Delta y$ will have different effects on the fitness of the affected strain. Therefore, we define as trade-off strength $\theta(x, y)$ the ratio $\frac{\Delta y}{y} / \frac{\Delta x}{x}$, or, in the limit of $\Delta x, \Delta y \to 0$ as

$$\theta(x, y) = \frac{dy}{dx} \cdot \frac{x}{y}. \tag{5}$$

To demonstrate that the trade-off strength $\theta$ is a more meaningful quantity than the slope to characterize the effect of the trade-off, we discuss in the following a few characteristic cases.

Assume the same strain with a trait value $x$ in two different systems, 1 and 2, with different trade-off functions $f_1, f_2$, respectively. In the first system, $x$ may be mapped to a large value of trait $y_1 = f_1(x)$, while in the second it may be mapped to a small value of trait $y_2 = f_2(x)$. Emergence of a mutant with a small change $\Delta x$ in trait $x$ causes different variations in the two systems, namely $\Delta y_1 \approx f'_1(x)\Delta x$ among large traits $\bar{y}_1$, and $\Delta y_2 \approx f'_2(x)\Delta x$ among small traits $\bar{y}_2$. Thus the same $\Delta x$ impacts the two systems differently, even if the slopes $f'_i(x)$ would be the same. The trade-off strength $\frac{\Delta y}{y} / \frac{\Delta x}{x}$ captures this difference as it is smaller for system 1 and larger for system 2.

$\theta$ is also expressive if the two systems have different values of the first trait $x_1 < x_2$ that are mapped to the same second trait $y$ and have the same slope of trade-off function in the respective range ($f'_1(x_1) = f'_2(x_2)$). The same changes in the first trait $\Delta x$ have different impacts on relative fitness of the two systems, which is again captured by trade-off strength $\frac{\Delta y}{y} / \frac{\Delta x}{x}$. Only in the special case $x_1 = x_2$ and $y_1 = y_2$, the slope $\frac{\Delta y}{\Delta x}$ is sufficient to compare the effects of trade-offs on fitness and dynamics.

For $x = C$ and $y = r$ the derivative $\frac{dy}{dx}$ is the derivative of the trade-off function in **Figure 2** of the main text, explicitly formulated in **Equation 3**.

In ITEEM, as explained in the previous section, the competitive ability $C$ is typically in the middle range, i.e. organisms with low or high competitive ability are rare. In this middle range, trade-off strength $\theta(C, r) = \frac{dr}{dC} \frac{C}{r}$ increases with increasing trade-off parameter $\delta$, as shown in **Appendix 1—figure 2** below.

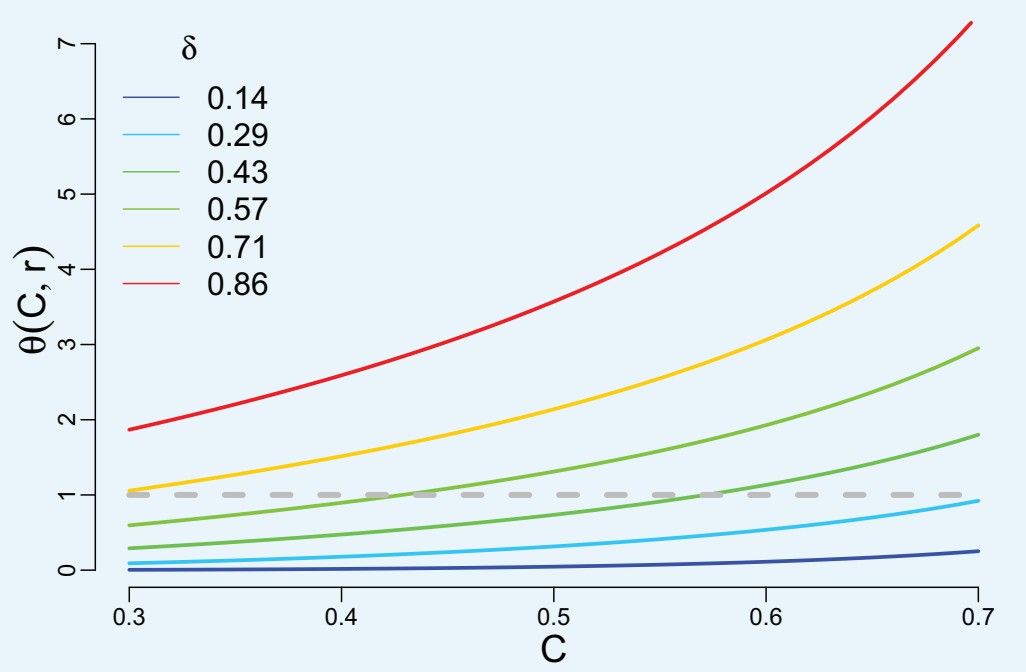

**Appendix 1—figure 2.** Trade-off strength $\theta(C, r)$ as a function of competitive ability for different values of trade-off parameter $\delta$. For $\delta > 0.5$, $\theta(C, r)$ increases rapidly and deviates from one but for low trade-offs $\theta(C, r) \ll 1$. Color code corresponds to the one used in *Figure 2* of the main text.

DOI: https://doi.org/10.7554/eLife.36273.010

## Generalized Lotka–Volterra (GLV) equation

As an individual-based model, ITEEM simulates systems consisting of distinct, interacting organisms, and thus can model non-equilibrium dynamics, demographic fluctuations, effects of diverse lifespans, and other features of real systems, as discussed in the main text. If these features were not of concern we could replace the ecological dynamics of ITEEM by the corresponding population-level model. In the following we show that such an abstraction of ecological interactions of ITEEM leads to the competitive generalized Lotka-Volterra (GLV) equation.

We start from the main equation of population dynamics for our model:

$$\dot{x}_\alpha = r_\alpha x_\alpha \left(1 - \sum_\beta x_\beta\right) + \sum_\beta r_\alpha x_\alpha I_{\alpha\beta} x_\beta - \sum_\beta r_\beta x_\beta I_{\beta\alpha} x_\alpha - d x_\alpha \tag{6}$$

In which $x_\alpha = \frac{n_\alpha}{N_s}$ is the relative abundance or probability of finding strain $\alpha$ in the system ($N_s$ is the number of sites in the system). The first term on the right side of *Equation 6* is the growth of the population of strain $\alpha$ when it produces progeny that is able to find an empty space in the system. The second term shows the growth of population $\alpha$ when after reproduction its offspring is able to invade a site occupied by another individual, the third term is the decrease of population $\alpha$ due to invasion by offspring of other strains and the last term is the decrease of population $\alpha$ due to the intrinsic death rate because of the attributed life span ($d = 1/\lambda$). We can rewrite the equation as follows:

$$\dot{x}_\alpha = x_\alpha\left(r_\alpha - d - \sum_\beta (r_\alpha - r_\alpha I_{\alpha\beta} + r_\beta I_{\beta\alpha})x_\beta\right)$$

$$= x_\alpha\left(r_\alpha - d - \sum_\beta (r_\alpha(1 - I_{\alpha\beta}) + r_\beta I_{\beta\alpha})x_\beta\right)$$

$$= x_\alpha\left(r_\alpha - d - \sum_\beta (r_\alpha + r_\beta)I_{\beta\alpha}x_\beta\right)$$

$$= x_\alpha\left(r_\alpha - d + \sum_\beta A_{\alpha\beta}x_\beta\right)$$

In which we used $1 - I_{\alpha\beta} = I_{\beta\alpha}$ (see *Equation 1* in the main text). The last equation above, which we can rewrite compactly as

$$\dot{\mathbf{x}} = \mathbf{x}\cdot(\mathbf{g} + \mathbf{A}\mathbf{x}), \qquad (7)$$

is the GLV equation. $\mathbf{g} = \mathbf{r} - d$ is the effective population growth rate and $\mathbf{A}$ is the community matrix; its elements $A_{\alpha\beta} = -(r_\alpha + r_\beta)I_{\beta\alpha}$ are always negative in our system which shows that ITEEM strains and species are competing.

The close relationship of our individual-based ecological dynamics with the GLV equation shows that organisms are competing in the sense that they expand their populations at the expense of their competitors populations to secure resources and increase fitness. This similarity of GLV with ITEEM ecological dynamics also explains why ITEEM individual-based dynamics corresponds to a well-mixed system. A 'site' in the model is not a patch of space or a piece of a spatially structured resource – neighborhood has no meaning, as in the GLV model. Instead, a 'site' stands for a discrete portion of the steadily replenished resource pool that is equally accessible to all extant individuals, and sufficient for their respective metabolisms. Being well-mixed means that any individual meets any other individual and site with the same probability. A difference between the individual-based ITEEM and the population-level version in *Equation 7* is that the former models encounters at the level of individuals whereas the latter maps encounters to interactions between populations.

As outlined above, ITEEM simulations can be interpreted in the framework of the competitive GLV equation. In evolving systems governed by this equation, fitness is determined by reproduction rate, carrying capacity and competitive abilities (*Gill, 1974*; *Masel, 2014*). In the present model, the carrying capacity is the same for all strains and species, and the fitness at the low density limit (corresponding to the initial phase of the simulations) is determined by replication $r$. At the high density limit typically simulated in the present work, the product of $r$ and $C$ determines the fitness (*Masel, 2014*).

## Classical multi-dimensional scaling (CMDS)

Multi-dimensional scaling (MDS) algorithms take sets of points in $N$-dimensional space and represent them in a lower-dimensional space (typically 2-dimensional) so that the original distances in $N$-dimensional space are preserved as well as possible. The lower-dimensional representation can then be easily visualized.

Classical MDS (CMDS) is a member of the family of MDS methods (*Cox and Cox, 2000*; *Borg and Groenen, 2005*; *Wang, 2012*). The algorithm takes as input an $N \times N$ distance matrix, where the distances could for example be dissimilarities between pairs of $N$ objects, and outputs a coordinate matrix that determines positions of the points in a lower-dimensional (often 2-dimensional) space with the condition of minimizing the loss function that measures discrepancy between the algorithm's output and the real distances. The quality of the lower-dimensional representation can be assessed from the eigenvalues of a factor analysis that shows the fraction of variation in the data explained by each dimension. CMDS is mathematically closely related to principal component analysis (PCA) (*Wang, 2012*).

In our analysis, objects are interaction traits of strains and our aim is to visualize the distribution of them in trait space. To this end we first calculate the Euclidean distances between all trait vectors

$$D_{\alpha\beta} = \mathrm{Dist}\left(\mathbf{T}_\alpha, \mathbf{T}_\beta\right) = \sqrt{\sum_{\gamma=1}^{N_{st}} \left(I_{\alpha\gamma} - I_{\beta\gamma}\right)^2}, \tag{8}$$

and then, by applying the *cmdscale* function of the R software (version 3.3.0) to that distance matrix, we project our trait space into 2-dimensional plots. Thus, each point in the 2-dimensional CMDS plot represents a trait vector, i.e. a strain. The larger the distance between two points, the more different the traits of corresponding strains. While CMDS is a handy tool for the visualization of evolutionary processes in trait space (*Figure 3b and f* of the main text), all quantitative analyses were performed in the original high-dimensional space.

In the very early stages of evolution (*Appendix 1—figure 3*-top) strains are very similar so that two dimensions are not sufficient to represent their dissimilarities accurately (relatively high eigenvalues beyond the 2nd eigenvalue in top right of *Appendix 1—figure 3*). But when evolutionary speciation's and branching's occur, the 2-dimensional space is more appropriate, as the eigenvalues from the factor analysis show (*Appendix 1—figure 3*-bottom and *Appendix 1—figure 4*).

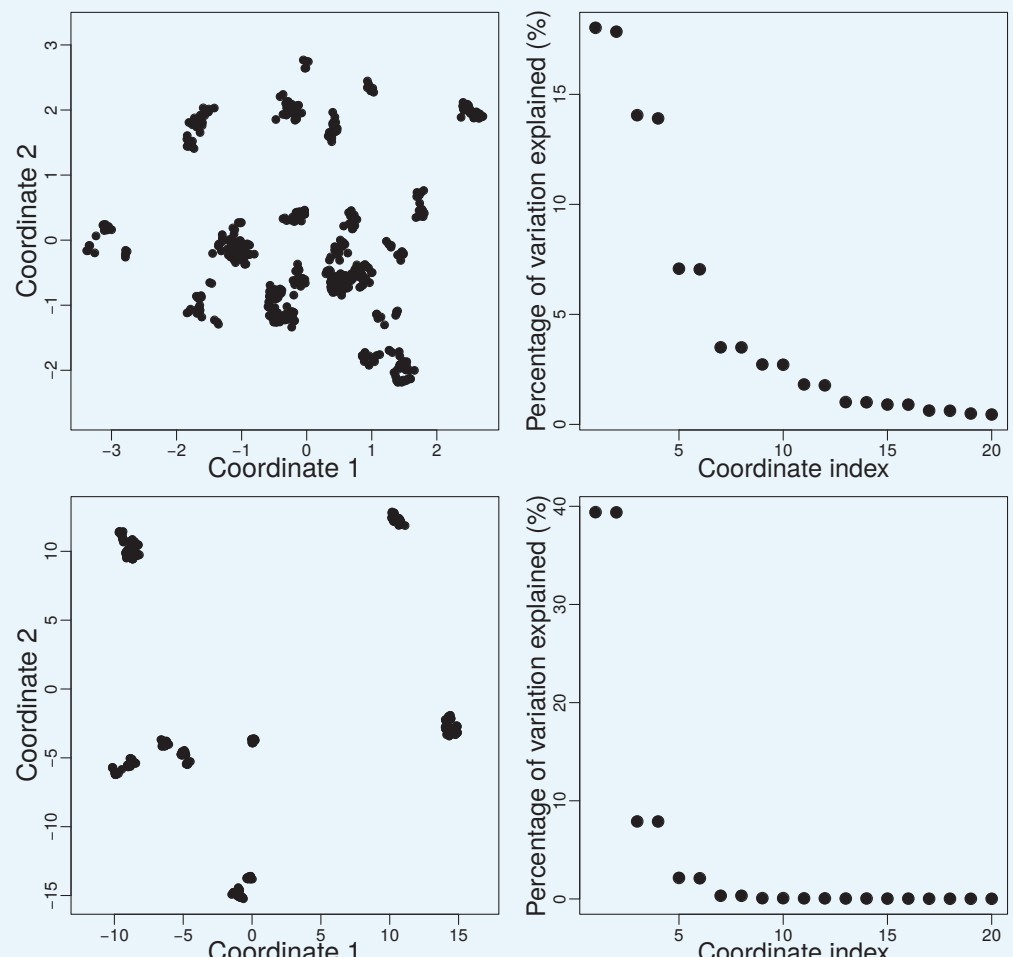

**Appendix 1—figure 3.** Top-left: Trait space at generation $6 \times 10^4$ projected into two dimensions using CMDS. Top-right: Percentage of variation explained by the first 20 eigenvectors from factor analysis. Here the first two eigenvectors explain around $2 \times 18 = 36\%$ of variation. Bottom: Same system as the top but for generation $1 \times 10^6$. Here, the first two eigenvectors explain around 78% of variation. Simulation was done with $\delta = 0.5$, $\lambda = 300$, $\mu = 0.001$ and $m = 0.02$.

DOI: https://doi.org/10.7554/eLife.36273.011

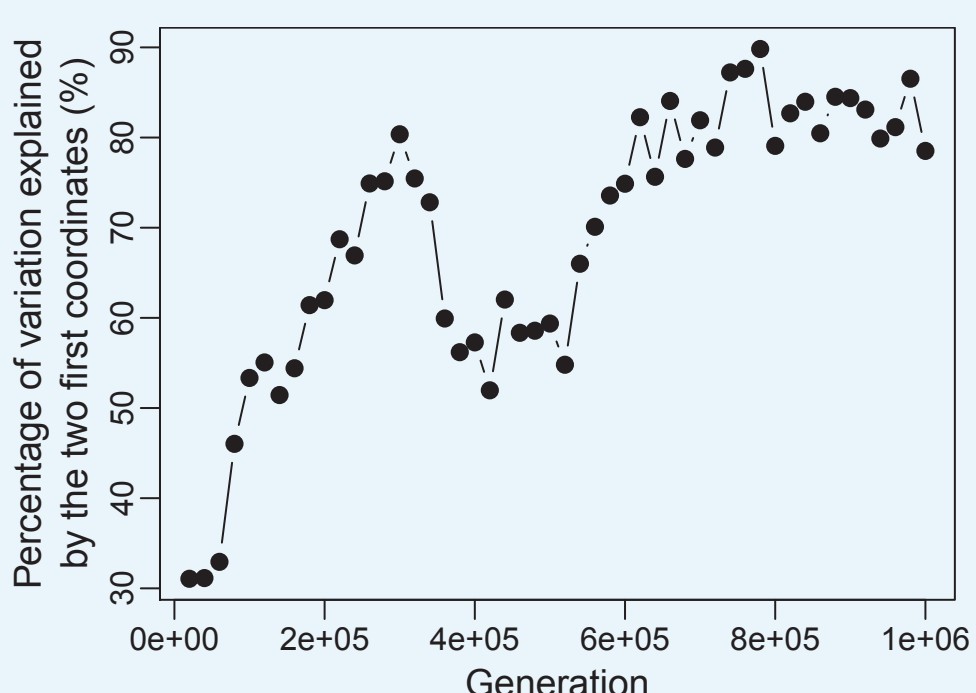

**Appendix 1—figure 4.** Percentage of variation explained by the first two coordinates versus time for one simulation. Simulation was done with $\delta = 0.5$, $\lambda = 300$, $\mu = 0.001$ and $m = 0.02$.
DOI: https://doi.org/10.7554/eLife.36273.012

## SMST and distribution of species and strains in trait space

The minimum spanning tree (MST) has been used as a tool to characterize the distribution of species and strains in interaction trait space. The MST of a graph is a tree that connects all nodes of that graph so that the total edge weight is minimized. For $N$ nodes, the MST has $N - 1$ edges. Specifically, if we have $N$ strains in interaction trait space as nodes, we compute as MST a tree of $N - 1$ edges that links all strains with a minimum sum of edge lengths (= distances between strains in interaction trait space [**Equation 8**]). For a more familiar example think of nodes as $N$ cities on a map and the MST as a tree of $N - 1$ edges that links all cities, so that the sum of edge lengths (= distances between pairs of cities) is minimized.

The sum of edges of the MST (SMST) can be used as a quantitative measure that characterizes the distribution of points. The MST of an evenly distributed structure, with $N$ nodes, has $N - 1$ edges with more or less the same length and thus, the SMST scales with $N$. For a hierarchical structure, MST consists of long edges between clusters and short edges that connect the nodes inside each cluster; in this case, the SMST scales with the number of clusters. The SMST increases by divergent evolution. The left panel of *Appendix 1—figure 5* shows an example MST in the trait space of our simulation, represented in 2D for the sake of visualization.

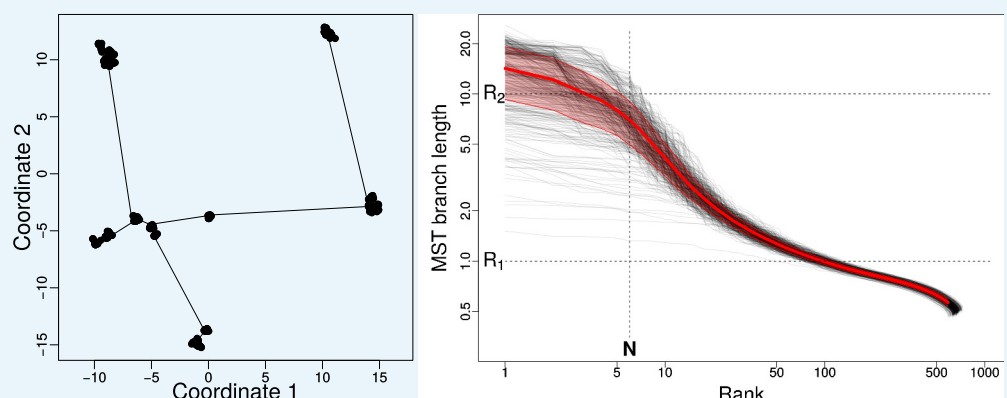

**Appendix 1—figure 5.** Minimum Spanning Tree (MST) in interaction trait space. Left: 2D representation of MST of a typical snapshot of a simulation with $\delta = 0.5$, $\lambda = 300$, $\mu = 0.001$, and $m = 0.02$. The community in this snapshot consists of 667 strains. Here we used R-package vegan 2.4–5 to find the MST. Right: the set of 500 gray curves shows the sorted lengths of the edges of the MST versus their ranks (rank 1 = longest edge) for 500 snapshots from simulations with the aforementioned parameters. The red curve is the average of all curves and the red shaded area is the corresponding standard deviation. $R_1$ and $R_2$ are approximate lengths of short branches connecting strains within clusters ($R_1$), and long branches ($R_2$) connecting species. $N$ approximates the number of long branches or species.
DOI: https://doi.org/10.7554/eLife.36273.013

The right panel of **Appendix 1—figure 5** plots the lengths of the length-sorted edges of the MST versus their ranks for 500 simulation snapshots ($\delta = 0.5$, $\lambda = 300$, $\mu = 0.001$, $m = 0.02$). From this log-log-scaled plot we see that there are two clearly different scales in the size of edges. $R_1$ is a representative value for the size of clusters (distance between strains within a typical species) and $R_2$ is a representative value for the scale of the trait space (typical distance between different species). $N$ in **Appendix 1—figure 5** represents approximately the number of distinct clusters (species) in the system.

## Diversity indexes and parameters of dynamics

A diversity index quantifies a certain aspect of diversity in a single number. Since diversity is itself complex, no single diversity index is sufficient to describe the diversity of a community. For example richness, i.e. number of species, has no information about the distribution of the population among species. Evenness or Shannon entropy takes into account this distribution but does not inform about the diversity of the trait of species, i.e. how diverse is a system with respect to the functionality of its species. Functional diversity indexes focus on this aspect but none of them exhaustively describes properties of trait space (**Mouchet et al., 2010**). For a comprehensive assessment of diversity and community dynamics, information about the density of species over resources, rate of extinction and emergence, and also details of community structure, for example interaction of species and topology of the network, should be considered, too.

## Diversity over time

The next plots show how SMST, an index for functional diversity, changes over time. Note that evolutionary collapses (mass extinctions) occasionally occur (see Appendix 1, Collapses of diversity) with a probability that depends on the trade-off parameter, lifespan and mutation probability. The plots in **Appendix 1—figure 6** follow SMST (see Appendix 1, SMST and distribution of species and strains in trait space) over time for different trade-offs $\delta$ but the same lifespan $\lambda$ (and mutation probability) in each plot.

**Appendix 1—figure 7** follows SMST over time for different lifespans $\lambda$ but the same trade-off $\delta$ in each plot. Comparison of **Appendix 1—figure 7** and **Appendix 1—figure 6** confirms

that diversity and dynamics are strongly associated with trade-off without a noticeable effect of lifespan (except for very short lifespans, upper left panel of *Appendix 1—figure 6* and bottom panel of *Appendix 1—figure 7*).

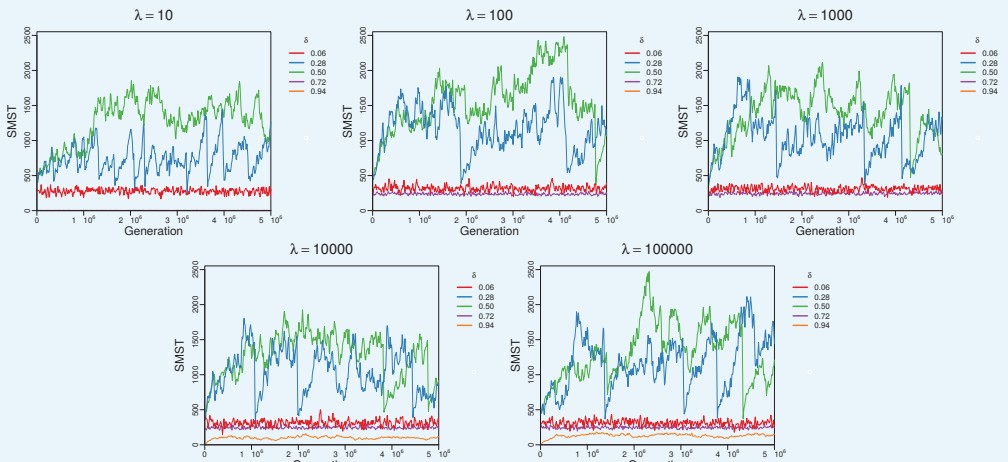

**Appendix 1—figure 6.** Changes of SMST over time for different trade-off parameters $\delta$ but fixed lifespan $\lambda$ in each plot. For very large $\delta$ diverse strategies cannot be adopted. For the smallest $\delta$ diverse strategies can emerge easily, but among them extreme strategies (Darwinian Demons) very quickly dominate leading to low diversity. Sustainable diversity emerges for moderate values of $\delta$. Very short lifespans $\lambda$ prevent increase in diversity, especially for big trade-offs (zero SMST for $\lambda = 10$ and high $\delta$). Simulations were done with $\mu = 0.001$ and $m = 0.02$.

DOI: https://doi.org/10.7554/eLife.36273.014

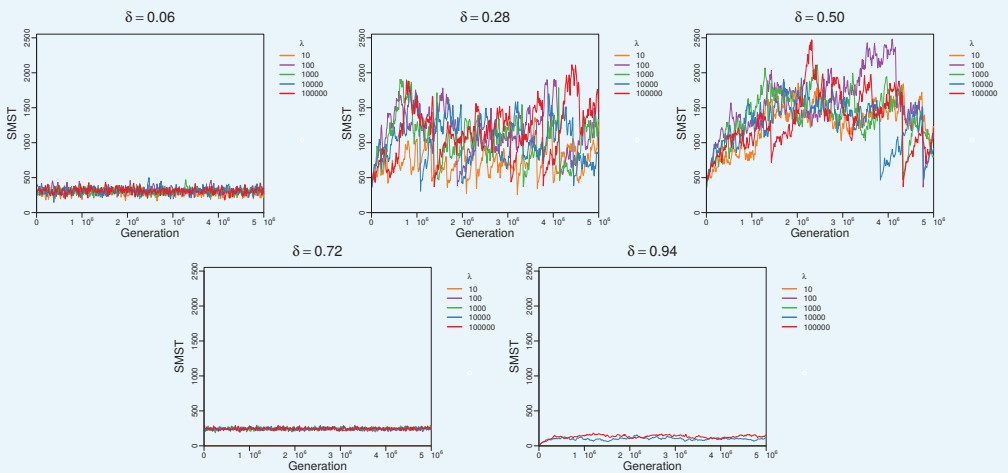

**Appendix 1—figure 7.** Changes of SMST over time for different lifespans $\lambda$ but fixed trade-off $\delta$ in each plot. Apparently lifespan has no large effect for small and moderate $\delta$. For big $\delta$, short lifespans suppress diversity completely (extinction). Simulations with $\mu = 0.001$ and $m = 0.02$.

DOI: https://doi.org/10.7554/eLife.36273.015

## Functional diversity (FD), functional group and functional niche

Univariate diversity indexes that are defined based on abundance of species, like richness and evenness, are routinely used to quantify diversity in biological communities. These indexes are most expressive if species are equal in their effect on their community and ecosystem functioning. However, in the last decades ecologists are increasingly realizing that without information on variety of functions in a community, diversity can not be correctly evaluated

(*Mouchet et al., 2010*), and that trait-based measures that reflect the importance, essentiality, or redundancy of species may be more meaningful than abundance-based measures (*Cadotte et al., 2011*). Inspired by the concept of Hutchinsonian niche, functional diversity (FD) was introduced by Rosenfeld as distribution of species in functional space (*Whittaker et al., 1973*; *Rosenfeld, 2002*). The axes of this space represent the functional features of species (*Mouchet et al., 2010*) which are usually measurable characteristics (traits) that are indicators of organismal performance, and that are associated with species fitness and their ecological function (*Violle et al., 2007*; *Májeková et al., 2016*). In ITEEM, we operate with the interaction trait, which is already an optimal indicator of function of strains and species: the whole vector of interactions that determines the role or function of strain/species in the actual community (*Hooper et al., 2002*; *Sander et al., 2015*). Thus, the distribution of interaction traits in trait space determines variety of functions in the system.

Different measures of FD have been introduced, each quantifying and explaining one facet of trait distribution in trait or functional space, very similar to SMST (Appendix 1, SMST and distribution of species and strains in trait space). In the following we also use three other indexes: functional dispersion, Rao index and functional evenness (*Mouchet et al., 2010*; *Mason et al., 2013*).

Distinct, well-separated clusters in functional (trait) space mean that species are diversified to different functional groups. A functional group is defined in ecology either as a set of species with similar effect on their environment, or as cluster in trait space (*Hooper et al., 2002*). In ITEEM, by using the framework of interaction-based models, these two definitions are interchangeable.

The positions of functional groups in functional space define their functional niches. The notion of functional niche was first introduced by Elton as the place of an animal in its community or its biotic environment (*Elton, 1927*). Then Clarke (*Clarke, 1954*) noted that the functional niche stresses the function of the species in the community, which is different from its physical niche, the latter determining its place in the habitat. A suitable definition of functional niche is the area occupied by a species in the $n$-dimensional functional space (*Clarke, 1954*; *Whittaker et al., 1973*; *Rosenfeld, 2002*). This concept was subsequently differentiated by Odum who considered the habitat as the organism's 'address' and the niche as its 'profession' (*Odum, 1959*). Following this picture and considering that there is no physical niche or habitat in our well-mixed model, we can say that in ITEEM, the position of trait vectors in functional space determines the profession (function) of species. In the eco-evolutionary dynamics of ITEEM, distinct functional groups with different professions/roles/functions emerge in a community of competitive organisms.

## Diversity indexes and parameters of dynamics for different trade-offs and lifespans

For the phase diagram in *Figure 4* of the main text we have synthesized a descriptive dimensionless diversity parameter by averaging over normalized values of several diversity indexes, namely *richness*, *Shannon entropy*, *standard deviation of replication r*, *maximum distance in trait space*, *standard deviation of interaction terms*, *sum of squared lengths of minimum spanning tree of trait space*, *functional diversity indexes* (functional dispersion, Rao index and functional evenness, all three in two versions: with and without abundance), and *strength of cycles*. This phase diagram gives a good overview about different characteristics of communities with different $\delta$ and $\lambda$, but, of course, the averaging procedure leads to a loss of detailed information. Therefore, we report in *Appendix 1—figure 8* some of the most important indexes of community state computed from our simulations for different trade-offs and lifespans. Each index value is averaged over $5 \times 10^6$ generations. The four phases described in *Figure 4* of the main text can be seen in nearly all the parameters. Functional diversity indexes (functional dispersion, functional evenness and Rao's quadratic entropy) are calculated using the dbFD function in R-package FD, version 1.0–12.

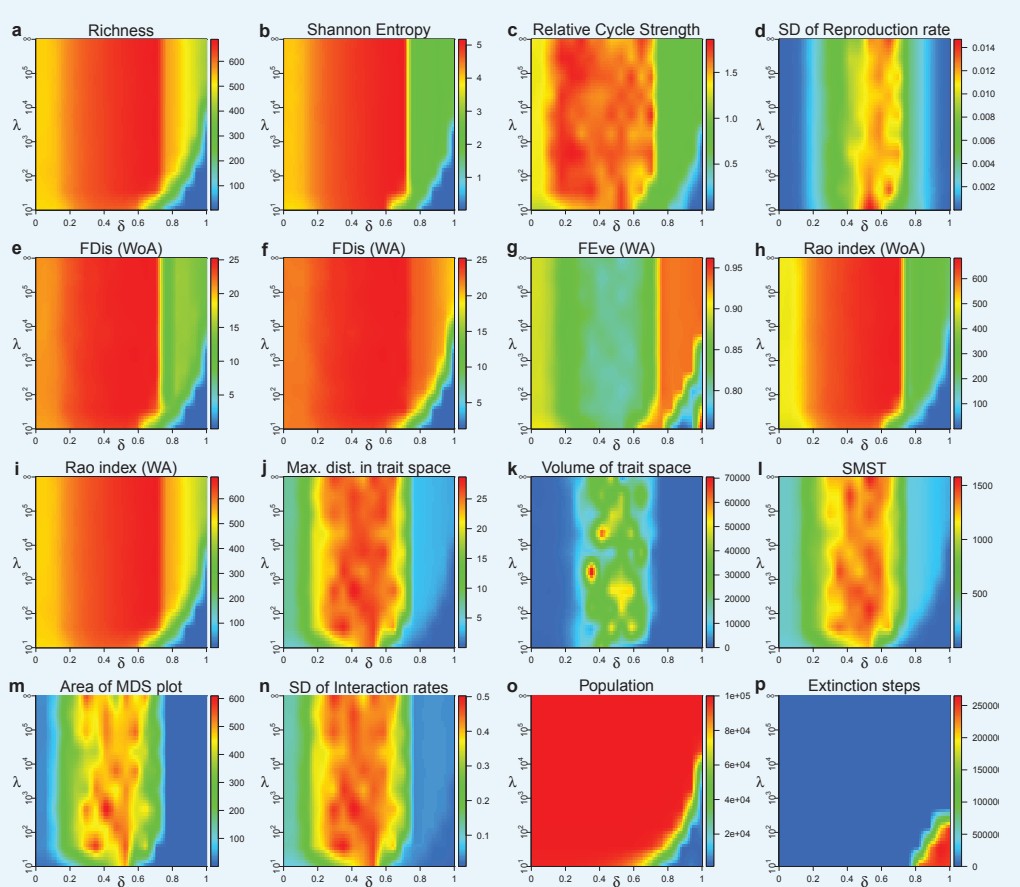

**Appendix 1—figure 8.** Diversity indexes used to compute the phase diagram of *Figure 4* of the main text. (**a**) Richness: number of different strains in community. (**b**) Shannon entropy: here a measure of evenness in strain population. (**c**) Relative strength of cycles of size three compared to random networks (see Appendix 1, Intransitive dominance cycles). (**d**) Standard deviation of replication $r$ as a measure of diversity in reproduction strategy. (**e**) Functional dispersion (FDis, without using the abundance vector): measures the mean distance of individual strains in trait space to their centroid. (**f**) Functional dispersion (FDis, using the abundance vector): measures the mean distance of individual strains (weighted by abundance vector) in trait space to their centroid. (**g**) Functional evenness (FEve, using the abundance vector): quantifies functional evenness and is higher when strains/species are spread homogeneously in trait space. When disruptive selection produces clusters of localized strains in trait space this index decreases. (**h**) Rao's quadratic entropy (without using the abundance vector): measures mean functional distance between two randomly chosen individuals. (**i**) Rao's quadratic entropy (using the abundance vector): measures mean functional distance between two randomly chosen individuals. (**j**) Maximum distance in trait space between strains. (**k**) Volume of trait space: calculated by multiplication of eigenvalues of factor analysis. (**l**) SMST (see Appendix 1, SMST and distribution of species and strains in trait space). (**m**) Area of MDS plot: calculated by multiplication of the two first eigenvalues of factor analysis. (**n**) Standard deviation of interaction terms. (**o**) Community population: number of individuals in community. (**p**) Number of mass extinction events over $5 \times 10^6$ generations. Each plot is the average of corresponding index over three simulations each over $5 \times 10^6$ generations. Simulations with $\mu = 0.001$, $m = 0.02$ and $N_s = 10^5$.

DOI: https://doi.org/10.7554/eLife.36273.016

## Intransitive dominance cycles

Flow of energy/mass between strains in ITEEM community is determined by the dominance matrix $W_{\alpha\beta}$:

$$W_{\alpha\beta} = \begin{Bmatrix} I_{\alpha\beta} - I_{\beta\alpha} & \text{if; } I_{\alpha\beta} > I_{\beta\alpha} \\ 0 & \text{otherwise} \end{Bmatrix} \tag{9}$$

The corresponding network is a directed network with one directed edge between each pair of nodes (strains), pointing from the dominating to the dominated one, with weight between 0 and 1 according to (**Equation 9**). Three or more directed edges in the dominance network can form cycles of strains in which each strain competes successfully against one cycle neighbor but loses against the other neighbor, a configuration corresponding to the rock-paper-scissors game. Even in a completely connected *random* dominance network, a randomly selected triplet of nodes forms a cycle with a probability of $\frac{1}{4}$. We are interested in characterizing evolved networks in ITEEM in comparison to random networks. Hence, we compare number, $N_{cyc}^{Network}$, and average strength, $S_{cyc}^{Network}$, of cycles of the evolving network at each time step with number, $N_{cyc}^{Random}$, and average strength, $S_{cyc}^{Random}$ of cycles of its equivalent shuffled random networks. For this purpose we

1. average over cycles: We select at random 3 nodes of the network and check if they form a cycle of size 3. If yes, the number of 3-cycles, $N_{cyc}^{Network}$, of the network increases by one and the minimum weight among its edges (corresponding to the limiting edge in that cycle for energy/mass flow) is the strength of that cycle. This procedure is repeated many times ($>10^5$), and then we average over the strengths of all cycles to obtain average strength of cycles, $S_{cyc}^{Network}$.
2. build the equivalent random networks: We shuffle the edges of original network to obtain a random network. Then we apply the procedure described in step one on this network to measure the number of cycles and their average strengths. This step is done several times ($>10$) to sample different random networks, and by averaging over them we obtain $N_{cyc}^{Random}$ and $S_{cyc}^{Random}$.
3. normalize the values: Number and average strength of ITEEM network are normalized to the number and average strength of the corresponding random network, respectively: $N_{cyc}^{Network} = \frac{N_{cyc}^{Network}}{N_{cyc}^{Random}}$ and $S_{cyc}^{Network} = \frac{S_{cyc}^{Network}}{S_{cyc}^{Random}}$.

The results of these three steps are plotted in **Figure 3g** and **Figure 4a** of the main text.

## Collapses of diversity

Collapses of diversity occur occasionally in ITEEM simulations. The probability of collapses depends on the trade-off parameter $\delta$, attributed lifespan $\lambda$, and mutation probability $\mu$. In our analysis, a diversity collapse is defined as a sharp decrease in diversity, i.e. a sudden drop of the SMST, larger than half of the temporal average of the SMST in $10^4$ generations (**Appendix 1—figure 9a**). In this way we excluded small or gradual decreases in a diversity measure.

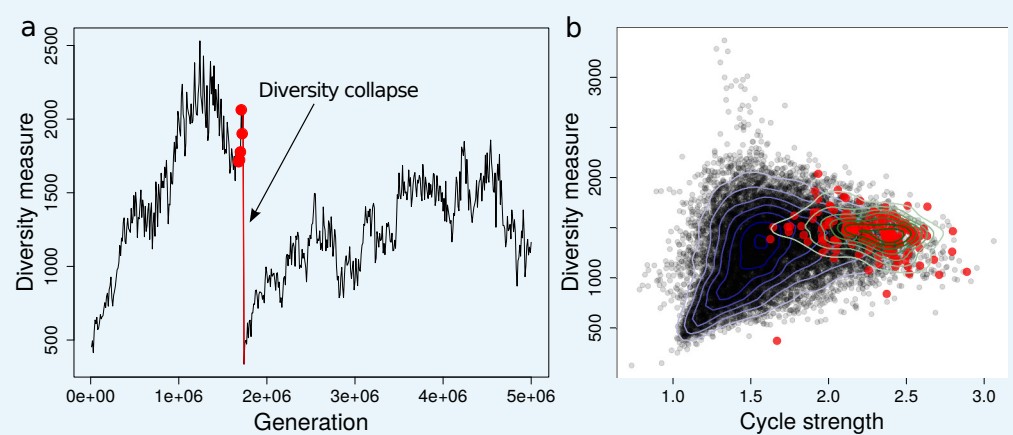

**Appendix 1—figure 9.** (**a**) Diversity collapse in a sample simulation with $\delta = 0.4$, $\lambda = 3 \times 10^4$, $\mu = 0.001$ and $m = 0.02$. Diversity collapse (red line) is defined as a sharp decrease in diversity. Red dots mark 5 sampling time steps before the collapse. (**b**) Diversity versus average cycle strength for 24 different simulations each over $5 \times 10^6$ generations with $\mu = 0.001$, $m = 0.02$, $\delta = 0.28, 0.33, 0.4, 0.44, 0.5, 0.56$ and $\lambda = 1 \times 10^4, 3 \times 10^4, 1 \times 10^5, \infty$. The red dots highlight the five sampling time steps before each observed collapse. The overall distribution of sampled values, and the distribution of the (red) points preceding the collapses are over plotted as two sets of contours.

DOI: https://doi.org/10.7554/eLife.36273.017

In order to find a qualitative explanation for diversity collapses in ITEEM, we examined the relation between diversity and average cycle strength (**Appendix 1—figure 9b**). During simulations, these two quantities are usually correlated, but this correlation is blurred by the stochastic nature of eco-evolutionary dynamics. Sometimes, diversity increases faster than cycle strength or vice versa. If we highlight the time steps before the sudden collapses, we see that they always lie in the right part of the cycle strength distribution, which means that cycle strengths are larger than expected at these time points.

## Size of the system

We checked the effect of system size by comparing simulations with sizes $N_s = 1 \times 10^4, 3 \times 10^4, 1 \times 10^5, 3 \times 10^5$, $\lambda = \infty$ and a set of trade-off parameters ($0 \leq \delta < 1$). One of the diversity indexes (SMST) is plotted in **Appendix 1—figure 10** for different sizes. With increasing $N_s$ the final diversities in trait space increase, and the drop at $\delta \approx 0.7$ becomes sharper, which is typical of a phase transition.

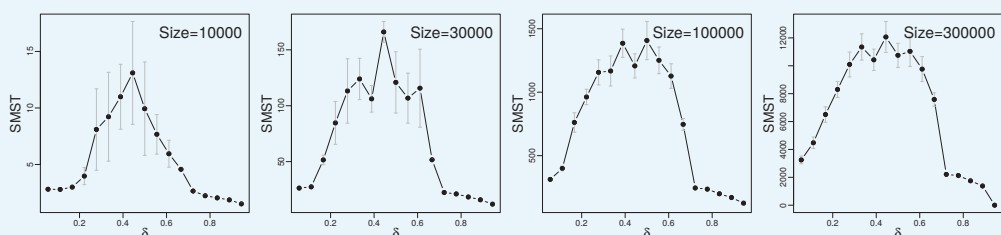

**Appendix 1—figure 10.** Functional diversity, measured by size of minimum spanning tree (SMST), as function of trade-off parameter for different system sizes $N_s$. Simulations with $\lambda = \infty$, $\mu = 0.001$, and $m = 0.02$.

DOI: https://doi.org/10.7554/eLife.36273.018

Plotting diversity versus size of the system for the middle rage of trade-off parameter $\delta$ in a log-log plot reveals a scaling relation with exponent around 2: $D \sim S^{1.97}$ (**Appendix 1—figure 11**).

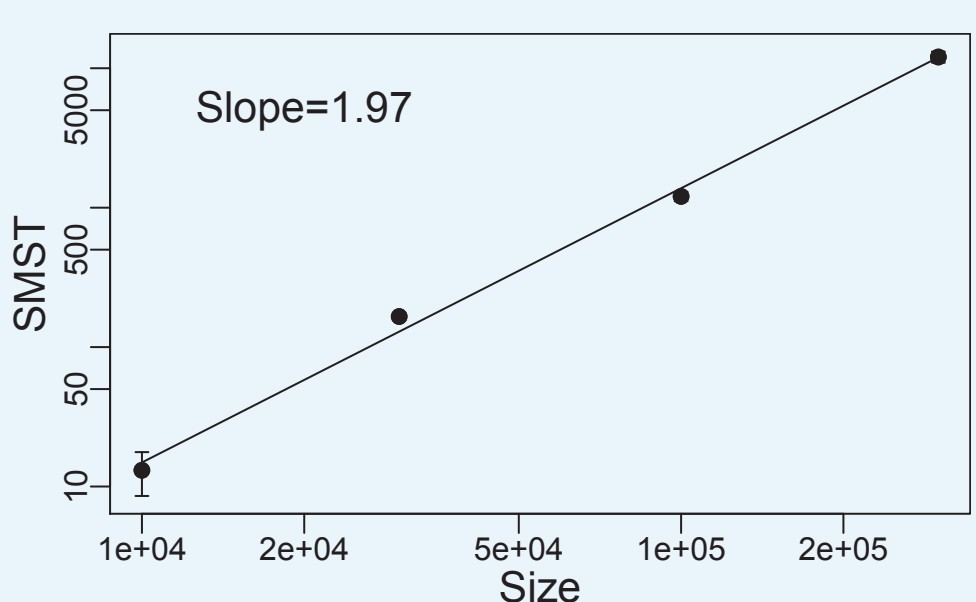

**Appendix 1—figure 11.** Functional diversity, measured by size of minimum spanning tree (SMST) versus size of the system in a log-log plot, averaged over middle range of trade-off parameter ($\delta = 0.44, 0.5, 0.56$). We see a scaling relation with exponent of $1.97$ between diversity and size of the system. Simulations with $\lambda = \infty$, $\mu = 0.001$, and $m = 0.02$.

DOI: https://doi.org/10.7554/eLife.36273.019

## Frequency-dependent selection

Frequency-dependent selection mediated by interaction of species could be a source of temporal correlation between eco-evolutionary 'events', for example speciation, invasion, and extinction of species. In the absence of such biotic selections, speciation and extinction of different species occur randomly with a constant rate without any autocorrelation in time. To examine if there is such a correlation we used the distribution of inter-event times, that is, the distribution of intervals between occurrence of consecutive events. For a completely random (Poisson) process – which is the null hypothesis for correlated speciation and extinction – this distribution follows an exponential distribution. Deviation from the exponential distribution is a signature of correlation between events. *Appendix 1—figure 12* compares the inter-event distribution of ITEEM data with the best fit of a geometric distribution (discrete version of exponential distribution) to the data. The clear deviation from the Poisson process supports that speciation and extinctions are not just random events, but that after occurrence of an event, with a delay ($\approx 10000$ generations), the probability of observing another event is higher than in a random process.

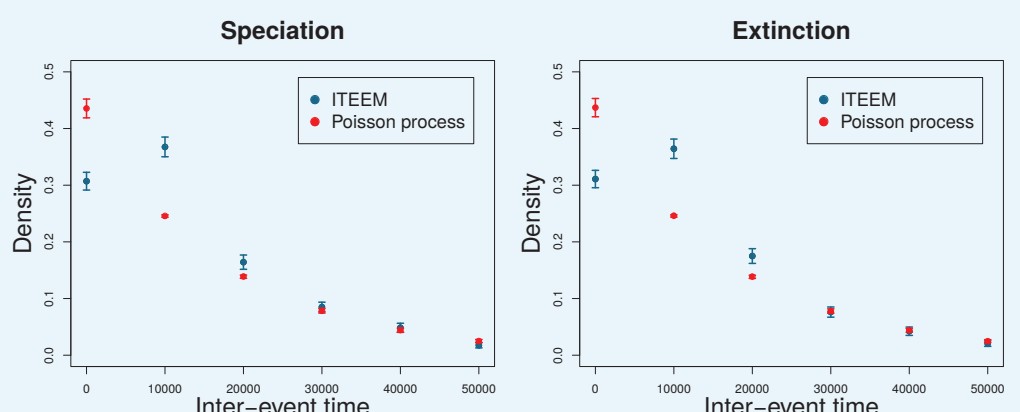

**Appendix 1—figure 12.** Distribution of time interval between speciation events (left) and extinction events (right). Blue points and error bars: data from 24 ITEEM simulations, each of $5 \times 10^6$ generations ($\delta = 0.5, \lambda = 10^5 \ldots \infty, N_S = 10^5, \mu = 0.001, m = 0.02$). Error bars are $\pm 2$ standard deviations calculated by bootstrapping. Red points and error bars: maximum likelihood fit (function fitdistr in R-package MASS, version 7.3–44) of the simulated data to a geometric distribution (discrete version of an exponential distribution), corresponding to an assumed Poisson process. Error bars are $\pm 2$ standard deviations estimated by a maximum likelihood fit.

DOI: https://doi.org/10.7554/eLife.36273.020

## Mutation probability

*Appendix 1—figure 13* shows the behavior of a typical diversity index (SMST) as function of $\delta$ and $\lambda$ for different mutation probabilities $\mu$. The overall dependency on trade-off and lifespan is the same for a wide range of $\mu$, but the value of diversity indexes depend on it: the smaller the $\mu$ the lower the diversity in community. $\mu$ also affects the rate of increase in diversity (*Appendix 1—figure 14*).

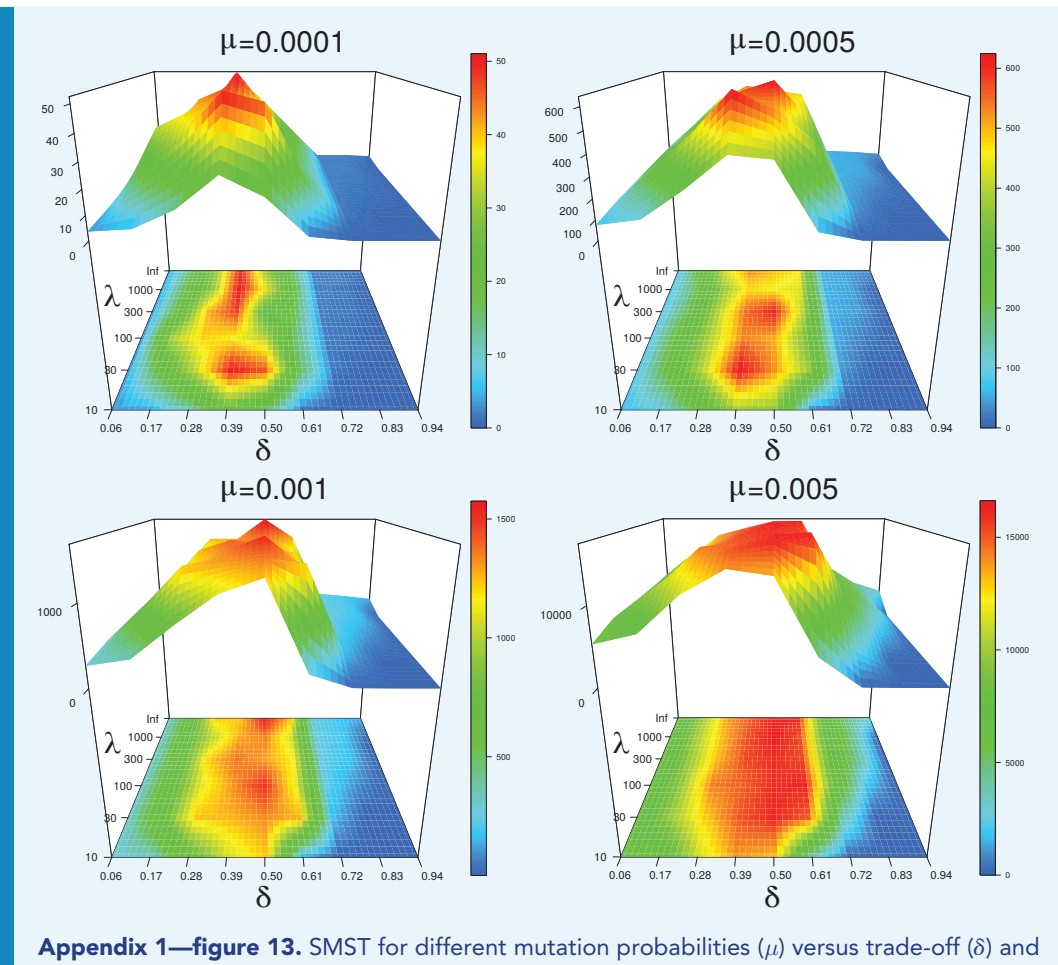

**Appendix 1—figure 13.** SMST for different mutation probabilities ($\mu$) versus trade-off ($\delta$) and lifespan ($\lambda$).

DOI: https://doi.org/10.7554/eLife.36273.021

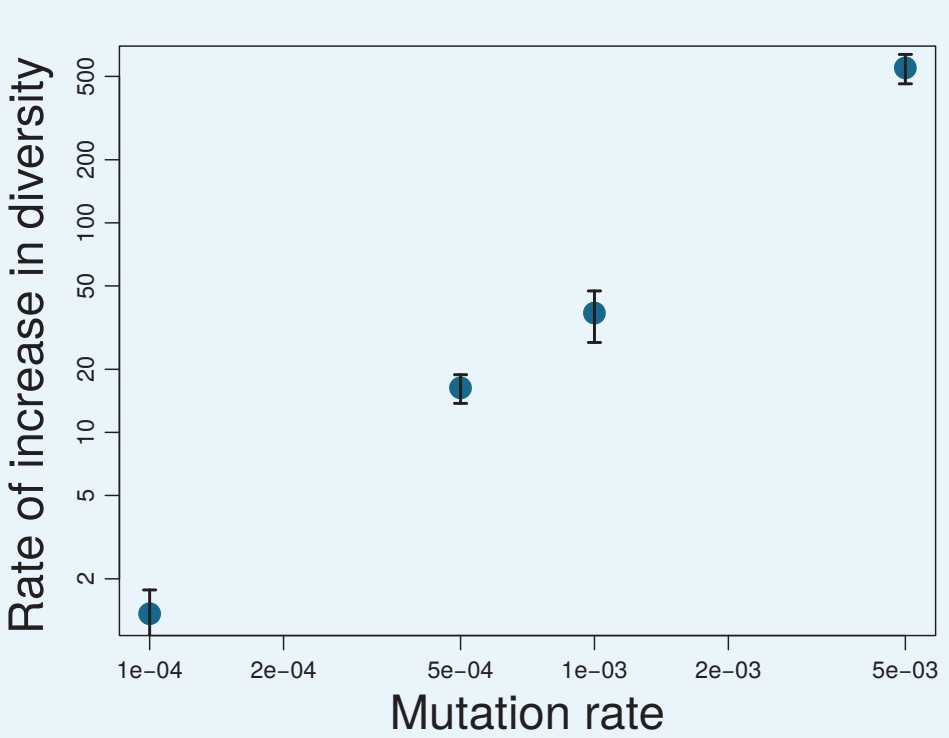

**Appendix 1—figure 14.** Rate of increase in diversity (measured as increase of SMST per 10000 generations) for different mutation probabilities $\mu$. Rate of increase in diversity is calculated by fitting a line to the first 80000 generations of each simulation and averaging is over five different simulations. Error bars show the errors estimated by the fit. Note the log-log scale of the plot.

DOI: https://doi.org/10.7554/eLife.36273.022

## Neutral model

To compare the diversity generated by genetic drift (neutral model) with diversity generated under selection pressure induced by competition with moderate trade-offs, we simulated neutral models in which all strains compete equally for resources, that means $I_{\alpha\beta} = 0.5$ for all pairs of strains, but traits evolve by mutation as before. In the neutral model, reproduction probabilities should also be the same for all strains, hence we attributed the same replication in each simulation to all strains. We carried out simulations for $r = 0.1, 0.5, 0.9$. The distribution of strains over trait space (*Appendix 1—figure 15*) shows that genetic drift is able to spread trait vectors in trait space and to produce a small cloud of strains, but the diversity generated in this way is much smaller than that of communities evolved under biotic selection pressure mediated by competition under moderate trade-offs (compare scale to that of bottom left panel of *Appendix 1—figure 3*).

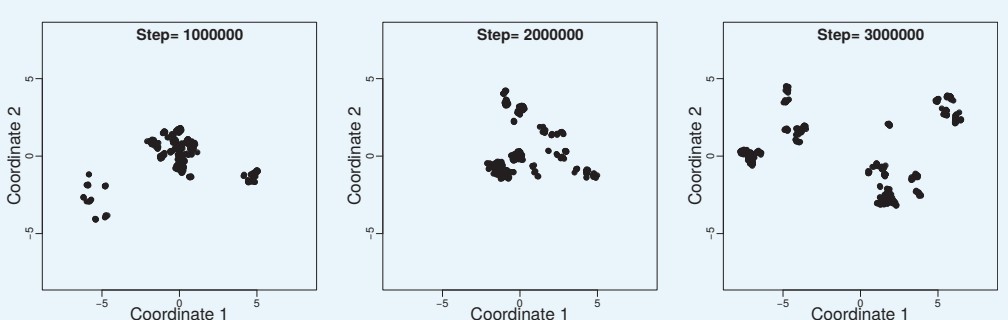

**Appendix 1—figure 15.** Trait space of a community evolved under neutral model at three time steps: $1 \times 10^6$, $2 \times 10^6$, and $3 \times 10^6$ with $\lambda = 100$, $r = 0.9$, $\mu = 0.001$, and $m = 0.02$. Note the small size of the trait space in comparison to a non-neutral model (bottom left panel of *Appendix 1—figure 3*).

DOI: https://doi.org/10.7554/eLife.36273.023

In order to show clearly the difference between the diversity produced in both models we also studied diversity measures and other parameters. The to panel in each column of *Appendix 1—figure 16* follows changes of a typical diversity index (SMST) over time for a simulation of genetic drift for one lifespan (with three different reproduction probability), and compares these changes with those in a typical simulation with competitive selection pressure for a moderate trade-off ($\delta = 0.56$).The bottom panels illustrate the corresponding comparison for cycle formation. The relative strengths of cycles in these simulations have large fluctuations around a value less than one, without any stable pattern over time. This means that community dynamics in the neutral model is determined by fluctuations, as expected from a model dominated by random genetic drift.

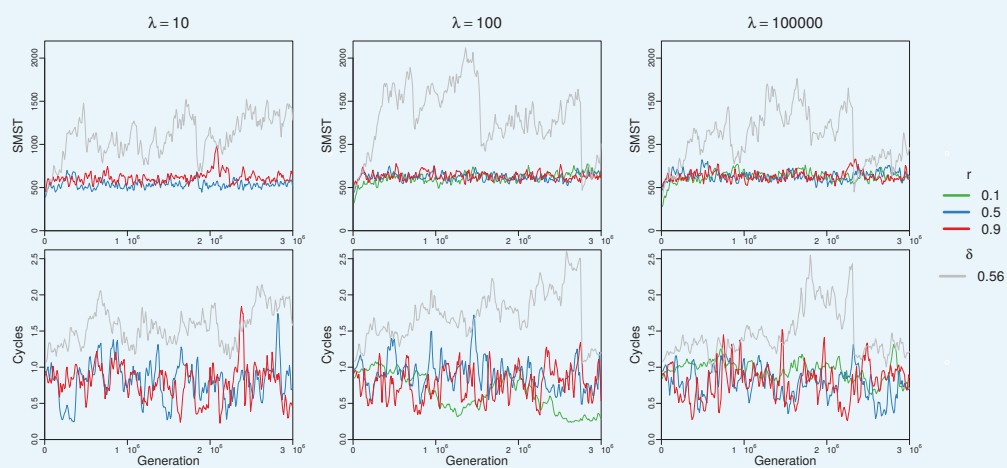

**Appendix 1—figure 16.** Changes of SMST (top panels) and relative strength of cycles (bottom panels) over time for three different lifespans. Colored curves are the results of the neutral model with different reproduction rates ($r = 0.1, 0.5, 0.9$). The results are compared with the outcome of one simulation with $\delta = 0.56$ and the corresponding lifespans (gray curves). For $\lambda = 10$ and $r = 0.1$ the population went extinct very quickly.

DOI: https://doi.org/10.7554/eLife.36273.024

## Phenotype-interaction map

A map between two spaces, for example interaction and phenotype space, can be constructed by the rules and laws that link them. The interaction of two individuals is a function of their phenotypic traits. Generally, this can be a complex relation with different functionality of different traits. When this function is known, any phenotypic variation can be mapped into the interaction space. To generally investigate this map, we borrow the term

*competition kernel* from adaptive dynamics theory as a phenotype-based model (***Doebeli, 2011***). A competition kernel measures the competitive impact of two individuals from different strains with different traits, i.e. for two individuals from strains $\alpha$ and $\beta$, the competition kernel is a function $a(\mathbf{x}_\alpha, \mathbf{x}_\beta)$ where $\mathbf{x}_\alpha = (x_{1\alpha}, x_{2\alpha}, \ldots)$ and $\mathbf{x}_\beta = (x_{1\beta}, x_{2\beta}, \ldots)$ are the phenotypic trait vectors of the corresponding strains. Elements of these vectors can be any relevant phenotype like size, color, expression of a gene, etc. Considering that in our model the carrying capacity is fixed and equal for all traits, interaction terms of ITEEM are proportional to the competition kernel, $I_{\alpha\beta} \propto a(\mathbf{x}_\alpha, \mathbf{x}_\beta)$.

The dimension of the phenotype space is equal to the number of traits with each axis representing a trait, and strains are distributed according to their traits over this space (***Appendix 1—figure 17a***). Interaction space, on the other hand, has one axis for each strain, and individuals are distributed based on their interactions with the strains that represent the axes (***Appendix 1—figure 17b***). The dimension of interaction space is dynamic because it increases with new strains and shrinks as strains go extinct.

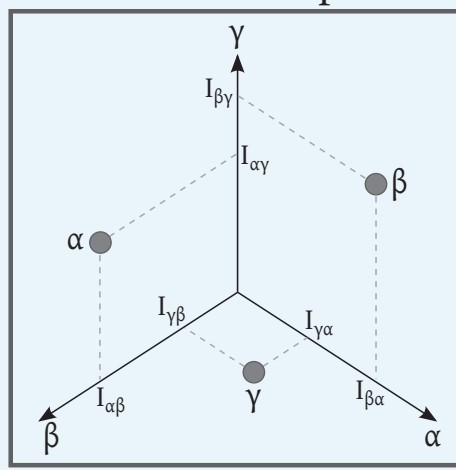

a Phenotype space b Interaction space

**Appendix 1—figure 17.** Distribution of strains in a) the phenotype space b) the interaction space. This system consists of 3 strains with two phenotypic traits. Here, without loss of generality, we ignore intra-specific competitions. Interaction terms are obtained from the competition kernel: $I_{\alpha\beta} \propto a(\mathbf{x}_\alpha, \mathbf{x}_\beta)$.
DOI: https://doi.org/10.7554/eLife.36273.025

A new, phenotypical mutant strain appears in phenotype space close to its parent (***Appendix 1—figure 18a and c***). Any phenotype variation that is not ecologically neutral and produces a new 'interaction' mutant, i.e. a strain with a novel interaction vector, adds a new dimension to the interaction space (***Appendix 1—figure 18 b and d***) while it is still close to its parent, as expected from their ecological similarity.

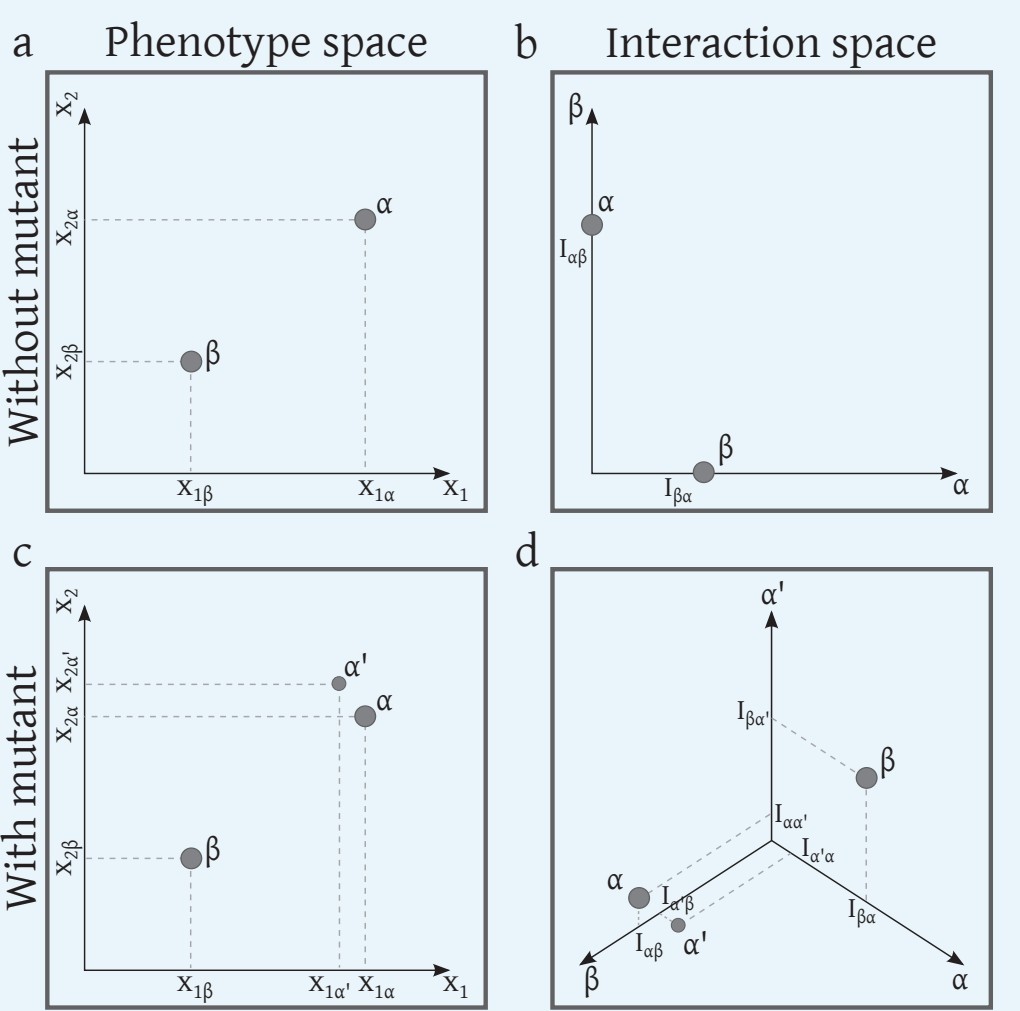

**Appendix 1—figure 18.** The phenotype and the interaction space of a system that first consists of 2 strains with two phenotypic traits and then is invaded by a mutant.

DOI: https://doi.org/10.7554/eLife.36273.026

A relevant question about interaction based-models could be if the random Gaussian variations in the interaction traits are biologically meaningful or not. The key to answering this question is the competition kernel, that is, how phenotypes are translated to the interactions, and hence depends on its functionality. However, if we consider that a mutant should be ecologically similar to its parent, the assumption of random Gaussian variations appears justified; it is also in line with models at the phenotype and genotype levels. Evolution is exploring the trait space, which, depending on the model, could be genotype, phenotype or interaction space. This exploration can be random (neutral evolution), or directional (adaptive evolution). In evolutionary models in phenotype space, usually phenotypic variations are drawn from a Gaussian distribution around the parent's trait vector. The fate of these mutants are then determined by either genetic drift, or the fitness landscape, or the community, i.e. when frequency-dependent selection is the driving evolutionary force. If we use such Gaussian distributions to model phenotype traits, we can use the competition kernel to map them into interaction space and test if such variations produce random variations in the interaction traits.

Consider a system with $P$ phenotypic traits in which each strain $\alpha$ is described by its trait vector $\mathbf{x}_\alpha = (x_1, x_2, \ldots, x_P)$. A mutant offspring $\alpha'$ should have a trait vector that is a random variation of the trait vector of its parent $\alpha$, i.e. $\mathbf{x}_{\alpha'} = \mathbf{x}_\alpha + \boldsymbol{\nu} = (x_1 + \nu_1, x_2 + \nu_2, \ldots, x_n + \nu_P)$, where elements $\nu_i$ $(i = 1, \ldots, P)$ of $\boldsymbol{\nu}$ are drawn independently from a normal distribution. To map this system to the interaction space, we need the competition kernel. In a system with $N$ strains, the interaction trait vector of parent strain $\alpha$ is $\mathbf{T}_\alpha = (I_{\alpha 1}, \ldots, I_{\alpha N}) =$

$(a(\mathbf{x}_\alpha, \mathbf{x}_1), \ldots, a(\mathbf{x}_\alpha, \mathbf{x}_N))$ while the interaction trait vector of mutant $\alpha'$ is $\mathbf{T}_{\alpha'} = (I_{\alpha'1}, \ldots, I_{\alpha'N}) = (a(\mathbf{x}_{\alpha'}, \mathbf{x}_1), \ldots, a(\mathbf{x}_{\alpha'}, \mathbf{x}_N)) = (a(\mathbf{x}_\alpha + \boldsymbol{\nu}, \mathbf{x}_1), \ldots, a(\mathbf{x}_\alpha + \boldsymbol{\nu}, \mathbf{x}_N))$. As the phenotypic variations are small, we can approximate the $\beta$-th element of $\mathbf{T}_{\alpha'}$ with a Taylor series expansion:

$$I_{\alpha'\beta} = a(\mathbf{x}_{\alpha'}, \mathbf{x}_\beta) = a(\mathbf{x}_\alpha + \boldsymbol{\nu}, \mathbf{x}_\beta) = a(\mathbf{x}_\alpha, \mathbf{x}_\beta) + \mathbf{g}^T.\boldsymbol{\nu} + \frac{1}{2}\boldsymbol{\nu}^T\mathbf{H}\boldsymbol{\nu} + \ldots = I_{\alpha\beta} + \eta_\beta. \tag{10}$$

$\mathbf{g} = \frac{d}{d\mathbf{x}_\alpha} a(\mathbf{x}_\alpha, \mathbf{x}_\beta)$ and $\mathbf{H} = \frac{d}{d\mathbf{x}_\alpha}\mathbf{g}$ are the gradient vector and the Hessian matrix (first and second order derivatives in single variable case) of the competition kernel and $\eta_\beta$ is a random number with normal distribution (*Equation 1* in the main text). For the last equation we have used the central limit theorem, which states that the sum of many independent random variables is approximated well by a normal distribution.

Despite the fact that the precise influence of random phenotypic variations on the interaction trait depends on the functionality of competition kernel, the above approximation (*Equation 10*) shows that those variations can be mapped to random variations in the interaction terms. This means that if a mutant emerges with random variation of its parent's phenotype, its interactions with the extant strains are random variations of the interactions of the parent. It is important to mention that neither in phenotype nor in interaction space, random trait variation between parent and mutant offspring leads necessarily to random independent characters of parent and offspring. The latter is true only if evolution is governed by neutral drift. In adaptive evolution, the fate of a mutant is determined by the interaction of that mutant with the community or the environment.

One important aspect of the interaction level modeling is that the organisms are defined in this space by their interaction traits. Thus, variations that occur in different phenotype traits are coarse-grained into the interaction terms. Interaction space is not restricted by the dimensions of the phenotype trait and hence allows for evolutionary innovations that happen due to emergence of new phenotypic dimensions, for example a novel metabolic pathway activated by epigenetic changes. However, this coarse-graining neglects phenotypic variations that do not affect ecological interactions, and thus maps different phenotypic mutations that lead to the same effective ecological interactions to the same interaction term, for example if those phenotypic variations yield the same ecological dominance (*Appendix 1—figure 19*). This is similar to the genotype-phenotype map if several genotypes are mapped to the same phenotype. The non-injectivity between the phenotype and interaction space is not an issue when the ecological outcome of the eco-evolutionary dynamics is studied.

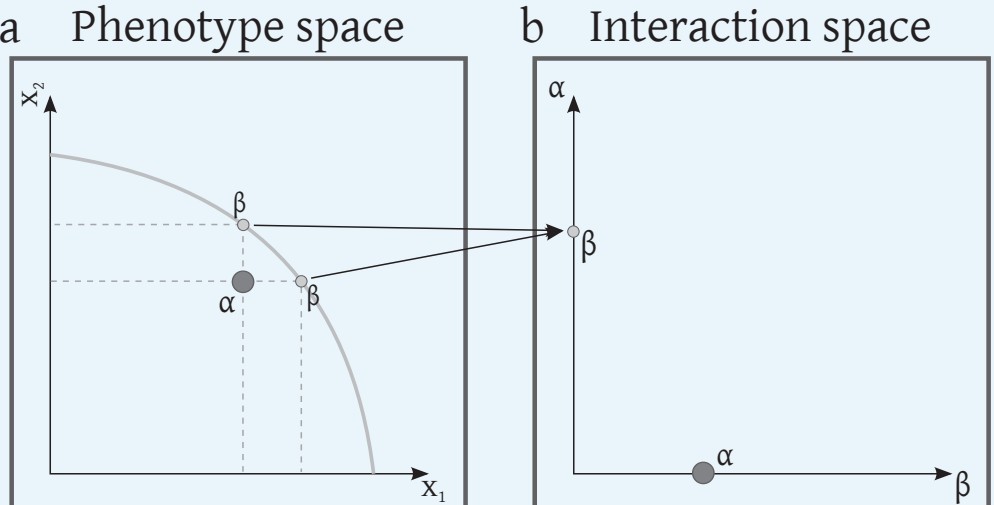

**Appendix 1—figure 19.** Non-injectivity of the phenotype-interaction map. Depending on the competition kernel, several phenotype arrangements can be mapped to the same interaction

arrangement. For example if we consider an asymmetric competition kernel like $a(\mathbf{x}_\alpha, \mathbf{x}_\beta) =$

$\sum_{i=1,2} d\left(1 - \frac{1}{1+be^{-c\left(x_{\alpha i}-x_{\beta i}\right)}}\right)$ (**Kisdi, 1999**), the two imaginary phenotypes that are shown as $\beta$ in (**a**)

will be mapped to the same point in the interaction space. Remember that these two are not present at the same time but both are possible phenotypic traits that give rise to the same interaction term with $\alpha$. In fact in this example for the aforementioned competition kernel set of points that all map to the same interaction term with $\alpha$ form a curve (thick gray line).

DOI: https://doi.org/10.7554/eLife.36273.027

