## [Decision Letter]

[Editors’ note: a previous version of this study was rejected after peer review, but the authors submitted for reconsideration. The first decision letter after peer review is shown below.]

Thank you for submitting your work entitled "Trade-off shapes diversity in eco-evolutionary dynamics" for consideration by *eLife*. Your article has been reviewed by four peer reviewers, one of whom is a member of our Board of Reviewing Editors, and the evaluation has been overseen by a Reviewing Editor and a Senior Editor. The following individuals involved in review of your submission have agreed to reveal their identity: Yaroslav Ispolatov (Reviewer #3).

Our decision has been reached after consultation between the reviewers. Based on these discussions and the individual reviews below, we regret to inform you that we have decided to reject your paper. However, we would like to offer the possibility of resubmission of a substantially revised manuscript. Our detailed assessment is as follows.

The paper has been carefully read by four expert reviewers. All of the reviewers found your paper thought provoking and potentially interesting. However, three reviewers raised substantial concerns. Two reviewers question the biological relevance of the models considered in the paper, and they think the model is too abstract to be useful for biologists thinking about long-term evolution. Since *eLife* is a "generalist" journal, it is imperative that papers have a high relevance not only for a narrow theoretical audience, but also for a general audience, in this case primarily in ecology and evolution. The reviewers also raised concerns that the work was not put into the context of existing theoretical work. In particular, one reviewer was concerned that a very similar approach has been taken in a previous paper, which is cited in the manuscript, but not discussed at all. All these concerns would need to be addressed in a revision. Specifically, the authors need to convince a mainly biological audience that the work presented is interesting and relevant, and they need to be clear about what advances the current work represents over previous work. Consequently, a serious effort would be required for revising the paper, and there is no guarantee for a positive outcome. Should you choose to resubmit a revised version, you would also need to take into account all the other reviewer comments in a constructive manner.

If you feel you can make the necessary revisions, we will make every effort to return a new version of the manuscript to the same editors and reviewers. Please note that this would be treated as a new submission.

Reviewer #1:

This is an interesting paper presenting a "minimal" model for the evolution of diversity in communities governed by competitive interactions. The model is minimal in the sense that only the outcome of interactions is modelled, but not their mechanistic basis, which greatly simplifies the description.

I have two main concerns, and a number of more technical points.

Essential revisions:

1) The work presented appears to be very similar to the theory presented in Shtilerman et al., (2015). The authors need to clarify to what extent their work is novel compared to that earlier paper.

2) In fact, I think the logistic equations presented by the authors in the appendix are similar to the ones used in Shtilerman et al., 2015. I think the authors need to consider using these equations, rather than their individual-based simulations, to derive their results. It appears to be straightforward to implement the evolutionary dynamics based on the deterministic logistic equations, with one differential equation per species, and with evolution being implemented by adding new equations to the system (and deleting those ones in which the frequency falls below a threshold). This would yield a computationally more efficient model, whose salient features would nevertheless be essentially the same as in the individual-based model (perhaps barring the nearly neutral variation within species clusters). If the results are not the same, then this needs to be known, as it would probably point to an important role of stochasticity. I think this line of analysis is necessary to obtain a complete picture of the system studied by the authors, which is required for *eLife*.

Essential revisions:

– Subsection “Model”: the procedure for the individual-based model is not clear: the text states that "we try 𝑁𝑖𝑛𝑑(𝑡) (number of individuals at that time step) replications of randomly selected individuals. Each selected individual of a strain 𝛼 can replicate with rate r_α…" this is imprecise; what does is it mean that an individual that has already been selected for replication replicates with a rare r_\α? Isn't it rather the case that individuals get selected to produce an offspring with a probability that is proportional to r_α?

– Also, it is not clear how the discreteness of time is dealt with exactly. For example, the paper says that each offspring is assigned a randomly selected site, then competes against the occupant of the site and probabilistically takes over. What if one offspring takes over, but then a second offspring is chosen (by chance) to compete for the same site? Is that second offspring then competing against the previous occupant, or against the new occupant, i.e., the offspring that has already taken over from the previous occupant?

– And when do individuals die? Before or after offspring production?

– The authors mention the existence of rock-paper-scissor interactions, but it is not at all clear what the importance of such interactions are for the overall dynamics of the system. Are such interactions very common, essential, just a quirk?

– Subsection “Impact of trade-off and lifespan on diversity”: it is not clear what "average cycle strength" is. Which cycles were used to calculate the average of what?

– Subsection “Impact of trade-off and lifespan on diversity”: shouldn't the average interaction strength, i.e. the average values of all I's, be ca. 0.5?

– I failed to understand Figure 2. Please explain better.

– Subsection “Impact of trade-off and lifespan on diversity”: the Darwinian demon doesn't really make sense: with no diversification, there is only one strain present, and it is impossible to say what kind of competitive ability this strain has, because no other strain is present… so what are you actually trying to say?

– I don't find the section about the relationship between tradeoff and productivity very convincing, because it is too vague. I think the authors need to be more specific and detailed here.

– Section "Frequency-dependent selection": I think a more mechanistic explanation for differences of speciation and extinction rates compared to Poisson processes is needed. What is the cause of these differences? When is this mechanism at play, and when not? In particular, why do speciation rates increase with the diversity in the system? Shouldn't this only happen at certain intermediate levels of diversity, because when the community reaches a high-level diversity, speciation rates presumably decline?

– I would like to bring the authors' attention to the recent paper by Doebeli and Ispolatov, (2017), which also presents models for community assembly and evolution of diversity are presented, with some parallels to the work discussed here. These parallels could be taken up in the Discussion section.

Reviewer #2:

The authors present a model for studying diversification dynamics as an outcome of, as they claim, competition for a single resource. I have substantial concerns regarding the insight that could be gained from this work, mainly due to the type of model and the lack of a mechanistic interpretation.

Subsection “Model” and subsection “Frequent-dependent selection”: Your trait space is Nsp-dimensional, where Nsp is the number of extant strains. In particular, the trait space grows explicitly with every strain added. Is it surprising that you get coexistence of Nsp strains in an Nsp-dimensional trait space? Further, is it surprising that "emergence of new species increases the probability for generation of further species"? It seems to me that this is quite expected, given your model. This also explains why you get these mass extinction events; there's a positive feedback loop between speciation rate and species number that is easy to recognize even without simulations.

Subsection “Generation of Diversity”: Please explain how you define "functional diversity" in your model at first mention. Appendix 6 was uninformative. The only information I found was in the caption of Figure 1, where you define functional diversity "in terms of the size of minimum spanning tree (SMST) in trait space". But your trait space is continuous ([0,1]Subsection “Generation of Diversity”: Please explain how you define "functional diversity" in your model at first mention. Appendix 6 was uninformative. The only information I found was in the caption of Figure 1, where you define functional diversity "in terms of the size of minimum spanning tree (SMST) in trait space". But your trait space is continuous ([0,1]Subsection “Generation of Diversity”: Please explain how you define "functional diversity" in your model at first mention. Appendix 6 was uninformative. The only information I found was in the caption of Figure 1, where you define functional diversity "in terms of the size of minimum spanning tree (SMST) in trait space". But your trait space is continuous ([0,1]Subsection “Generation of Diversity”: Please explain how you define "functional diversity" in your model at first mention. Appendix 6 was uninformative. The only information I found was in the caption of Figure 1, where you define functional diversity "in terms of the size of minimum spanning tree (SMST) in trait space". But your trait space is continuous ([0,1]^Nsp) so a priori there is no tree structure connecting species (apart from phylogeny). So how do you define a spanning tree, and how is this spanning tree not a function purely of the number of strains?

Subsection “Generation of Diversity”: Related to the previous point. What is a "functional niche" in your model? Since you did not discuss mechanisms underlying the interaction matrix, it is not clear what a function is.

Subsection “Generation of Diversity”: Saying your model has no geographical isolation nor resource partitioning sounds meaningless. You have not specified the underlying mechanisms for your "interaction trait", so it is hard to draw a comparison to other more mechanistic models (i.e. where the physiological/metabolic traits are explicit).

Subsection “Model”: Since replication is non-sexual, what is the purpose of defining "species" separately from "strain" in your model? In Appendix-I you mention that you define "species" operationally based on divergence time and using some arbitrary cutoff threshold, so this sounds a bit analogous to operational taxonomic units (OTUs) in microbiology. However, you model mutation/selection/growth dynamics at the strain level. Please explain early on what additional insight one may get from counting "species".

Discussion section: You claim that modeling in terms of interaction traits (which in your case means in terms of outcome probabilities of local competitive exclusion) "coarse-grains these complex systems in a natural, biologically meaningful way.". But many of your interpretations and comparisons to other models or data require translating your Abstract "interaction traits" to functions or life histories. You did not discuss at all what real traits could possibly give rise to your interaction trait matrix. Your interaction matrix seems to loosely represent competitive interactions; but then it can only explain diversity within a single trophic level (e.g. in the case of animals) or a single metabolic niche (e.g. in the case of bacteria); yet, you keep referring to "functional diversity".

Discussion section: Related to the previous comment. You say that your formulation was "chosen to reflect reasonable properties" and that you "have assumed a single, limiting resource in a well-mixed system". However, you did not provide any plausible mechanism for how such an interaction matrix could arise from competition for a single resource in a well-mixed system. This is actually a big question: how can one obtain a Nsp-dimensional trait space through competition for a single resource pool?

*Reviewer #3:*

The manuscript "Trade-off shapes diversity in eco-evolutionary dynamics" presents a quite original evolutionary-ecological individual-based model based on a compromise between growth rate and competitiveness. The model is simple but exhibits quite rich evolutionary properties, which, while being not entirely unpredictable, are intriguing and provide useful insights and generalizations. This combination of simplicity of the rules and relevance of the dynamics usually distinguishes successful and long-living models from the rest and makes them understandable and appealing to a wide audience. Besides, I cannot see any potentially "hidden" flaws in the model and interpretation of results that could cast doubts on the main conclusions. Thus, in my opinion, the manuscript may be published in *eLife* after the following and, perhaps, other comments are addressed:

1) The complexity of algebra in (3) and subsequent definition of s is definitely unwarranted by an otherwise quite accessible level of math in the rest of the paper. Furthermore, it apparently confuses even the Authors when they classify the domains of weak and strong trade-offs:

It looks like the definitions of high-tradeoff (\δ ~ 1) and intermediate trade-off (\δ ~ 1/2) are misleading. The strongest dependence of r on C appears to be when \δ=1/2, which also follows from Figure 1. The phrase "In high-trade-off phase III, any small change in C changes r drastically" is simply wrong for all but very small C. I would call both the \δ=1 and \δ=0 limits as small tradeoffs as they look perfectly symmetric, or, even better, choose another, more heuristic and transparent, form of parametrization of r vs. C.

2) Would it be possible to say anything about population dynamics of "species"? Is it cyclic?

3) At least qualitatively, what are mechanisms of mass extinction?

4) An explicit plot of diversity vs. system size (perhaps just for the δ optimal for diversity) and, ideally, an estimate of corresponding scaling, would be very revealing.

5) Similarly, how the level of diversity and the typical number of traits in a species depends on the mutation amplitude m?

6) For a general reader of *eLife*, the MDS algorithm needs to be explained and properly referenced. It plays a major role in interpreting the results, however, is presented only by the name of the corresponding function in R.

7) A qualitative explanation about the minimal spanning tree analysis would help as well.

Reviewer #4:

I've carefully read the paper "Trade-off shapes diversity in eco-evolutionary dynamics" by Farnoush Farahpour and colleagues. In it, they use an individual-based eco-evolutionary model to understand the emergence of community structure based on evolving interactions. The model produces some interesting patterns, such as clustering in trait-space and the evolution of intransitive competitive loops.

There are a number of aspects of this paper that I liked: the focus on evolution of interactions, the emergence of intransitive loops, the dimension reduction applied to the vector-valued traits, and the comparison with a neutral model variant. These are all creative contributions to the modeling literature.

While I was intrigued by the idea of modeling the evolution of interactions directly, it was hard for me to connect it to real ecological systems. The interactions between species are not determined by phenotypic traits of the organisms but evolve independently. It's based on an unstated assumption that species interactions are totally idiosyncratic and unpredictable. I feel that evolution in this model is too unconstrained, despite the trade-off between competitive ability and reproduction, resulting in the prevalence of intransitive loops. As a complex-systems researcher, I'm fascinated, but as an evolutionary ecologist, I'm skeptical.

The authors need to do a better job putting their work in the context of the extensive literature on eco-evolutionary dynamics. The text had many statements that had me scratching my head. Examples:

1) Framing the problem in terms of resources, but in the model there are no actual resources.

2) The competitive exclusion principle only holds at equilibrium (Armstrong, Levins) and must count resources plus shared predators (Levin, 1970).

3) The "eco-evolutionary models" cited in the Introduction seem to be just ecological models.

4) In the Introduction, the "observed eco-evolutionary dynamics" cited that the model "closely resembles" aren't empirical patterns observed in real systems, but just results of other models.

5) Discussion of speciation overlooks that the species here are clonal, so what's hard about speciation?

Also, some of the sentences throughout were hard to understand the meaning of. E.g., "Evolutionary changes at the genetic level influence ecology if they cause phenotypic variations that affect biotic or abiotic interactions of species which in turn changes the species composition and occasionally forces species to evolve their strategies."

Some of the details of the model implementation weren't clear. For example:

How exactly do births happen (subsection “Model”)?

Is mu a mutation "rate" or probability of mutation during a replication event (subsection “Model”)?

Why is lifespan drawn from a Poisson distribution (Subsection “Model”) and how can that be infinite (Figure 2)?

If each individual stays in a site, is it really well-mixed?

Does mutation of one species' interaction coefficients end up changing another species' reproduction rate through the trade-off (2)?

Could you not get at the same questions more efficiently using deterministic Lotka-Volterra dynamics?

[Editors’ note: what now follows is the decision letter after the authors submitted for further consideration.]

Thank you for resubmitting your work entitled "Trade-off shapes diversity in eco-evolutionary dynamics" for further consideration at *eLife*. Your revised article has been evaluated by Ian Baldwin (Senior editor), a Reviewing editor, and 4 reviewers (the same reviewers as for the initial submission, reports below).

The reviewers agree that substantial effort was put into the revision. However, there are a number of remaining issues that call for a further in-depth revision. Reviewers 1, 2 and 3 (who was previously reviewer 4) still have substantial concerns about aspects of the paper. These concerns are all related to issues raised in the initial reports and will need to be addressed in a constructive manner if the paper is to be published in *eLife*.

Essential revisions:

1) Individual-based vs deterministic systems: both reviewer 1 and 4 (formerly reviewer 3) point out that it is not at all clear that these different approaches would yield different results. In other words, it is not clear that salient results reported in the paper depend crucially on the presence of noise, as appears to be the contention of the authors. This needs to be explored at least to some extent. This issue cannot be dealt with by simply "citing it away".

2) The assumption of independent evolution of the elements of the interaction matrix needs to be discussed in more detail and clarity, and in the context of biological realism. This point was raised by both reviewers 1 and 3 (formerly reviewer 4).

3) The "single resource" issue is related to point 2, and also needs a revised treatment.

4) I am sympathetic with reviewer 3's (formerly reviewer 4) concern about some of the references to previous work, particularly with regard to tradeoffs.

5) Please also address the remaining points, e.g. the definition of "species" raised by reviewer 2, as well as other minor points.

Reviewer #1:

The authors did a very good and thorough job in revising their paper. Almost all of my concerns have been addressed satisfactorily. The one remaining issue is that I don't buy the authors case for only using individual-based simulations (my original comment 2). The authors state in their rebuttal that "It has been extensively discussed in the literature that continuum approaches are unsuitable in cases of non-equilibrium dynamics,.…", and they somehow conclude from this that the deterministic Lotka-Volterra description (which in my mind is the same as a "logistic" description) would not be appropriate for the problem at hand. But that is exactly the question: in some general sense, one would expect that in the limit of large population sizes, the individual-based model used by the authors would converge to some deterministic model, and my guess is that this model would be at least close to the "mean-field" Lotka-Volterra model. It then becomes important to understand just which features of the individual-based models can be understood by studying the much simpler mean-field model. The authors present no arguments why there even are *any* features of their individual-based model that could not be observed in the deterministic model. They refer to near-neutrality, but that's exactly one of the features that was observed in similar deterministic Lotka-Volterra models by Shtilerman et al., (2015). I am not convinced that the salient results reported in this paper could not also be obtained with deterministic models. Just claiming that some results cannot be obtained in that way on general grounds does not make it true in this particular case. The obvious advantage of using deterministic models would be that such models are much more tractable analytically (e.g. from a statistical physics point of view), and it is therefore important to know how far one can get using them. I think it would not be too onerous to at least do some tests using the deterministic models to either confirm or refute the claim that they can produce similar results as the individual-based models. The question is: does stochasticity really play a major role in producing the results reported in this paper? If so, then this would be important to know (but this point would need to be made based on more than just a vague statement that their "system falls into the category of those better modeled by individual- based models"). If not, then the deterministic models should do a good job reproducing these results.

Reviewer #2:

The authors have partly addressed my concerns and those of the other reviewers. I do however still have two major concerns:

1) I agree that a low-dimensional phenotype space (e.g. pertinent to exploitation of/competition for a single resource) can give rise to an Nst x Nst interaction matrix that encodes the competitive interaction between strains. However, the crucial assumption of the authors is that each term in this matrix (well, half of the terms) can vary independently. How this could come about in reality is unclear to me.

In other words, if "P" is the underlying phenotype space (solely related to consumption of/competition for the common resource) and "I" is the space of possible interaction matrices, what could the mapping f:P->I possibly look like, such that f(P) is an Nst x Nst/2 dimensional manifold?

I strongly recommend:

(a) Avoid any comparison to "single resource" models or real systems.

(b) Acknowledge early on that an important assumption of the model is that the terms of the interaction matrix (well, half of them) can in principle vary independently (i.e. are not constrained explicitly due to genetics or ecology). Whether this assumption is met in reality is an open question.

(c) Clarify in the discussion that this paper does not address the important question of how such a high-dimensional interaction trait space (i.e. with Nst x Nst/2 independent axes) might arise, or provide a plausible example.

2) The definition of "species" by the authors is still confusing and of questionable relevance. The authors define "species" operationally based on a cutoff threshold in phylogenetic distance. While this is common practice in microbial ecology (where such clusters are called Operational Taxonomic Units), few would claim that the emergence and disappearance of OTUs is comparable to "speciation" dynamics in sexually reproducing organisms.

What I also found confusing is that in their "response to reviewers", the authors explain that "species" are "well-separated non-transient clusters in trait space". This does not align with the definition provided in their manuscript (Appendix 1), where species are defined as "clusters of strains separated by long-lasting gaps in a phylogenetic tree". Are these definitions equivalent in your model?

While the emergence of clusters in trait space is indeed interesting, I would recommend not calling these clusters "species", since clusters in trait space need not always be monophyletic and could in principle also consist of distantly related strains that happen to have converged in trait space.

Reviewer #3:

This is reviewer 4 from the original submission again. This remains an interesting yet frustrating manuscript. The authors resisted many of the good suggestions from the other reviewers and myself in how they can place their manuscript in the broader context. In the end, it's the authors' manuscript, but I still think they could do a better job in the introduction and discussion to not confuse potential readers.

To me, the most interesting part of the manuscript is the idea that species interactions might be so high dimensional that it is best to focus on interaction traits that summarize many idiosyncratic phenotypes. This is described in the discussion but should also be highlighted more in the Introduction. The relationship between phenotypic traits and interaction traits should be clarified to better address comments 1 and 6-7 of reviewer 2. Maybe could be described as a rugged "phenotype-interaction map", in analogy to the idea of "genotype-phenotype maps"? By the way, this is a big assumption of the model, not an established empirical fact, but still an interesting basis for the theory.

Reviewer 2 and I were confused by statements about limiting resources and the competitive exclusion principle. In the revision, the authors still make statements like "GLV equations model competition over renewable resources" (Subsection “Model”), "we observe high diversity in a well-mixed homogenous system without violating the competitive exclusion principle" (subsection “Trade-off anchors eco-evolutionary dynamics in physical reality”) and "we have assumed a single, limiting resource" (Subsection “Power and limitations of ITEEM”). Such statements will misdirect many readers into thinking about resource competition, R* rules, and the impossibility of coexistence of more species than resources. This is not appropriate, because in this model, the species interactions are direct (interference competition) and idiosyncratic. Allelopathy among plants or microbes would be a more relevant example than resource competition. The authors should remove all mentions about resources in the paper, because they will only confuse readers.

Concerning my previous comment 4, please don't portray other theoretical results as empirical support for your new model. Keep them clearly distinct.

It's a big stretch to say claim "life-history trade-offs" are a missing ingredient in existing theory (Introduction). Almost all existing eco-evolutionary theory is built around trade-offs. Models without trade-offs are the exception, not the rule.

The references cited in subsection “Model”, do not represent current, mainstream ecological thinking.

Reviewer #4:

I think that the authors addressed the main points from my previous review, just a few minor issues remain.

I still disagree with the definition of “strong' and “weak' tradeoff limits, seeing, for example, plots in Figure 1 as completely symmetric. In my opinion, the strongest dependence of birthrate on competitiveness happens at the line in the middle of the plot, presumably for δ=0.5. I guess it's more a terminological discussion, however, I find the definition of the strong tradeoff adopted by the Authors rather confusing.

Caption to Figure 2, “Disc diameter scales with total abundance of species' Does it mean that it scales with the number of individuals in a species? Or the number of species in the system? What kind of scaling is it?

Subsection “Generation of diversity”, “Occasionally diversity collapses from medium levels abruptly to very low levels, usually followed a recovery”, should it read “by a recovery”?

Appendix 1 (Eq.8) Is γ the index of summation, running from one to N_st? If not, what is the index?

I don't know if I should dwell on that, yet I also strongly disagree with the Authors' reply to the second comment of the first reviewer, and especially, with the apparent misuse of the term “mean field'.

In short, I believe that both implementations of this process, the individual-based and continuous-populations models should yield very similar phenomenology. Both those implementations are mean field in their nature as neither has any spatial correlations (in phenotypic or geographical space) or long temporal memory. The only difference between those is the presence of some stochasticity or noise in the individual-based model. If the Authors truly believe that such noise is the necessary source of the observed phenomenology, it should be clearly stated in the manuscript. Which, I believe, would have strongly depreciated the generality of conclusions. However, I don't think this is the case; on contrary, a continuous population version of this model would have enabled one to get “cleaner' results, speeding up the simulations and expanding the scaling range by including more species. The main distinct features that the model develops, such as temporal changes in the population, interaction cycles, speciations, etc., do not appear to be fluctuation-dominated. A minor related comment, contrary to what is said int he appendix, the per capita death rate can be included into the continuous description (which is often also called the logistic model as all elements of matrix A are negative) by simply reducing the birth rate.

---

## [Author Response]

[Editors’ note: the author responses to the first round of peer review follow.]

Our decision has been reached after consultation between the reviewers. Based on these discussions and the individual reviews below, we regret to inform you that we have decided to reject your paper. However, we would like to offer the possibility of resubmission of a substantially revised manuscript. Our detailed assessment is as follows.The paper has been carefully read by four expert reviewers. All of the reviewers found your paper thought provoking and potentially interesting. However, three reviewers raised substantial concerns. Two reviewers question the biological relevance of the models considered in the paper, and they think the model is too abstract to be useful for biologists thinking about long-term evolution. Since eLife is a "generalist" journal, it is imperative that papers have a high relevance not only for a narrow theoretical audience, but also for a general audience, in this case primarily in ecology and evolution. The reviewers also raised concerns that the work was not put into the context of existing theoretical work. In particular, one reviewer was concerned that a very similar approach has been taken in a previous paper, which is cited in the manuscript, but not discussed at all. All these concerns would need to be addressed in a revision. Specifically, the authors need to convince a mainly biological audience that the work presented is interesting and relevant, and they need to be clear about what advances the current work represents over previous work. Consequently, a serious effort would be required for revising the paper, and there is no guarantee for a positive outcome. Should you choose to resubmit a revised version, you would also need to take into account all the other reviewer comments in a constructive manner.If you feel you can make the necessary revisions, we will make every effort to return a new version of the manuscript to the same editors and reviewers. Please note that this would be treated as a new submission.Reviewer #1:This is an interesting paper presenting a "minimal" model for the evolution of diversity in communities governed by competitive interactions. The model is minimal in the sense that only the outcome of interactions is modelled, but not their mechanistic basis, which greatly simplifies the description.I have two main concerns, and a number of more technical points.Major concerns:1) The work presented appears to be very similar to the theory presented in Shtilerman et al., (2015). The authors need to clarify to what extent their work is novel compared to that earlier paper.

We have cited Shtilerman et al., (2015) in the previous and the current version of the manuscript as one of the most recent papers that use interaction-based modeling. Shtilerman et al., (2015) introduce their model as a modified version of the Ginzburg et al., (1988) model, a Lotka-Volterra model with mutation events. The main flaw – and insight – of the Ginzburg et al., (1988) model had been its failure to generate the stable high diversity observed in actual competitive communities. Thus, it became clear that important aspects of reality had been missed. One way of fixing this problem is the introduction of *ad hoc* modifications likely to stabilize complex communities, which is what Tokita and Yasutomi (2003), Shtilerman et al., (2015) and others did. The main *ad hoc* modification in Shtilerman et al., (2015) is that they introduced explicit speciation events coupled to a large and sudden change of interaction between parent and o spring species, so that competition pressure between a parent species and its o spring species is much smaller than the pressure within each group. However, in real systems evolution happens gradually, and, whenever a mutation occurs, the resulting o spring species is most similar to its parent species, and experiences the highest competitive pressure from the latter.

In contrast, our Interaction and Trade-o based Eco-Evolutionary Model (ITEEM) does not make these assumptions. In particular, it does not enforce speciation with an explicit mechanism, but novel species emerge out of the eco-evolutionary dynamics.

In the dynamics simulated in ITEEM, parent and o spring strain are in fact most similar, as in natural communities. The main conceptual advance that comes with ITEEM is the following. If we acknowledge the natural fact that eco-evolutionary dynamics is bound by physical constraints – in ITEEM acknowledged as trade-o between the abilities to compete and to replicate – major features of real competitive communities, such as stable high diversity, just emerge.

In a direct comparison with ITEEM, the model by Shtilerman et al., (2015) corresponds to the limiting case of zero trade-off, though with ad hocfeatures to enforce diversity. In terms of evolution, the Shtilerman et al., (2015) model could be considered closer to cladogenesis or punctuated equilibrium, while ITEEM models phyletic gradualism in an interaction-based formalism.

Besides, we study further aspects not addressed by Shtilerman et al., (2015) in their model, such as the effects of lifespan, system size, and mutation rate on diversity. Moreover, we think that the investigation of the role of cycles in the manuscript brings us closer to understanding the mechanisms behind the dynamics of diversity, be it in the growth of stable complex communities or collapses.

In the new version of the manuscript we have expanded the fourth paragraph of theDiscussionwithadescriptionofthehistoryofinteraction-basedmodels, including a description of Shtilerman et al., (2015).

2) In fact, I think the logistic equations presented by the authors in the appendix are similar to the ones used in Shtilerman et al., 2015. I think the authors need to consider using these equations, rather than their individual-based simulations, to derive their results. It appears to be straightforward to implement the evolutionary dynamics based on the deterministic logistic equations, with one differential equation per species, and with evolution being implemented by adding new equations to the system (and deleting those ones in which the frequency falls below a threshold). This would yield a computationally more efficient model, whose salient features would nevertheless be essentially the same as in the individual-based model (perhaps barring the nearly neutral variation within species clusters). If the results are not the same, then this needs to be known, as it would probably point to an important role of stochasticity. I think this line of analysis is necessary to obtain a complete picture of the system studied by the authors, which is required for eLife.

As explained in the Model section and Appendix 3, and now expanded in the Introduction and the last paragraph of section Model, ITEEM is an interaction based model for Lotka-Volterra dynamics of competitive communities. It is not a logistic model.

A standard way of simulating Lotka-Volterra dynamics has been to treat populations as continuous quantities, and to integrate continuous Lotka-Volterra equations over time. This avenue was chosen by Ginzburg et al., (1988) and all descendant models, including Shtilerman et al., (2015). A second standard approach is the use of individual-based models (IBMs) (Black and McKane, 2012; DeAngelis and Grimm,2014) that simulate the Lotka-Volterra dynamics at the level of individuals. We have used the second approach but we show in the Supplementary material that, if we apply a mean-field approximation to our model and treat the population as continuous, we formally obtain the well-known Lotka-Volterra equations.

It is important to realize that although the two approaches are equivalent in the mean-field limit, but correspond to distinct models of reality. Clearly, communities are composed of individuals, and disregarding this fact by replacing them by a continuum will lead to artifacts. It has been extensively discussed in the literature that continuum approaches are unsuitable in cases of non-equilibrium dynamics, if demographic variations influence the dynamics, when size of the system or discreteness of individuals matter or when chaotic dynamics can cause unexpected behavior, and that then stochastic, individual-based models are better models of nature (McKane and Newman, 2004; DeAngelis and Mooij, 2005; Black and McKane, 2012; DeAngelis and Grimm, 2014).

Since our system falls into the category of those better modeled by individual based models, we chose such a model right from the beginning and do not see the necessity to reiterate the above studies. Specifically, the following points of our study required an individual-based model: the eco-evolutionary dynamics is of a non-equilibrium type; we wanted to study the effects of demographic variations of lifespans; we had to study the effect of system-size, e.g. to test the observed phase transition.

To illustrate these points further, we refer to the important fact were the Reviewer herself/himself sees a difference between continuous and individual-based model, namely that the numerical solution of the continuum equations fail to produce neutral variations within species (“perhaps barring the nearly neutral variation within species clusters” in reviewer’s comment). However, as explained in the manuscript, this variation plays an important role in the first stage of speciation inside each cluster (see e.g. Vellend, (2006)). Similarly, stochasticity is known to impact extinction (Melbourne and Hastings, 2008).

To make it more clear that the choice of an individual-based model is well-founded, we have provided the key arguments in the first paragraph in the Appendix 3.

Essential revisions:1) Subsection “Model”: the procedure for the individual-based model is not clear: the text states that "we try 𝑁𝑖𝑛𝑑(𝑡) (number of individuals at that time step) replications of randomly selected individuals. Each selected individual of a strain 𝛼 can replicate with rate r_α…" this is imprecise; what does is it mean that an individual that has already been selected for replication replicates with a rare r_\α? Isn't it rather the case that individuals get selected to produce an offspring with a probability that is proportional to r_α?

The selected individuals do not reproduce deterministically but with a probability *r_↵_*, as is customary in individual-based models. In the new version of the Model section we have clarified the description of the procedure by using the term “replication trial” (third and fourth paragraph of Model section quoted here for convenience; see esp. the respective first sentence):

“Every generation or time step consists of *N^ind^*_(*t*)_ replication trials of randomly selected individuals, followed at the end by a single death step. In the death step all individuals that reached their lifespan at that generation will vanish. Lifespans of individuals are drawn at their births from a Poisson distribution with overall fixed mean lifespan. This is equivalent to an identical per capita death rate for all species. For comparison, simulations with no attributed lifespan (At each replication trial, a randomly selected individual of a strain= 1) were carried out, too. *_↵_*can replicate with probability *r _↵_*. With a fixed probability *µ* the o spring mutates to a new strain *↵*0. Then, the newborn individual is assigned to a randomly selected site. If the site is empty, the new individual will occupy it. If the site is already occupied, the new individual competes with the current holder in a life-or-death struggle. In that case, the surviving individual is determined probabilistically by the “interaction” *I_↵_,*defined for each pair of strains *↵,. I_↵_*is the survival probability of an *↵* individual in a competitive encounter with a. All interactions *↵* form an interaction matrix individual, with that encodes the outcomes of all *I_↵_*2 [0,^1]^ and *I_↵_*+ *I_↵_*^= 1^. *_I_***_I_**_(*t*)_ possible competitive encounters.”

2) Also, it is not clear how the discreteness of time is dealt with exactly. For example, the paper says that each offspring is assigned a randomly selected site, then competes against the occupant of the site and probabilistically takes over. What if one offspring takes over, but then a second offspring is chosen (by chance) to compete for the same site? Is that second offspring then competing against the previous occupant, or against the new occupant, i.e., the offspring that has already taken over from the previous occupant?

This is now clarified in the fourth paragraph of the revised subsection “Model” (second quoted paragraph in previous response).

3) And when do individuals die? Before or after offspring production?

Individuals die after o spring production, see newly formulated third paragraph of subsection “Model” (first quoted paragraph in response to point 1).

4) The authors mention the existence of rock-paper-scissor interactions, but it is not at all clear what the importance of such interactions are for the overall dynamics of the system. Are such interactions very common, essential, just a quirk?

Rock-paper-scissor interactions are synonymous to cycles in the dominance network (see the response to the 6th technical point of the same reviewer). Such cycles are commonly emerging in competitive systems in ITEEM (not in the neutral model) and in nature as essential stabilizers of diversity. In the revised manuscript, the relationship between diversity and cycles is shown in Figure 2D, 2G and Figure 3A, and described in the corresponding text, especially subsection “Generation of diversity” and subsection “Impact of trade-off and lifespan on diversity”. See also the new section in Appendix 8 “Collapses of Diversity” which describes the role of cycles in evolutionary collapses.

5) Subsection “Impact of trade-off and lifespan on diversity”: it is not clear what "average cycle strength" is. Which cycles were used to calculate the average of what?

In the revised manuscript, we have provided a detailed description of the algorithm in the Supplementary material, Appendix 7. Shortly, we compute the average cycle strength in the dominance network as average over a large number of cycles between randomly selected triplets of nodes (=individuals).

6) Subsection “Impact of trade-off and lifespan on diversity”: shouldn't the average interaction strength, i.e. the average values of all I's, be ca. 0.5?

To improve the presentation of the network results, we define in the revised manuscript the dominance networkand detail its computation (subsection “Generation of diversity” and Appendix 7). Briefly, a dominance edge points from an individual *↵* to an individual if *↵* dominates, i.e. if *↵* wins against in an encounter. The weight of the edge is given by the interactions as *I^↵^_I_^↵^*. The mean dominance is now reported also in Figure 3a, instead of average interaction strength. In ITEEM system that starts from monomorphic population, deviation of the mean dominance from 0 and the corresponding mean of neutral model shows that selection generates species with significant pairwise dominance

7) I failed to understand Figure 2. Please explain better.

In the revision we have clarified the terminology (dominance network, see previous response), reworked the Figure (now Figure 3) and its caption, reformulated and extended the corresponding description in the main text (subsection “Impact of trade-off and lifespan on diversity”) and Appendix 6.

8) Subsection “Impact of trade-off and lifespan on diversity”: the Darwinian demon doesn't really make sense: with no diversification, there is only one strain present, and it is impossible to say what kind of competitive ability this strain has, because no other strain is present… so what are you actually trying to say?

We have reformulated the corresponding paragraph of the Results section (“Without trade-o (= 0).…”). To summarize the argument here: We follow in our definition of the Darwinian demon the notion in Law (1979), i.e. a hypothetical organism which can maximize all aspects of fitness simultaneously and would exist if the evolution of species was entirely unconstrained. In our model fitness is composed of reproductive and relative competitive fitness. In the zero trade-o limit, strains with high relative fitness emerge and quickly conquer all resources. If established, a mutant of that type could acquire yet another advantageous variation that would make it superior to its ancestor without any penalty in another trait. Thus, we see in phase I of Figure 3 a sequential dynamics of emergence, invasion, extinction with constantly changing traits. This makes the phase I different from the phase III, in which no diversification occurs and strains remain closely related variants of the same species.

9) I don't find the section about the relationship between tradeoff and productivity very convincing, because it is too vague. I think the authors need to be more specific and detailed here.

We have extended the description of this relationship in the revised manuscript.

10) Subsection "frequency-dependent selection": I think a more mechanistic explanation for differences of speciation and extinction rates compared to Poisson processes is needed. What is the cause of these differences? When is this mechanism at play, and when not? In particular, why do speciation rates increase with the diversity in the system? Shouldn't this only happen at certain intermediate levels of diversity, because when the community reaches a high level diversity, speciation rates presumably decline?

In a system with balanced speciation and extinction rates a null hypothesis is that these events can occur randomly with a constant rate. This happens when ecological interactions and frequency-dependent selection are not the underlying mechanisms for diversification, speciation and extinction. By comparison of distribution of time intervals between branching points with the Poisson distribution we reject this null hypothesis. In ITEEM when one species (as a group of similar traits) emerges in a branching event, it provides a new function to the community and this increases the probability for the generation of further species. The most plausible explanation for the observed deviation from Poisson process is frequency-dependent selection which is mediated by interaction of species and causes autocorrelation between evolutionary events over time. In the new version of the manuscript we have expanded the subsection “Frequency-dependence selection” and also modified Appendix 10 to clarify these points. Please also see the response to the first major concern of reviewer 2 about speciation rate.

Regarding the question about the increase in speciation rate: We did not explicitly report the speciation rate versus diversity and so did not claim that speciation rate increases with diversity. We agree with the reviewer that in the early stage of simulation, speciation rate increases (the positive feedback loop between number of species and speciation that is discussed in the first major concern of reviewer 2), and then will decrease when the number of species in the system saturates to a more or less stable value (Rabosky, 2013). In ITEEM simulations, this saturation depends on system size and mutation rate (Figure 2A and 2D, Appendix 6 (Figures 4 and 5) where diversity increases very fast and then saturates to a constant value). This saturation confirms the reviewer’s point about the decline in speciation rate. See also the answer to the first concern of reviewer 2.

– I would like to bring the authors' attention to the recent paper by Doebeli and Ispolatov, (2017), which also presents models for community assembly and evolution of diversity are presented, with some parallels to the work discussed here. These parallels could be taken up in the Discussion section.

We are grateful for this reference and have included it in our discussion.

Reviewer #2:The authors present a model for studying diversification dynamics as an outcome of, as they claim, competition for a single resource. I have substantial concerns regarding the insight that could be gained from this work, mainly due to the type of model and the lack of a mechanistic interpretation.1) Subsection “Model” and subsection “Frequent-dependent selection”: Your trait space is Nsp-dimensional, where Nsp is the number of extant strains. In particular, the trait space grows explicitly with every strain added. Is it surprising that you get coexistence of Nsp strains in an Nsp-dimensional trait space? Further, is it surprising that "emergence of new species increases the probability for generation of further species"? It seems to me that this is quite expected, given your model. This also explains why you get these mass extinction events; there's a positive feedback loop between speciation rate and species number that is easy to recognize even without simulations.

This is a complex comment that we split up into three questions: (1) Is the co-existence of *N_sp_*species in *N_sp_*dimensional trait space surprising? (2) Is it surprising that emergence of new species increases probabilities for generation of further species? (3) Is there a positive feedback loop between speciation rate and species that explains mass extinctions?

The first question may have been arisen because of a misunderstanding of the term trait. We have to emphasize that in our model the term trait does not carry the usual meaning of a phenotypic trait (wikipedia: “A phenotypic trait is an obvious, observable, and measurable trait; it is the expression of genes in an observable way”), but that it is an interaction trait. Even in a system of n strains that are distinct with only one phenotypic trait, each strain has n interaction traits, quantifying its interactions with all strains in the system. In ITEEM we only consider interaction traits and are agnostic about the number of phenotypic traits. Thus, the underlying eco evolutionary system could well have only a few phenotypic traits, and we consider it surprising to see under these conditions the emergence of stable diversity (As it has been shown that a simple GLV model without further assumptions fails to produce diversity. See the fourth paragraph of Discussion section).

However, a crucial point is that even in a system with an *N^st^*dimensional trait vector, diversity only emerges if we account for physico-chemical constraints that appear in ITEEM as trade-o between replication rate and competitive ability. Depending on the trade-o strength, the diversity can be low or high, stable or unstable (see the phase diagram in Figure 3 of manuscript), although the trait vector has always more or less the same dimension *N_st_*. As we emphasized in the manuscript (section Generation of diversity) diversity is not measured as the number of strainsbut as the number of species, i.e. coexisting clusters of strains that emerge via diversification.

We hope that the new version of the manuscript clarifies these points (see Materials and methods section and the first paragraph of Discussion section.)

The second question of this comment by reviewer 2 was about whether it is surprising that new species increase the probability for further species. This question might have been arisen by a confusion between strain and species, possibly because of the unfortunate symbol *N_sp_*for the dimension of the interaction trait vector *T_↵_*. In the new version we have replaced this symbol by *N_st_*, clarifying that the dimension of *T_↵_*is given by the number of strains (not the number of species). We hope that the more clear nomenclature, especially the use of *N_st_*instead of *N_sp_*introduced above should sharpen the distinction between species and strain that we make, in contrast to other works. Strains emerge under certain conditions (trade-o regimes) continuously as individuals successfully replicate with mutation, while speciation events are more rare and require a branching of the phylogenetic tree and diversification in trait space. Given our model it is indeed “quite expected” that the number of strains *N_st_*initially increases in the almost empty system, and that we then have “a positive feedback loop between speciation (emergence of strains) and species (strain) number”. But this early stage comes quickly (after less than as the system saturates with individuals. The number of⇠ 2 ⇥ 10 strains^5^ generations) to an end is then more or less stable for long times. Likewise, the number of speciessaturates at values that depend on trade-off, system size, and mutation rate (see Figure 2D and Appendix 6-1).

Our analysis focuses not on the initial stage, but on the stage after onset of saturation when we have a balance between speciation and extinction. In such a system, the null hypothesis is that speciation and extinction occur by random drift. However, in the manuscript we can quantitatively reject this null hypothesis. Instead, our analysis demonstrates that in ITEEM the emergence of new species (cluster with similar traits) by branching phenomena provides a new function to the community, which increases the probability for the generation of further species (Herron and Doebeli, 2013). We hope that the new subsection “Frequency-dependent selection” and Appendix 10 provide a better explanation than the first version of the manuscript.

For the mass extinctions, the third point in the comment, similar arguments apply: The phenomenon is occasionally observed in the saturation phase, and it is not just a trivial consequence of the basic model but depends on trade-o strength and other parameters. To better explain this complex phenomenon we have improved the analysis and presentation of mass extinctions, especially in subsection “Generation of diversity”and the new Appendix 8.

Please also see the answer to the tenth technical point of reviewer 1 about speciation rate and the response to the third concern of reviewer 3 about mass extinction.

2) Subsection “Generation of Diversity”: Please explain how you define "functional diversity" in your model at first mention. Appendix 6 was uninformative. The only information I found was in the caption of Figure 1, where you define functional diversity "in terms of the size of minimum spanning tree (SMST) in trait space". But your trait space is continuous ([0,1]^Nsp) so a priori there is no tree structure connecting species (apart from phylogeny). So how do you define a spanning tree, and how is this spanning tree not a function purely of the number of strains?

We have added in the revised manuscript in caption of Figure 2d and in the text referencing this Figure (end of the subsection “Generation of diversity”) as a short explanation that the functional diversity (FD) quantifies the spread of the population in interaction trait space. On both occasions we refer the reader to Appendix 6. The revised and expanded Appendix 6 covers evaluation of FD including further explanations and references.

On the minimum spanning tree (MST): The Reviewer gives a good example for a tree structure in a continuous space induced by a distance structure on species, namely phylogenetic trees. In practice, phylogenetic trees are often inferred from estimated evolutionary distances (i.e. continuous quantities) between pairs of all considered taxa, assuming minimum evolution (see e.g. Saitou and Nei, (1987); Felsenstein, (1997)). In our manuscript, the MST is computed in an analogous way. Here we have between each pair of strains a distance in interaction trait space, and from the set of these distances we infer a minimum spanning tree (MST). The MST is a tree that links all strains and has a minimum sum of edges (sum of edges of MST = SMST). For the computation of the SMST see Appendix 5.

The SMST quantifies the spread of the total population of strains in interaction trait space, i.e. the space that matters for function in the community. To us it was therefore the most natural way of expressing functional diversity. As we detail in Appendix 6-3 there are a number of FD measures used in the literature that we have computed (Mouchet et al., 2010). However, all FD measures show the same strong dependency of trade-off (Figure 6 of Appendix) and are not simply reflecting the number of strains, to answer the last question of the reviewer. For example in Figure 1 of the Appendix, we compare a set of strains evolved by neutral evolution (no difference in the interactions and no trade-o regime) with a set of strains at an intermediate trade-off, both with the same number of strains. The former set is compact (low FD) while the latter is dispersed in interaction trait space (high FD, note the different scaling).

3) Subsection “Generation of Diversity”: Related to the previous point. What is a "functional niche" in your model? Since you did not discuss mechanisms underlying the interaction matrix, it is not clear what a function is.

In ITEEM the function of a strain is its interaction trait vector, describing its interaction with each member of the community. A functional niche is a cluster of strains in the space of interaction trait vectors. This definition is given in subsection “Generation of diversity” (paragraph ITEEM systems self-organize toward structured communities), and it is in line with the notion of functional niche developed in the literature (Elton, 1927; Clarke, 1954; Whittaker et al., 1973; Rosenfeld, 2002; Odum, 1959) as detailed in the new version of Appendix 6-2, subsection “Functional diversity, functional group and functional niche”.

4) Subsection “Generation of Diversity”: Saying your model has no geographical isolation nor resource partitioning sounds meaningless. You have not specified the underlying mechanisms for your "interaction trait", so it is hard to draw a comparison to other more mechanistic models (i.e. where the physiological/metabolic traits are explicit).

As we explain in the revised subsection “Model”, the system can be considered well mixed so that no individual is assigned to a specific spatial location and each individual accesses the same resources. In this respect ITEEM is not different from referenced interaction-based models, the Lotka-Volterra competition equation, or the replicator equation.

5) Subsection “Model”: Since replication is non-sexual, what is the purpose of defining "species" separately from "strain" in your model? In Appendix-I you mention that you define "species" operationally based on divergence time and using some arbitrary cutoff threshold, so this sounds a bit analogous to operational taxonomic units (OTUs) in microbiology. However, you model mutation/selection/growth dynamics at the strain level. Please explain early on what additional insight one may get from counting "species".

In ITEEM we observe reliably the emergence of clusters of strains of similar interaction traits, and we find that these clusters play important roles in ecoevolutionary dynamics. Thus, it seemed natural to us to introduce the term species to simplify the text and to offer a biological interpretation. We follow there the common usage of species in models of asexual populations as a term for separated clusters of very similar genotypes or phenotypes, depending on the model, see e.g. Dieckmann and Doebeli, (1999); Bonsall et al., (2004); Sevim and Rikvold, (2005); Scheffer and van Nes, (2006); Mathiesen et al., (2011); Ispolatov et al., (2016). See also the expanded Appendix 1.

6) Discussion section: You claim that modeling in terms of interaction traits (which in your case means in terms of outcome probabilities of local competitive exclusion) "coarse-grains these complex systems in a natural, biologically meaningful way.". But many of your interpretations and comparisons to other models or data require translating your Abstract "interaction traits" to functions or life histories. You did not discuss at all what real traits could possibly give rise to your interaction trait matrix. Your interaction matrix seems to loosely represent competitive interactions; but then it can only explain diversity within a single trophic level (e.g. in the case of animals) or a single metabolic niche (e.g. in the case of bacteria); yet, you keep referring to "functional diversity".

In our coarse-grained model the interaction trait of a strain summarizes all negative effects of the competing populations on the population of that strain, no matter whether these strains compete for the common resources by interference or by exploitation, and, in this sense, our interaction trait matrix is similar to the community matrix of the well-established Generalized Lotka-Volterra equation (see penultimate paragraph of revised Model section). Specifically, the coarse-grained interaction trait summarizes over all phenotype traits, abilities and mechanisms that increase the chance of organism to take over resources, help the species to tolerate reduction in contested resource availability (Aarssen, 1984), or prevent competitors from gaining resources (Gill, 1974). The current version of Appendix 3 takes up this point again.

On the last point of the comment, the use of “functional diversity”: We agree with the reviewer that the presence of different trophic levels implies functional diversity. However, the reverse conclusion is not necessarily true if we understand function as the role in a community, not limited to the mode of resource consumption. We follow here this broader interpretation of function (see e.g. Petchey and Gaston, (2006)) that, for instance, considers interaction traits. If we accept this broader interpretation of function, functional diversity can be measured as spread in interaction trait space (see also responses to comments 2 and 3 of this reviewer).

7) Discussion section: Related to the previous comment. You say that your formulation was "chosen to reflect reasonable properties" and that you "have assumed a single, limiting resource in a well-mixed system". However, you did not provide any plausible mechanism for how such an interaction matrix could arise from competition for a single resource in a well-mixed system. This is actually a big question: how can one obtain a Nsp-dimensional trait space through competition for a single resource pool?

The comment of the reviewer implies that we are modeling a competition over the consumption of limited resources. However, the situation modeled here is that of a competition over one common, renewable resource pool (see section Model). There is a long history of models for this situation, for instance the Lotka-Volterra equation (Volterra, 1928) and interaction-based eco-evolutionary models (Ginzburg et al., 1988); in our case, we are using the Lotka-Volterra approach, which models both exploitative and interference competition. In these models, an interaction matrix quantifies how competitive organisms influence, by any means, *N_st_*_⇥_*N_st_*each other’s populations in pairwise dominance relations. The final question by the reviewer is closely related to his/her first comment, and we refer the reviewer to our response there.

Reviewer #3:The manuscript "Trade-off shapes diversity in eco-evolutionary dynamics" presents a quite original evolutionary-ecological individual-based model based on a compromise between growth rate and competitiveness. The model is simple but exhibits quite rich evolutionary properties, which, while being not entirely unpredictable, are intriguing and provide useful insights and generalizations. This combination of simplicity of the rules and relevance of the dynamics usually distinguishes successful and long-living models from the rest and makes them understandable and appealing to a wide audience. Besides, I cannot see any potentially "hidden" flaws in the model and interpretation of results that could cast doubts on the main conclusions. Thus, in my opinion, the manuscript may be published in eLife after the following and, perhaps, other comments are addressed:1) The complexity of algebra in (3) and subsequent definition of s is definitely unwarranted by an otherwise quite accessible level of math in the rest of the paper. Furthermore, it apparently confuses even the Authors when they classify the domains of weak and strong trade-offs:It looks like the definitions of high-tradeoff (\δ ~ 1) and intermediate trade-off (\δ ~ 1/2) are misleading. The strongest dependence of r on C appears to be when \δ=1/2, which also follows from Figure 1. The phrase "In high-trade-off phase III, any small change in C changes r drastically" is simply wrong for all but very small C. I would call both the \δ=1 and \δ=0 limits as small tradeoffs as they look perfectly symmetric, or, even better, choose another, more heuristic and transparent, form of parametrization of r vs. C.

We are grateful to the reviewer for pointing out that the description of the trade-o needs clarification. To this end we have moved Eq (3) from section Model to a revised Appendix 2-1, and, instead, added to subsection Model the new Figure 1, which depicts actual shapes of the trade-o function in a more accessible way. We have also made the behavior of the trade-o function and the meaning of more transparent (subsection “Model”).

The new section Appendix 2-3 should now make it more clear why the trade-o

increases from = 0 to = 1. To summarize the main point here: The strength of the trade-o is not simply given by the slope of the ∆rr/∆CC=drdC.Cr trade-off curve, but the computation of the actual strength of the trade-o requires consideration of the community context. Specifically, the trade-o strength can be approximated by, with the replication rate *r* and competitive ability *C* of the current community, slope drdC of the trade-o curve, and perturbations of reproduction rate *r* and competitive ability *C* due to emergence of a new mutant. This trade-o strength increases with trade-o parameter (Appendix 2-3).

2) Would it be possible to say anything about population dynamics of "species"? Is it cyclic?

We have looked for cyclic population dynamics by Fourier analysis but could not see clear signals. One problem may be our limited time resolution (10^4^ generations as sampling interval) so that we do not see short cycles, but the dynamics may also be chaotic because we have many interacting and evolving species.

3) At least qualitatively, what are mechanisms of mass extinction?

In the revised manuscript we sketch a mechanism possibly underlying mass extinctions in Results, subsection “Generation of diversity” (paragraph “Formation of strong cycles…”). Appendix 8 provides more details.

A problem for this part of the study was that we see mass extinctions only occasionally so that we do not have a strong statistics. Moreover, we have a limited time resolution of 10^4^ generations. We do not see conspicuous diversity values preceding mass extinctions. However, one remarkable pattern that occurred prior to all observed mass extinctions was high cycle strength (Appendix 8, Figure 7B). It could be that new species emerge or species go extinct, resulting in conflicting forces in the strong network that discharge in a collapse, similar to a crystal defect leading to geometrical frustration and destabilization of a highly ordered crystal lattice.

4) An explicit plot of diversity vs. system size (perhaps just for the δ optimal for diversity) and, ideally, an estimate of corresponding scaling, would be very revealing.

Appendix 9 provides a set of plots of diversity vs. system size for different trade-off-parameters. The new Figure 9 in Appendix 9 shows the scaling of diversity with size for middle values of.

5) Similarly, how the level of diversity and the typical number of traits in a species depends on the mutation amplitude m?

We have added a description of the effect of variation in *m* to the Results, subsection “Effect of mutation on diversity”. Briefly, the smaller *m*, the slower the dynamics and the smaller the extension of individual species (= clusters of strains) in trait space. Large *m* leading to big trait changes suppress the establishing of diverging evolution and selection

6) For a general reader of eLife, the MDS algorithm needs to be explained and properly referenced. It plays a major role in interpreting the results, however, is presented only by the name of the corresponding function in R.

Explanations of MDS in Appendix 4 and the caption of Figure 2 (previously Figure 1) of the main text have been expanded. However, MDS is only used for visualization. All quantitative analyses are based on evaluations of the original high dimensional trait-space.

7) A qualitative explanation about the minimal spanning tree analysis would help as well.

The explanations on the minimum spanning tree analysis in Appendix 5 have been expanded.

Reviewer #4:I've carefully read the paper "Trade-off shapes diversity in eco-evolutionary dynamics" by Farnoush Farahpour and colleagues. In it, they use an individual-based eco-evolutionary model to understand the emergence of community structure based on evolving interactions. The model produces some interesting patterns, such as clustering in trait-space and the evolution of intransitive competitive loops.There are a number of aspects of this paper that I liked: the focus on evolution of interactions, the emergence of intransitive loops, the dimension reduction applied to the vector-valued traits, and the comparison with a neutral model variant. These are all creative contributions to the modeling literature.While I was intrigued by the idea of modeling the evolution of interactions directly, it was hard for me to connect it to real ecological systems. The interactions between species are not determined by phenotypic traits of the organisms but evolve independently. It's based on an unstated assumption that species interactions are totally idiosyncratic and unpredictable. I feel that evolution in this model is too unconstrained, despite the trade-off between competitive ability and reproduction, resulting in the prevalence of intransitive loops. As a complex-systems researcher, I'm fascinated, but as an evolutionary ecologist, I'm skeptical.The authors need to do a better job putting their work in the context of the extensive literature on eco-evolutionary dynamics. The text had many statements that had me scratching my head. Examples:1) Framing the problem in terms of resources, but in the model there are no actual resources.

There is one durable resource or resource pool in the model for which individuals compete (see the first paragraph of Model section and the last paragraph of Appendix 3). In this respect ITEEM is very similar to established models in ecology and evolution that are based on Lotka-Volterra equations (Gill, 1974; Masel, 2014) which model exploitative and interference competition for durable resources.

Note that these models are different from the consumer-resource model, which models indirect competition for consumable resources, and from the logistic equation, which models exploitative competition for durable resources.

2) The competitive exclusion principle only holds at equilibrium (Armstrong, Levins) and must count resources plus shared predators (Levin, 1970).

We agree with the reviewer that the principle holds only at equilibrium although it is still referred to in non-equilibrium systems and even in the presence of disturbance, adaptation and evolution (Posfai et al., 2017; Germerodt et al., 2016; Kinnersley et al., 2009; Pfei er and Bonhoeffer, 2004). We had referenced the principle as a contribution to the historical context of the diversity debate. But since it might lead to confusion and because it is not the concern of the manuscript, we have removed the reference from the Introduction.

3) The "eco-evolutionary models" cited in the Introduction seem to be just ecological models.

In the revised manuscript we have rephrased the paragraph accordingly.

4) In the Introduction, the "observed eco-evolutionary dynamics" cited that the model "closely resembles" aren't empirical patterns observed in real systems, but just results of other models.

The references are mixed empirical (see for example Maynard et al., (2017); Kvitek and Sherlock, (2013); Bolnick and Fitzpatrick, (2007); Coyne, (2007); Herron and Doebeli, (2013)) and theoretical of well-described evolutionary and ecological phenomena such as sympatric speciation, emergence of two or more levels of differentiation similar to phylogenetic structures, large and complex biodiversity over long times, evolutionary collapses and extinctions and emergence of cycles.

5) Discussion of speciation overlooks that the species here are clonal, so what's hard about speciation?

Speciation would indeed be trivial if it was the generation of new (clonal) *strains*. However, speciesare not clonal (see e.g. subsection “Model” or Figure 2). The model continuously produces new strains, but the diversification and branching of species, i.e. well-separated non-transient clusters in trait space, are rare and heavily depends on trade-off.

6) Also, some of the sentences throughout were hard to understand the meaning of. E.g., "Evolutionary changes at the genetic level influence ecology if they cause phenotypic variations that affect biotic or abiotic interactions of species which in turn changes the species composition and occasionally forces species to evolve their strategies."

We have split this sentence up to make it more intelligible.

7) Some of the details of the model implementation weren't clear. For example:a) How exactly do births happen (subsection “Model”)?

This has been clarified in the revised manuscript. See also response to 1st and 2nd technical point of reviewer 1.

b) Is mu a mutation "rate" or probability of mutation during a replication event (subsection “Model”)?

This has been clarified in the revised manuscript. See also response to 1st technical point of reviewer 1.

c) Why is lifespan drawn from a Poisson distribution (Subsection “Model”) and how can that be infinite (Figure 2)?

We used the Poisson distribution as a simple positive valued probability distribution that for large means approaches a normal distribution.

In the case of an infinite lifespan, no lifespan is attributed to individuals. In that case the only cause of death is defeat in a competitive encounter. The corresponding sentence in the subsection “Model” has been expanded for clarification.

If each individual stays in a site, is it really well-mixed?

Yes, because there is no neighborhood relation between sites. A site is just a portion of the durable resources, sufficient for the sustenance of an individual. The property of being well-mixed means that arbitrary encounters between new individuals and sites happen with equal probabilities. The subsection “Model” and Appendix 3 have been modified to clarify these points.

Does mutation of one species' interaction coefficients end up changing another species' reproduction rate through the trade-off (2)?

Indirectly, yes. Adding a mutant strain introduces a new interaction trait and thus in general influences the competitive ability *C*. A change in the competitive ability will then change reproduction rate *r* through the trade-off. This is qualitatively similar to reproductive plasticity observed in natural systems (Claridge and Franklin, 2002; Buffi et al.,, 2013; Goldstein et al.,, 2016). However, when ITEEM systems reach high diversity (i.e. complex interaction trait vectors), this effect is usually small.

Could you not get at the same questions more efficiently using deterministic Lotka-Volterra dynamics?

Please see the response to the point 2 of reviewer 1.

[Editors' note: the author responses to the re-review follow.]The reviewers agree that substantial effort was put into the revision. However, there are a number of remaining issues that call for a further in-depth revision. Reviewers 1, 2 and 3 (who was previously reviewer 4) still have substantial concerns about aspects of the paper. These concerns are all related to issues raised in the initial reports and will need to be addressed in a constructive manner if the paper is to be published in eLife.Among the issues to be addressed are:1) Individual-based vs deterministic systems: both reviewer 1 and 4 (formerly reviewer 3) point out that it is not at all clear that these different approaches would yield different results. In other words, it is not clear that salient results reported in the paper depend crucially on the presence of noise, as appears to be the contention of the authors. This needs to be explored at least to some extent. This issue cannot be dealt with by simply "citing it away".

To avoid misunderstandings, we would like to state our notion of the terminology used in the referee report and in our response below. In the referee reports, four non-synonymous terms have been used to denote alternative eco-evolutionary approaches that should or could be compared to ITEEM: (1) deterministic models, (2) continuum approaches or continuous models, (3) population models, and (4) mean field models.

The fourth term, mean-field models, is used with different meanings in the comments of reviewers 1 and 4. In the response to the comment 5 of the fourth reviewer we explain our understanding of the different usages of the term “mean-field” in statistical mechanics and in population biology. To avoid misunderstandings, we have replaced in the manuscript the term “mean-field” with “population-level”, a term that should be more clear for the target audience.

Thus, we are left with three non-synonymous terms used by editor and reviewers. We hope that the reviewing team agrees with our understanding of the terms:

1) Deterministic model: A mathematical model that unfolds completely deterministically, in contrast to models that make use of randomness.

2) Continuous model (in time): A mathematical model with variables changing continuously in time without any abrupt transition between discrete states, in contrast to models with discrete time steps.

3) Population model: A mathematical model that studies dynamics of populations and their interactions. Such models often describe populations as continuous quantities, in contrast to models with discrete individuals and individual interactions.

The editor takes up comments by reviewers 1 and 4 on the comparison of ITEEM with deterministic/continuous/population/mean-field models. Since both reviewers solely refer to Shtilerman et al., (2015) as reference for this latter class of models, we assume that they propose a comparison of ITEEM with the model by Shtilerman et al., (2015), or equivalent models. Note that the model by Shtilerman et al., (2015) is a hybrid with stochastic and deterministic elements that are applied iteratively: each stochastic evolutionary “kick” is followed by an ecological relaxation of fixed length in which the deterministic Lotka-Volterra population dynamics is integrated. This “kick-relax” model as a whole is neither deterministic, nor continuous in time, nor continuous in population.

One reason suggested by the reviewers for a comparison of such a kick-relax model to ITEEM is that the former would speed up the simulations, thus enabling to get cleaner results with larger systems. We have therefore developed a kick-relax model that is as close as possible to ITEEM and compared its output to the ITEEM output. We have further extended subsection “Interaction-based models” on this topic.

To not water down by technical details the main point of the manuscript, i.e. the effect of trade-off on eco-evolutionary dynamics, we provide the following detailed information here in our response, but we have not included it in the manuscript.

Tests with a kick-relax code show that the method is much less efficient than ITEEM (though with a caveat, see below). At the same system size of 10^5^, i.e. number of sites in ITEEM and carrying capacity in the kick-relax model, mutation rate 0.001 and lifespan 10^5^, we achieve 15×10^4^ generations with ITEEM and 200 generations with the kick-relax code per CPU hour. The main reason is that the integration of a Lotka-Volterra system of hundreds or thousands of strain equations is computationally far more demanding than the simple operations on an array of individuals in ITEEM. Because of these computational problems we could not collect as many generations for kick-relax as for ITEEM: with kick-relax we sampled at substantial computational cost about 6 × 10^5^ generations for each of 9 simulations (9 trade-off parameters and one life-span); with ITEEM we sampled 5 × 10^6^ generations for each of 540 simulations (3 runs for 18 trade-off parameters and 10 life-spans). Our kick-relax code is an R-script with the CPU time critical integration step performed by a fast FORTRAN routine. We have made the kick-relax and ITEEM source codes freely available (see the end of subsection “Model” of the manuscript).

The kick-relax model is also very sensitive to the length of the relax interval, i.e. the time interval between two successive kicks. To obtain results close to the nonequilibrium model of ITEEM, this time interval must be small which makes the simulation slow. Increasing this time interval 10-fold abolishes the adaptive evolutionary dynamics and gives rise to communities with no speciation and diversity.

The caveat mentioned above is that a clean comparison in terms of efficiency requires that the results are the same. However, the results are qualitatively different. One important biological result of ITEEM is the phase diagram of diversity as function of trade-off 𝛿, and the humpback shaped cross-section through that phase diagram (Figure 4 of the current version of the manuscript and Figure 8 of Appendix). The Figure below opposes this humpback curve, in some diversity indices, with the corresponding curve obtained with the kick-relax model.

The reason for the discrepancy lies in the interface of the kick and relax stages: We have to decide in the evolutionary kick stage what happens to which population, how and when we introduce a mutant, and when we consider a population extinct in the ecological relaxation stage and remove it from the pool for evolution. This interface is not motivated by biology but it is a consequence of the technical realization of the model and therefore necessarily artificial. In our kick-relax simulations we set the extinction threshold to 1. In the kick stage, the number of new mutants that each extant strain 𝛼 could produce during the relax interval is calculated from the LV equations as

⌊" close="⌋" separators="|">μrαnα1-∑βxβ+∑βrαxαIαβxβ,

where the symbols have their usual meaning and the outermost brackets indicate the “floor” function, i.e. we take the next smaller integer number below the value of the term enclosed by these brackets. Then these mutants are added to the community, each with an initial population of 1.

In the early stages of eco-evolutionary simulations, ITEEM and kick-relax model differ due to individuality of the agents. With the large system size and the negligible competitive dominance of a mutant to its parent, a mutant can invade very slowly. In kick-relax, it then can produce a new mutant of its own if its population reaches a large value (typically around 2000 in our simulations) to be able to compensate the small product of mutation rate and competitive ability. Taken together, this means that an adaptive character displacement takes a long time. However, during this time the parent produces numerous novel mutants. Thus, we have a slow and more or less even spread in trait space, effectively without branching. In ITEEM, on the other hand, an individual mutant has a non-negligible chance (proportional to its reproduction rate and competitive ability) to double its population in one generation, and in the process, to produce a mutant of its own. Hence, from the early stages of the simulation onwards, adaptive forces are active.

Shtilerman et al., solve the above problem technically with an arbitrary assumption: they switch a certain percentage (e.g. 5%) of a mother strain population selected for mutation in one step to a daughter strain which is different from its parent by an arbitrary overlap reduction parameter ℎ introduced to enforce niche separation. The artificial character of the kick-relax model may also have forced Shtilerman et al., to make further assumptions, leading to more parameters with seemingly arbitrary values, such as initial population of a mutant (explained above), an extinction threshold 𝑛_0_ below which a species is removed forever, and a relax time interval 𝑇_𝑠_between stochastic kicks.

Considering that we are trying to understand features of a complex non-equilibrium eco-evolutionary dynamics, it is not surprising that models like kick-relax that implement a number of technically motivated artificial assumptions concerning exactly this non-equilibrium dynamics will in general yield different features of this dynamics than a model that does not make these assumptions.

In contrast to the kick-relax model, the nature of ITEEM requires only a few elementary assumptions. It imposes no separation of ecological and evolutionary steps or time scales. In this conceptual sense it is more continuous than kick-relax models.

The new short paragraph added to the Discussion section explains briefly that why we adopt the individual-based approach.

2) The assumption of independent evolution of the elements of the interaction matrix needs to be discussed in more detail and clarity, and in the context of biological realism. This point was raised by both reviewers 1 and 3 (formerly reviewer 4).

To respond to the first comments of reviewers 2 and 3 (we guess “1 and 3” was a typo), we have added now introductory context, and a description and discussion of the mapping from phenotype to interaction. Moreover, we have added a substantial new Appendix-13 (“Phenotype-interaction map”) where we explain how a random variation in phenotype space yields a random variation in interaction space. However, we emphasize that despite the randomness of interaction term variations at the timepoint of their Introduction (in accordance with random variations of phenotypic traits), the evolutionary dynamics in ITEEM is adaptive, and the fate of mutants are determined by frequency-dependent selection (in accordance with phenotype-based models like adaptive dynamics).

3) The "single resource" issue is related to point 2. and also needs a revised treatment.

We have taken precautions to not misdirect readers into thinking ITEEM was a consumer-resource model. Please see the response to the reviewer's comment.

4) I am sympathetic with reviewer 3's (formerly reviewer 4) concern about some of the references to previous work, particularly with regard to tradeoffs.

We have exchanged or dropped references accordingly. Please see the response to the corresponding comments of the third reviewer.

5) Please also address the remaining points, e.g. the definition of "species" raised by reviewer 2, as well as other minor points.

Please find below our responses to all comments of the reviewers.

Reviewer #1:The authors did a very good and thorough job in revising their paper. Almost all of my concerns have been addressed satisfactorily. The one remaining issue is that I don't buy the authors case for only using individual-based simulations (my original comment 2). The authors state in their rebuttal that "It has been extensively discussed in the literature that continuum approaches are unsuitable in cases of non-equilibrium dynamics,…", and they somehow conclude from this that the deterministic Lotka-Volterra description (which in my mind is the same as a "logistic" description) would not be appropriate for the problem at hand. But that is exactly the question: in some general sense, one would expect that in the limit of large population sizes, the individual-based model used by the authors would converge to some deterministic model, and my guess is that this model would be at least close to the "mean-field" Lotka-Volterra model. It then becomes important to understand just which features of the individual-based models can be understood by studying the much simpler mean-field model. The authors present no arguments why there even are *any* features of their individual-based model that could not be observed in the deterministic model. They refer to near-neutrality, but that's exactly one of the features that was observed in similar deterministic Lotka-Volterra models by Shtilerman et al., (2015). I am not convinced that the salient results reported in this paper could not also be obtained with deterministic models. Just claiming that some results cannot be obtained in that way on general grounds does not make it true in this particular case. The obvious advantage of using deterministic models would be that such models are much more tractable analytically (e.g. from a statistical physics point of view), and it is therefore important to know how far one can get using them. I think it would not be too onerous to at least do some tests using the deterministic models to either confirm or refute the claim that they can produce similar results as the individual-based models. The question is: does stochasticity really play a major role in producing the results reported in this paper? If so, then this would be important to know (but this point would need to be made based on more than just a vague statement that their "system falls into the category of those better modeled by individual- based models"). If not, then the deterministic models should do a good job reproducing these results.

As described in the response to the editor, the model of Shtilerman et al., (2015) is not a deterministic model but a kick-relax model with periodic stochastic evolutionary kicks followed by deterministic population relaxations. All previous interaction-based models including that of Shtilerman et al., (2015) are of this kick-relax type. We presume that the reasons for reviewer #2 the adoption of kick-relax models were historical: the scientists in the field were familiar with Lotka-Volterra continuous models for ecological systems, and they transferred them therefore to eco-evolutionary problems, adding just periodic stochastic kicks for the evolutionary part. Not only do such kick-relax models artificially disentangle evolution and ecology and therefore are of questionable value for studying dynamics, but the artificial split necessitates a number of additional assumptions and parameters. This includes for instance the discrete extinction threshold (value not reported), and length of the relax interval (i.e. interval between two stochastic kicks) of 2000 generations. Another value that is difficult to justify is the low mutation rate of about 10^−8^ per replication. Diversification is imposed ad hoc by a discontinuous flipping of 5% of a parent population into a mutant population with artificial niche separation, described by another ad hoc parameter ℎ. The arguments given in the introduction of Shtilerman et al., (2015) explains the necessity of such discontinuous branching in the model to enforce diversity. It is noteworthy that most of these technically motivated parameters in kick-relax models do not correspond to observables. Conversely, in ITEEM the number of parameters is small, they correspond to observables, the model does not artificially split the dynamics into alternating evolutionary and ecological stages, and diversity is emerging in a simple, natural process as a function of a trade-off that links eco-evolutionary dynamics to physical constraints.

As described in the response to the editor's first comment, we have nevertheless developed our own kick-relax model to match ITEEM as closely as possible. It is much less efficient than ITEEM and the results are qualitatively different (see response to editor's comment above). Given the artificial character of the kick-relax protocol, we find the latter not surprising.

The kick-relax model is also not suitable to answer the question about the role of stochasticity because in the kick step speciation events or mutations are introduced randomly. However, as described in the response to the editor, the discrepancy of some diversity indices between kick-relax model and ITEEM can be explained by the individuality of the agents and stochasticity of population dynamics that allow for faster adaptive character displacement in the early stage of the simulations.

Reviewer #2:The authors have partly addressed my concerns and those of the other reviewers. I do however still have two major concerns:1) I agree that a low-dimensional phenotype space (e.g. pertinent to exploitation of/competition for a single resource) can give rise to an Nst x Nst interaction matrix that encodes the competitive interaction between strains. However, the crucial assumption of the authors is that each term in this matrix (well, half of the terms) can vary independently. How this could come about in reality is unclear to me.In other words, if "P" is the underlying phenotype space (solely related to consumption of/competition for the common resource) and "I" is the space of possible interaction matrices, what could the mapping f:P->I possibly look like, such that f(P) is an Nst x Nst/2 dimensional manifold?I strongly recommend:(a) Avoid any comparison to "single resource" models or real systems.(b) Acknowledge early on that an important assumption of the model is that the terms of the interaction matrix (well, half of them) can in principle vary independently (i.e. are not constrained explicitly due to genetics or ecology). Whether this assumption is met in reality is an open question.(c) Clarify in the discussion that this paper does not address the important question of how such a high-dimensional interaction trait space (i.e. with Nst x Nst/2 independent axes) might arise, or provide a plausible example.

On recommendation (a): In the revision, we avoid the expressions “single” or “limiting” resource and it is clarified that ITEEM is not a consumer-resource model with a single limiting resource (Discussion section). With ITEEM we provide a minimalist framework for eco-evolutionary dynamics that shows the qualitative behavior of evolutionary biological systems. On (b): Variations of the interaction matrix appear at random but only a few survive the frequency-dependent selection that emerges in the system. This is akin to the principle observed by Luria and Delbrück for genotypes (Luria and Delbrück, 1943): mutations appear at random but only a few are selected. These random variations generate mutants that are ecologically similar to their parent, i.e. close to the parent trait in trait space. Please see the Appendix-13on the random variations of interaction terms that are mapped from random variations in phenotypic traits. Please also see the response to the second point summarized by the reviewer. On (c): As described in the response to the second comment by the editor, we have added substantial explanations concerning the mapping of phenotype space to interaction space, including an extended example in Appendix-13.

2) The definition of "species" by the authors is still confusing and of questionable relevance. The authors define "species" operationally based on a cutoff threshold in phylogenetic distance. While this is common practice in microbial ecology (where such clusters are called Operational Taxonomic Units), few would claim that the emergence and disappearance of OTUs is comparable to "speciation" dynamics in sexually reproducing organisms.What I also found confusing is that in their "response to reviewers", the authors explain that "species" are "well-separated non-transient clusters in trait space". This does not align with the definition provided in their manuscript (Appendix 1), where species are defined as "clusters of strains separated by long-lasting gaps in a phylogenetic tree". Are these definitions equivalent in your model?While the emergence of clusters in trait space is indeed interesting, I would recommend not calling these clusters "species", since clusters in trait space need not always be monophyletic and could in principle also consist of distantly related strains that happen to have converged in trait space.

We have revised the manuscript, especially Appendix-1 (see also subsection Model), to clarify the term species. Shortly, we follow the widely-used phylo-phenetic species definition of Rosselló Mora, (2001), i.e. species are monophyletic clusters in trait space. This definition is biologically meaningful and applies also to asexual populations. To implement the species definition we identified long-lasting mono-phyletic clusters of strains, i.e. strains that have the same parent (as inferred from the genealogical tree), and that have been separated from this parent in trait space for a long time (see Figures 3A and 3C of the current version and the new figure in Appendix-1 for examples).

Reviewer #3:This is reviewer 4 from the original submission again. This remains an interesting yet frustrating manuscript. The authors resisted many of the good suggestions from the other reviewers and myself in how they can place their manuscript in the broader context. In the end, it's the authors' manuscript, but I still think they could do a better job in the introduction and discussion to not confuse potential readers.To me, the most interesting part of the manuscript is the idea that species interactions might be so high dimensional that it is best to focus on interaction traits that summarize many idiosyncratic phenotypes. This is described in the discussion but should also be highlighted more in the introduction. The relationship between phenotypic traits and interaction traits should be clarified to better address comments 1 and 6-7 of reviewer 2. Maybe could be described as a rugged "phenotype-interaction map", in analogy to the idea of "genotype-phenotype maps"? By the way, this is a big assumption of the model, not an established empirical fact, but still an interesting basis for the theory.

We have reorganized introduction and discussion to help the reader to see this work in the broader context. This includes also a new Figure 1 that should make more clear the idea of coarse-graining to the interaction-level. Moreover, we have provided a substantial new Appendix-13 on the genotype-phenotype map. Please see also the response to the second comment of the editor.

Reviewer 2 and I were confused by statements about limiting resources and the competitive exclusion principle. In the revision, the authors still make statements like "GLV equations model competition over renewable resources" (Subsection “Model”), "we observe high diversity in a well-mixed homogenous system without violating the competitive exclusion principle" (subsection “Trade-off anchors eco-evolutionary dynamics in physical reality”) and "we have assumed a single, limiting resource" (Subsection “Power and limitations of ITEEM”). Such statements will misdirect many readers into thinking about resource competition, R* rules, and the impossibility of coexistence of more species than resources. This is not appropriate, because in this model, the species interactions are direct (interference competition) and idiosyncratic. Allelopathy among plants or microbes would be a more relevant example than resource competition. The authors should remove all mentions about resources in the paper, because they will only confuse readers.

To avoid misdirection of readers we have in the revision stressed that the model is not a consumer-resource model, we avoid the expression “limiting resource” (except for one appropriate occasion at the end of the Discussion section), and we describe in the subsection “Model” how the model deals with resources. We cannot avoid the term “resource” completely because trade-off as a key concept of the manuscript is about options for resource allocation (competitive ability vs replication).

Concerning my previous comment 4, please don't portray other theoretical results as empirical support for your new model. Keep them clearly distinct.

We removed all the citations to theoretical models and replaced them with empirical studies, except for those that contain both empirical and theoretical results.

It's a big stretch to say claim "life-history trade-offs" are a missing ingredient in existing theory (Introduction). Almost all existing eco-evolutionary theory is built around trade-offs. Models without trade-offs are the exception, not the rule.

We revised the Introduction accordingly, see paragraphs starting “To identify candidate mechanisms …” and “An essential component missing.

The references cited in subsection “Model” do not represent current, mainstream ecological thinking.

We agree with the reviewer, but we think that these publications are relevant in the current context and that they should be known more widely.

Reviewer #4:I think that the authors addressed the main points from my previous review, just a few minor issues remain.I still disagree with the definition of “strong' and “weak' tradeoff limits, seeing, for example, plots in Figure 1 as completely symmetric. In my opinion, the strongest dependence of birthrate on competitiveness happens at the line in the middle of the plot, presumably for δ=0.5. I guess it's more a terminological discussion, however, I find the definition of the strong tradeoff adopted by the Authors rather confusing.

To avoid confusion, we no longer use the term trade-off strength in the main text. Appendix-2 explains that the outcome of eco-evolutionary dynamics is not only determined by the slope of the trade-off function (dependence of birthrate on competitiveness) but also by the community context.

Caption to Figure 2, “Disc diameter scales with total abundance of species' Does it mean that it scales with the number of individuals in a species? Or the number of species in the system? What kind of scaling is it?

The disc diameter 𝑑_𝑖_of species 𝑖 indicates the relative abundance of the respective species:

di=N1ind∑αϵinα=∑αϵixα,

with 𝑛_𝛼_the abundance of strain 𝛼 ∈ 𝑖, 𝑁_𝑖𝑛𝑑_the total number of individuals in the system, and 𝑥_𝛼_the fraction of individuals belonging to strain 𝛼.

This size of 𝑑_𝑖_is proportional to the relative abundance of species in Muller diagram (Figure 3A in the current version). For a better visualization, all 𝑑_𝑖_sizes are scaled with a same arbitrary factor to avoid disc overlaps or very small disc diameters. To clarify this, a short description has been added to the caption of Figure 3 (in the current version).

Subsection “Generation of diversity”, “Occasionally diversity collapses from medium levels abruptly to very low levels, usually followed a recovery”, should it read “by a recover”?

Corrected.

Appendix 1 (Eq.8) Is γ the index of summation, running from one to N_st? If not, what is the index?

Yes, and it is now corrected.

I don't know if I should dwell on that, yet I also strongly disagree with the Authors' reply to the second comment of the first reviewer, and especially, with the apparent misuse of the term “mean field'.In short, I believe that both implementations of this process, the individual-based and continuous-populations models should yield very similar phenomenology. Both those implementations are mean field in their nature as neither has any spatial correlations (in phenotypic or geographical space) or long temporal memory. The only difference between those is the presence of some stochasticity or noise in the individual-based model. If the Authors truly believe that such noise is the necessary source of the observed phenomenology, it should be clearly stated in the manuscript. Which, I believe, would have strongly depreciated the generality of conclusions. However, I don't think this is the case; on contrary, a continuous population version of this model would have enabled one to get “cleaner' results, speeding up the simulations and expanding the scaling range by including more species. The main distinct features that the model develops, such as temporal changes in the population, interaction cycles, speciations, etc., do not appear to be fluctuation-dominated. A minor related comment, contrary to what is said int he appendix, the per capita death rate can be included into the continuous description (which is often also called the logistic model as all elements of matrix A are negative) by simply reducing the birth rate.

Regarding the usage of the term “mean-field” it is necessary to mention that we had used it in the spirit of statistical mechanics, which is different from the usage in some population biology papers where the term refers to non-spatial models (McKane and Newman, 2004). In statistical mechanics a “mean-field model” is a model that replaces all interactions between constituents (here: individuals) by an average interaction, i.e. an interaction between individuals and other components (Brézin, 2010). In this approach the cross correlations between individual degrees of freedom are neglected and the product of independent random variables can be approximated by the product of their means. Such a mean field model is also applicable to spatial models (McKane and Newman, 2004). Deterministic models at the population-level, like Lotka-Volterra equations, logistic models and resource-competition models, are, in the statistical mechanics spirit, the mean-field approximation of their individual-based equivalents. To avoid confusion, we have replaced in the revised manuscript the term “mean-field” by “population-level”.

Regarding the minor comment about death rate, we agree with the reviewer. For a better comparison between the individual-based ecological dynamics and the presented population-level model we have introduced this term into Equation 6 in Appendix-3.

Regarding the point about comparison of individual-based and continuous-populations models, please see the responses to the first comment by the editor, and to the first comment by reviewer 1.